# Hydrolase mimic via second coordination sphere engineering in metal-organic frameworks for environmental remediation

Xin Yuan[1,4], Xiaoling Wu[1,4] ✉, Jun Xiong[1], Binhang Yan[2], Ruichen Gao[1], Shuli Liu[1], Minhua Zong[1], Jun Ge[3] ✉ & Wenyong Lou[1] ✉

Enzymes achieve high catalytic activity with their elaborate arrangements of amino acid residues in confined optimized spaces. Nevertheless, when exposed to complicated environmental implementation scenarios, including high acidity, organic solvent and high ionic strength, enzymes exhibit low operational stability and poor activity. Here, we report a metal-organic frameworks (MOFs)-based artificial enzyme system via second coordination sphere engineering to achieve high hydrolytic activity under mild conditions. Experiments and theoretical calculations reveal that amide cleavage catalyzed by MOFs follows two distinct catalytic mechanisms, Lewis acid- and hydrogen bonding-mediated hydrolytic processes. The hydrogen bond formed in the secondary coordination sphere exhibits 11-fold higher hydrolytic activity than the Lewis acidic zinc ions. The MOFs exhibit satisfactory degradation performance of toxins and high stability under extreme working conditions, including complicated fermentation broth and high ethanol environments, and display broad substrate specificity. These findings hold great promise for designing artificial enzymes for environmental remediation.

Degradation of refractory carcinogenic contaminants is important for environmental remediation, for example, the detoxification of toxins as well as many other toxic or carcinogenic organics. Enzymes, capable of catalyzing various thermodynamically challenging biochemical reactions under facile conditions, hold great promise in the field of environmental protection[1]. Typically, amide bonds, which are remarkably stable under physiological conditions, with estimated half-lives of 350–600 years, can be efficiently hydrolyzed by enzymes in biological systems[2]. Nevertheless, the retention of enzymatic activity in complicated environmental scenarios still remains challenging.

Enzymes tend to lose their three-dimensional structure when exposed to harsh conditions such as pH extremes, high ionic strength and polar organic solvents[3]. In real-world applications, wastewater from the biotechnology field, for example, fermentation wastewater,

contains refractory ochratoxin with amide bonds and is usually accompanied by high salt concentration and high ethanol concentration, the conditions of which destroy the delicate and complicated three-dimensional structure of native enzyme and result in failure of enzymatic degradation of toxins. Therefore, designing artificial enzymes with a simple and stable structure that mimics the binding pocket and catalytic active sites of natural enzymes is promising for the detoxification of toxic pollutants and environmental remediation.

Homogeneous catalysts serving as artificial enzymes based on metal complexes often suffer from loss of reactivity and difficult reuse due to their instability under working conditions[4]. To circumvent these limitations, insoluble metal-organic frameworks (MOFs), which have been discovered to mimic enzymes, have received considerable attention[5–7] in recent years. MOFs are self-assembled closed host

[1]Lab of Applied Biocatalysis, School of Food Science and Technology, South China University of Technology, 510640 Guangzhou, Guangdong, China. [2]Department of Chemical Engineering, Tsinghua University, 100084 Beijing, China. [3]Key Laboratory of Industrial Biocatalysis, Ministry of Education, Department of Chemical Engineering, Tsinghua University, 100084 Beijing, China. [4]These authors contributed equally: Xin Yuan, Xiaoling Wu. ✉e-mail: wuxl18@scut.edu.cn; junge@tsinghua.edu.cn; wylou@scut.edu.cn

architectures with abundant accessible porous cavities that allow free exchange of solvent molecules, substrates and products. The confined reaction sites located inside MOFs enable electrostatic stabilization of substrate-catalyst intermediates, which endows them with good catalytic activity and selectivity. Despite encouraging achievements, over 90% of the reported MOF-based artificial enzymes have been focused on modeling redox enzymes[8–10], and MOF-based artificial hydrolytic enzymes (Supplementary Table 1) are limited[7,11,12]. Moreover, the enzymatic activities of artificial enzymes are still far from satisfactory. This can be attributed, in part, to the complexity of natural active sites and the limited knowledge of the miscellaneous mechanisms of hydrolases[13], which involve primary coordination spheres and second coordination spheres. Artificial hydrolytic enzymes have thus far been designed to tune the microenvironment and electronic state of a single metal node from the perspective of the primary coordination sphere[14]. However, beyond the first coordination sphere, the second coordination spheres comprising weak interactions of the active pockets in native enzymes[15] have not been fully investigated with artificial enzymes, which involve hydrogen bonding catalytic processes. In native hydrolases, hydrogen bonds formed between the catalytic triad make serine highly nucleophilic and activate the carbonyl carbon in the substrate, and proton transfer between amino acids is the key feature of the catalytic process[16]. Therefore, precisely controlling the secondary coordination sphere adjacent to the active site of the artificial enzyme via hydrogen bonding, and thereby modulating the catalytic efficiency, remains a major challenge.

Inspired by natural hydrolytic enzymes, we proposed a strategy involving the preorganization of amino acids and organic ligands of MOFs in a single scaffold, where zinc ions in the primary coordination sphere followed a Lewis acid-mediated pathway, and the second coordination sphere generated a new active site for amide hydrolysis via hydrogen bonding-enabled oxygen anions. To realize such an architecture, zinc-based MOFs are required to provide an accessible active metal site, and organic ligands featuring abundant nitrogen atoms not directly coordinated to metal ions serve as potential hydrogen bond interaction sites in the second coordination sphere. Amino acids bearing hydroxyl groups serve as hydrogen bond donors and are anchored nearby via coordination of the carboxylate with metal ions, which satisfies the design of our strategy.

Here, we report a zinc azolate framework (ZAF) artificial hydrolytic enzyme prepared via in situ assembly, and a serine with a hydroxyl group side chain can be simultaneously introduced to generate ZAF(Ser). The carboxylate group facilitates the anchoring of the amino acid in the adjacent microenvironment near the primary coordination sphere. The hydroxyl group of serine and the dangling nitrogen from the pristine ligand of the ZAF support the formation of hydrogen bonds in the secondary coordination sphere (Fig. 1). To our delight, the resulting hydrogen bond endows ZAF(Ser) with 3–21 times higher catalytic activity than its parent counterpart. The formation of the hydrogen bonds is confirmed by a combination of experiments and molecular simulations. Computational modeling of the reaction is carried out to support the Lewis acid-mediated and hydrogen bond-mediated hydrolysis pathways of amide bonds catalyzed by ZAF(Ser), which resembled the catalytic processes of metallohydrolases. The hydrogen bond formed in the secondary coordination sphere exhibits 11-fold higher hydrolytic activity than the Lewis acidic zinc ions. The

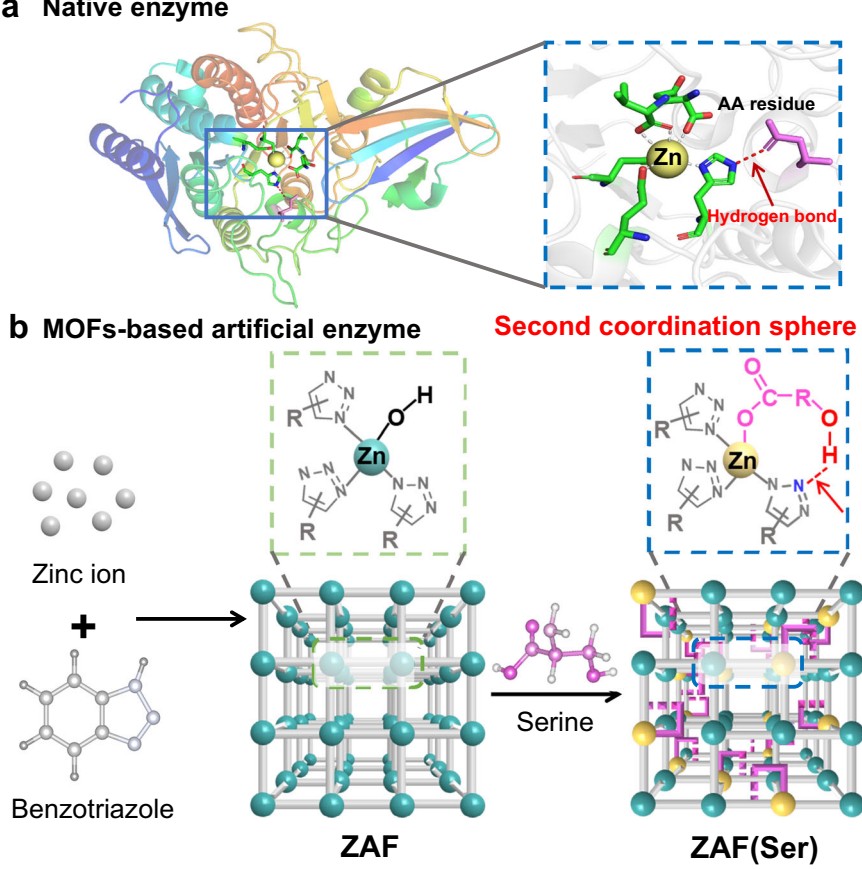

**Fig. 1 | Design of the artificial hydrolytic enzyme ZAF(Ser). a** Schematic illustration of natural hydrolytic enzyme carboxypeptidase (CPA) and its active site, in which the imidazole group in a metal complex forms a secondary coordination sphere with the adjacent amino acid (AA) residue. **b** Dual catalytic active site moieties are located in the primary coordination sphere and secondary coordination sphere of ZAF(Ser), which resemble the Lewis acidic active site and the typical oxygen anion active site mediated by hydrogen bonding in hydrolase, respectively.

porous and rigid framework coupled with dual highly accessible active sites allows for highly efficient hydrolysis of amide-containing pollutants and satisfactory stability. Given this versatility, we envisage the strategy presented herein as a potential alternative for the design of artificial enzymes and protein engineering in the future.

## Results

### Synthesis and structural characterization of ZAF(Ser)

A solvothermal reaction of benzotriazole (BTA) and L-serine (Ser) with zinc nitrate afforded white powders of serine-incorporated zinc azolate frameworks, which are denoted as ZAF(Ser). For comparison, samples with BTA and serine as the single ligand were prepared via a similar procedure and were denoted as ZAF and ZnSer, respectively. ZAF with different amino acids incorporated, denoted as ZAF(X), was fabricated following a similar process. ZAF(Ser) showed the highest catalytic performance and was obtained after optimization of the

reaction conditions (Supplementary Figs. 1 and 2 and Supplementary Table 2) and used for further characterization unless stated otherwise.

Scanning electron microscopy (SEM) and transmission electron microscopy (TEM) experiments revealed that ZAF(Ser) exhibited irregular particles with diameters of 200 nm (Fig. 2a, b), consistent with dynamic light scattering (DLS) results (Supplementary Fig. 3). The bumps on the particle surfaces are expected to generate a large surface area, which would provide abundant accessible active sites. Energy-dispersive X-ray spectroscopy (EDS) mapping (Fig. 2b) and EDS line scans across the particle (Supplementary Fig. 4) confirmed the uniform distributions of C, N, O and Zn in ZAF(Ser). To eliminate the possibility that O came from the adsorbed $H_2O$ of the air, EDS mapping of ZAF and ZAF(Ser) by using much longer scanning time to obtain stronger signals was carried out, which suggested that without serine incorporated, the signal of O observed in ZAF was extremely low and can be considered as background noise (Supplementary Fig. 5). Thus,

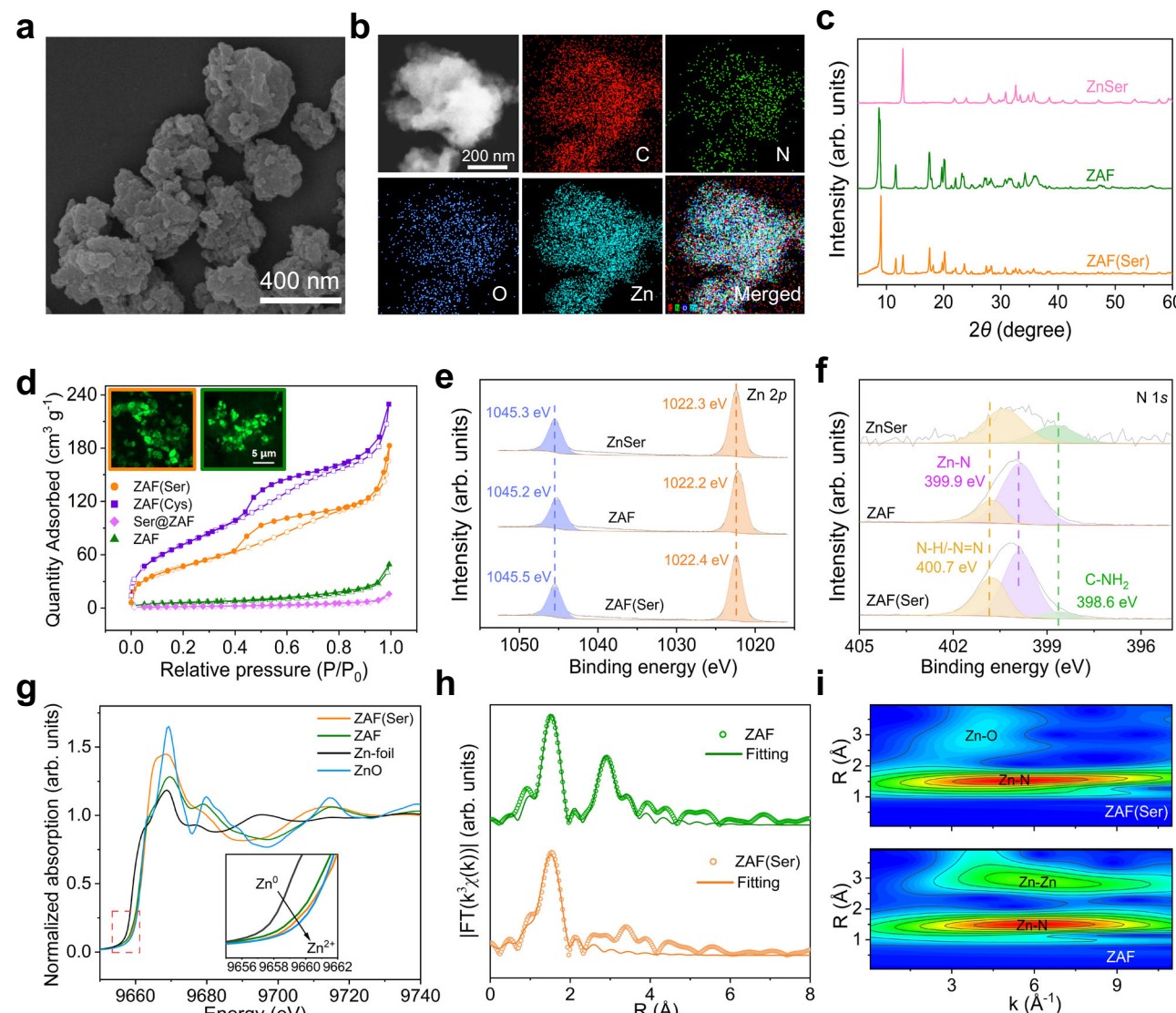

**Fig. 2 | Structural characterization of ZAF(Ser). a** SEM images of ZAF(Ser). The experiment was independently repeated three times with similar results. **b** HAADF-STEM images of ZAF(Ser) and the corresponding EDX elemental mapping images for C, N, O and Zn. The experiment was independently repeated three times with similar results. **c** PXRD patterns for ZAF(Ser), ZAF and ZnSer. **d** Nitrogen sorption curves for ZAF, ZAF(Ser), ZAF(Cys) and Ser@ZAF. The inset shows confocal laser scanning microscopy images of fluorescein molecules

entering the pores of ZAF(Ser) (left) and ZAF (right). The experiment was independently repeated three times with similar results. High-resolution Zn 2*p* XPS spectra (**e**) and N 1*s* XPS spectra (**f**) of ZAF(Ser), ZAF and ZnSer. Normalized Zn *K*-edge XANES spectra (**g**) and corresponding Fourier transform EXAFS fitting curves (**h**) of ZAF(Ser) and ZAF. **i** Wavelet transforms for the $k^3$-weighted Zn *K*-edge EXAFS signals of ZAF(Ser) and ZAF. Source data are provided as a Source Data file.

the uniform distribution of O in ZAF(Ser) suggested homogeneous dispersion of serine throughout the ZAF(Ser).

The powder X-ray diffraction (PXRD) pattern of ZAF(Ser) resembled that of ZAF after serine incorporated, consistent with previous studies[9,17] (Fig. 2c), which was further validated by Fourier transform infrared (FT-IR) spectroscopy (Supplementary Fig. 6 and Supplementary Note 1). A small characteristic peak of Znser suggested the presence of a tiny amount of ZnSer nanoparticle inside. The unique stretching vibration bands of Zn-N (550 cm$^{-1}$) and Zn-O (435 cm$^{-1}$), the latter of which was absent from a physical mixture of serine and ZAF, were both observed in ZAF(Ser) and verified the hybrid coordination microenvironment of zinc[18,19]. The $^{1}$H NMR and $^{13}$C NMR signals (Supplementary Figs. 7 and 8) of acid-digested ZAF(Ser) further demonstrated the presence of the coordinated ligands of BTA and Ser. $^{1}$H NMR of ZAF and ZnSer proved the presence of the corresponding ligands. (Supplementary Figs. 9 and 10). Thermalgravimetric analysis (TGA) showed that ZAF(Ser) was stable up to 470 °C (Supplementary Fig. 11) and exhibited higher thermal stability than its parent counterpart ZAF and ZnSer, which was attributed to the hybrid coordination states of zinc. Nitrogen sorption isotherms revealed that both ZAF(Ser) and ZAF contained mesopores (Fig. 2d).

The chemical states of ZAF(Ser) artificial enzyme were analyzed by X-ray photoelectron spectroscopy (XPS) (Supplementary Figs. 12 and 13). High-resolution Zn 2$p$ XPS spectra showed that the Zn 2$p$ binding energy of ZAF(Ser) (1022.4 eV) shifted toward higher energy compared with that of ZAF (1022.2 eV) (Fig. 2e), which suggested a higher partial positive charge for zinc due to electronic interactions between zinc and the oxygen from serine[20,21]. In the high-resolution N 1$s$ spectrum, a peak at 399.9 eV ascribed to Zn-N was observed for ZAF and ZAF(Ser) but was absent from the spectrum for ZnSer (Fig. 2f). Notably, the high-resolution O 1$s$ spectrum of ZAF(Ser) was deconvoluted into three individual peaks at 531.5, 532.5 and 534.0 eV, which were assigned to C=O, C-O and Zn-O moieties[22,23], respectively (Supplementary Fig. 13), which further suggested that the serine in ZAF(Ser) was anchored via coordination of carboxylate groups with zinc ions.

To distinguish the electronic and coordination states of the zinc species, X-ray absorption near-edge structure (XANES) and extended X-ray absorption fine structure (EXAFS) analyses were carried out. Compared with ZnO with a single coordination environment, ZAF(Ser), which contained both coordination structures of Zn-N and Zn-O, led to different chemical environments of Zn with slightly different energy levels, resulting in a broad white line. As indicated by the $K$-edge XANES spectra (Fig. 2g), ZAF and ZAF(Ser) exhibited absorption edge with energy lower than that of ZnO, suggesting a positive valence of less than +2. Compared with ZAF, The $K$-edge absorption edge of ZAF(Ser) is closer to that of ZnO with high energy, indicating a slightly higher average valence state of Zn in ZAF(Ser) than that in ZAF. In the normalized Fourier transform k$^{3}$-weighted EXAFS (FT-EXAFS) spectrum (Supplementary Fig. 14), ZAF exhibited an obvious peak located at 1.52 Å, which was attributed to the Zn-N scattering path, and small scattering peaks derived from Zn-Zn coordination were also observed[24]. Moreover, a shoulder peak located at 1.54 Å in ZAF(Ser) (Fig. 2h) was associated with the Zn-N and Zn-O scattering paths. The Zn-N/O coordination number in ZAF(Ser) was increased to 3.5, compared with 3.0 for ZAF (Fig. 2h, Supplementary Table 3), which was in good agreement with the XPS results. Furthermore, the higher intensity maximum at 4.95 Å$^{-1}$ in the k space associated with the Zn-N/O scattering path for ZAF(Ser) was observed in the EXAFS wavelet transforms (Fig. 2i and Supplementary Fig. 15). The contour intensity maximum for ZAF(Ser) was shifted slightly compared with that of ZAF due to the introduction of serine.

## Hydrogen bond-dependent hydrolytic activities of artificial enzymes

The intrinsic hydrolytic activity of ZAF(Ser) was evaluated by hydrolyzing hippuryl-L-phenylalanine (HPPA) to generate hippuric acid (HA) with an absorbance wavelength of 254 nm (Supplementary Figs. 16 and 17). As expected, in the presence of ZAF(Ser), ZAF and ZnSer, hydrolysis of HPPA was observed, as indicated by the increased absorbance at 254 nm (Fig. 3a). In contrast, the substrate solution without catalyst did not show obvious hydrolysis. To confirm the role of the artificial enzyme, a control experiment was also carried out with a physical mixture of the building blocks (Fig. 3a and Supplementary Fig. 18). Zinc ions alone showed almost negligible hydrolytic activity. None of the building block combinations showed any significant hydrolytic activity in HPPA hydrolysis. Interestingly, ZAF(Ser) with a small amount of serine incorporated exhibited 3.2-fold higher catalytic activity than ZAF, while the activity of ZnSer for hydrolysis of HPPA was only 12% that of ZAF(Ser).

The vast difference in activities of ZAF(Ser) and ZAF was investigated in depth. It was speculated that, during the catalytic processes of artificial metalloenzymes, increased surface area or enhanced Lewis acidity at the metal active site would allow the artificial enzyme to accelerate hydrolysis of the amide substrate[25,26]. Brunauer-Emmett-Teller (BET) specific surface areas of 174.8 m$^2$ g$^{-1}$ and 21.9 m$^2$ g$^{-1}$ were determined for ZAF(Ser) and ZAF, respectively, which is in agreement with previous studies that ZIFs with triazoles as the ligand exhibited essentially low surface area[27–29]. Mesopores with sizes in the range of 2–12 nm were indicated by porosity distribution calculations performed with the DFT method (Fig. 2d, Supplementary Fig. 19 and Supplementary Table 4). Despite the relatively lower surface area of pure ZAF compared with ZAF(Ser), the mass transfer rate monitored with a fluorescein probe and confocal laser scanning microscopy showed negligible differences in fluorescein distribution and fluorescence intensity (inset of Fig. 2d and Supplementary Fig. 20), implying that mesopores in ZAF(Ser) and ZAF provided transport pathways for diffusion of substrates and thus facilitated entrance of the substrate into the accessible active site of the catalyst during the hydrolytic process. Furthermore, ZAF(Cys) with cysteine substituted for serine was fabricated by following the same procedure (Figs. 2d, 3b, Supplementary Fig. 19), which featured a 16 times larger surface area than that of ZAF but exhibited no evident difference in activity compared with ZAF under the identical amount of zinc. Thus, increased surface area did not lead to activity enhancement, which was further confirmed by the case of alanine (Ala), where ZAF(Ala) bearing 5 times higher surface area exhibited almost identical catalytical activity to that of ZAF (Supplementary Fig. 21, Supplementary Table 4). Thus, the enhanced catalytic activity of ZAF(Ser) compared with ZAF could not be simply ascribed to the increased surface area. Furthermore, MOFs with missing linker or with electron-withdrawing group on the linker have been reported to increase the Lewis acidity of metal sites[30]. Thus, ZAF(X) with amino acids bearing various side chain groups other than OH was prepared (Supplementary Figs. 22–24). ZAF(Asp) and ZAF(Glu) with carboxylate groups showed activities similar to that of ZAF at identical Zn content, although the surface area was ~3 times higher than that of ZAF (Supplementary Table 4 and Supplementary Fig. 21). Moreover, by normalizing the reaction rate with specific surface area, the catalytic activity of ZAF(Asp), ZAF(Glu) is almost identical to that of ZAF(Ser) (Supplementary Table 5), implying that change of amino acid with different electron-withdrawing capability and the resultant enhanced Lewis acidity did not lead to activity change of the obtained composite. This was further confirmed by the almost unchanged activity of ZAF(Cys) with varying ratio of cysteine to BTA during the synthetic process to regulate the Lewis acidity of Zn in the final composite (Supplementary Fig. 25). Thus, the possibility of enhanced Lewis acidity after the hybrid coordination of amino acid that led to activity increase is ruled out. These results together indicated that the

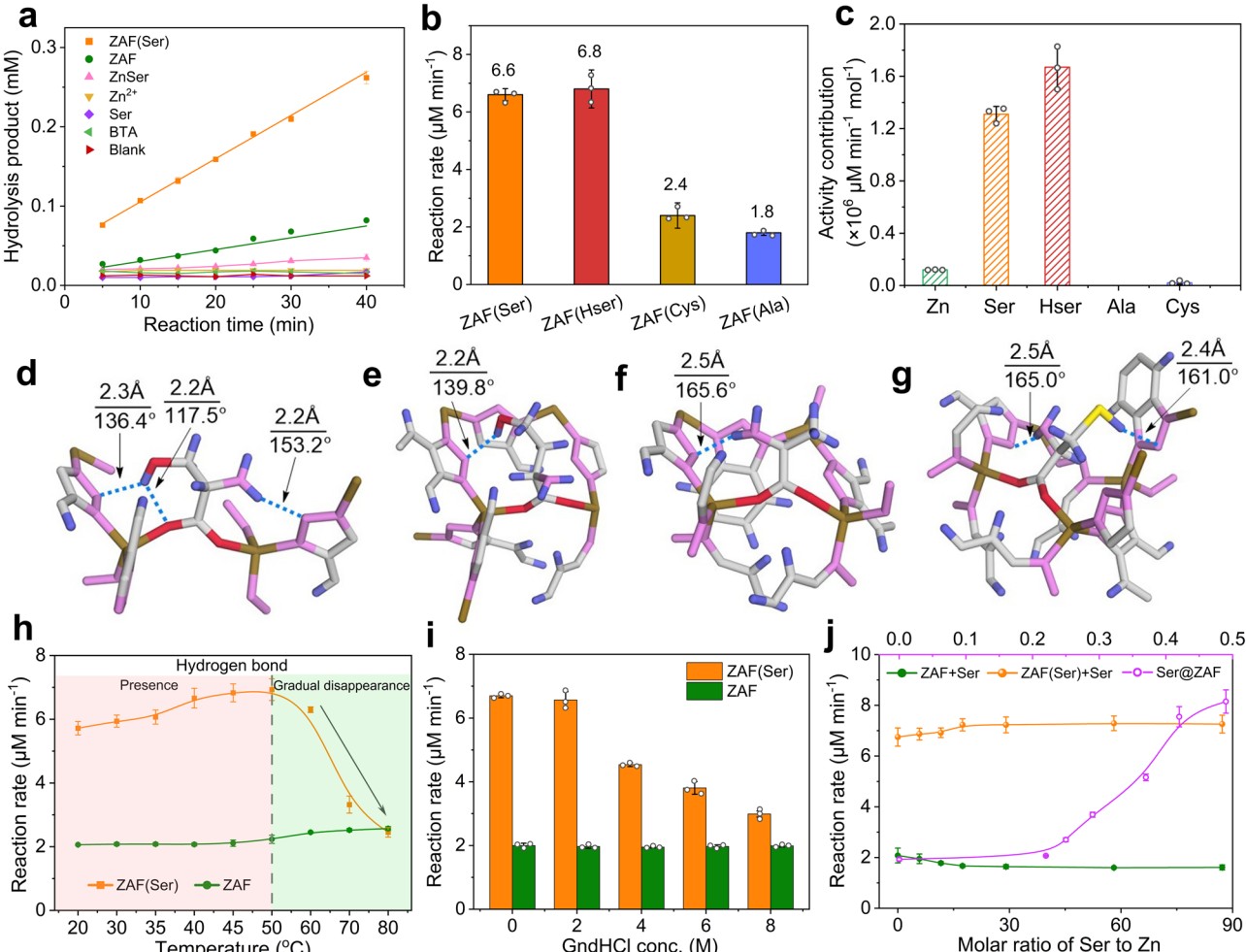

**Fig. 3 | Hydrogen bonding-dependent activity of ZAF(Ser). a** Control experiments showing the catalytic activities of ZAF(Ser), ZAF, ZnSer and the corresponding building blocks under identical concentrations. **b** Catalytic activities of ZAF(Ser), ZAF(Hser), ZAF(Ala) and ZAF(Cys). Data were represented as mean ± SD ($n = 3$). **c** Incremental activity per mole of zinc ions in ZAF and incremental activity per mole of amino acid with the same zinc content in ZAF(Ser), ZAF(Hser), ZAF(Ala) and ZAF(Cys). Data were represented as mean ± SD ($n = 3$). **d**–**g** Hydrogen bonding pattern formed in models of **d** ZAF(Ser), **e** ZAF(HSer), **f** ZAF(Ala), and **g** ZAF(Cys). The red, light gray, blue, deep yellow, carmine and yellow, orange spheres represent O, C, H, Zn, N, and S atoms, respectively. The blue dashed line indicates hydrogen bonding. **h**, **i** The presence and gradual disappearance of hydrogen bonds induced by temperature (**h**) and guanidine hydrochloride (GndHCl) (**i**) led to changes in the catalytic activities of ZAF and ZAF(Ser). Data were represented as mean ± SD ($n = 3$). **j** Catalytic activity of serine incorporated in ZAF (Ser@ZAF), a physical mixture of serine and ZAF(Ser) (ZAF(Ser)+Ser) and a physical mixture of serine and ZAF (ZAF+Ser) with varying concentrations of serine. Data were represented as mean ± SD ($n = 3$).

remarkably higher activity of ZAF(Ser) cannot simply be ascribed to the defects and increased surface area due to the incorporation of the small amino acids or the regulated Lewis acidity of the zinc ions in the primary coordination sphere but was generated by the structure of ZAF(Ser), which points to the presence of hydrogen bonds involving OH groups in serine and dangling N atoms from BTA, as inspired by the native enzyme. Before going further, the possibility that the formation of hydrogen bonding resulting from the residual solvent molecules but not the incorporated amino acid was eliminated by using deuterium water ($D_2O$) instead of water used during the synthesis of ZAF(Ser) and deuterated methanol ($CD_3OD$) instead of methanol used during the washing process (Supplementary Figs. 26–29). Results showed that even with residual solvent inside, the unwashed ZAF did not show any activity enhancement (Supplementary Fig. 29). And after the washing process, the amount of solvent molecule left inside the catalyst is negligible (Supplementary Note 2).

Therefore, ZAF(X) with amino acids with chemical structures similar to that of serine but with different side chains were further compared, including serine (-OH) for ZAF(Ser), cysteine (-SH) for

ZAF(Cys), alanine (-H) for ZAF(Ala), and homoserine (-$CH_2OH$) for ZAF(Hser). ZAF(Ala) and ZAF(Cys) without hydrogen bonding exhibited catalytic reaction rates almost identical to that of pure ZAF with the same amounts of zinc ions (Supplementary Table 6). In contrast, ZAF(Hser) showed 3.3-fold higher activity than pure ZAF (Fig. 3b), which was slightly higher than that of ZAF(Ser) with the same concentration of zinc ions due to hydrogen bonding in the presence of OH groups. Thus, the activity of ZAF originated from only one Lewis acidic active site of zinc, while the activity of ZAF(X) can be divided into two kinds of active site, the Lewis acidic active site of zinc and the amino acid involved active site. The quantitative activity contribution of zinc was obtained by dividing the activity of ZAF by the amount of zinc, while the quantitative activity contribution of each amino acid involved active site was calculated by subtracting the contribution of zinc from the total activity of ZAF(X). Calculation result revealed that each serine or homoserine involved catalytic active site in ZAF(Ser) and ZAF(HSer) exhibited a $1.3 \times 10^6$ μM min$^{-1}$ mol$^{-1}$ and $1.7 \times 10^6$ μM min$^{-1}$ mol$^{-1}$ in hydrolysis rate, which was 11-fold and 14-fold higher than the catalytic activity of zinc ions serving as Lewis acidic active sites

(Fig. 3c). While the activity contribution of cysteine and alanine involved active site was almost negligible. Thus, the steric structure of the secondary coordination sphere as well as hydrogen bonding endowed ZAF(Ser) with extraordinary catalytic activity.

Following the experiments and the hypothesis that hydrogen bonds were responsible for catalysis, we carried out theoretical calculations to examine hydrogen bond formation inside the framework. The distances of the hydrogen bonds were analyzed by first-principles calculations using the SIESTA package[31]. A single crystal X-ray diffraction analysis (SCXRD) was first carried out, which revealed that ZAF crystallized with the formula Zn(BTA)$_2$ in a monoclinic crystal system[32] with space group P 21/c (Supplementary Table 7 and Supplementary Fig. 30). Despite our attempt to obtain the single crystal of ZAF(Ser), the quantitative measurement of serine in ZAF(Ser) single crystal was calculated to be ~2.7 wt% based on the result of ICP-OES and elemental analysis (Supplementary Table 8). Such a low content of serine implied that only partial zinc coordinated with serine inside the single crystal particle randomly and generated much lower diffraction intensity compared with other facets intrinsic to ZAF. The presence of serine was confirmed by combined $^1$H NMR (Supplementary Fig. 31) and FT-IR analyses (Supplementary Fig. 32).

In the framework, the zinc ions exhibited four-coordinate geometries and were connected with four nitrogen atoms of the BTA ligands. Each bidentate ligand generated a dangling nitrogen site in situ, which served as a potential hydrogen bonding site for modulation of the secondary coordination sphere. Based on the coordination structure derived from SCXRD, models of serine- (Fig. 3d), homoserine- (Fig. 3e), alanine- (Fig. 3f), and cysteine-incorporated ZAFs (Fig. 3g) were estimated and optimized (Supplementary Fig. 33). The side chain groups of all investigated amino acids were located near the nitrogen of BTA, which enabled formation of hydrogen bonds, and the distance between the hydrogen bond acceptor atom and the hydrogen atom did not exceed 2.5 Å, and the angle was larger than 90°[33]. The distance between the nitrogen of BTA and the hydrogens of serine (Fig. 3d) and homoserine (Fig. 3e) were 2.3 Å and 2.2 Å, respectively, and the angles were larger than 90°, which favored hydrogen bonding[33]. In the cases of alanine and cysteine, no hydrogen bond-mediated hydrolysis of HPPA was expected due to the absence of hydrogen donors, and this was confirmed by preparing the corresponding experimental samples and carrying out enzymatic assays ($^1$H NMR in Supplementary Figs. 34–36). The slightly higher activity contribution of homoserine than that of serine (Fig. 3c) can be attributed to the shorter distance between nitrogen and hydrogen and a higher possibility of collision and hydrogen bond formation.

To substantiate this hypothesis, the hydrogen bonding interaction inside ZAF(Ser) was disrupted by exposing the catalyst to high temperature (Fig. 3h) and guanidine hydrochloride treatment (Fig. 3i), where guanidine hydrochloride is commonly used for protein denaturation. The catalytic activity of ZAF exhibited negligible variation when the temperature was within the range of 20 to 80 °C, suggesting the high thermal stability of ZAF. However, for ZAF(Ser), when the temperature exceeded 70 °C, the rate for hydrolysis of HPPA showed a drastic decrease (70%) to 2.5 μM min$^{-1}$, indicating that the enhancement contributed by serine disappeared for the intact structure of ZAF(Ser) (Fig. 3h). The drastic decrease in the activity of ZAF(Ser) and the identical catalytic activities for pure ZAF and ZAF(Ser) possibly resulted from the absence of hydrogen bonding between serine and BTA at higher temperatures. Hydrogen bonding between amino acid residues of the protein backbone stabilizes the three-dimensional structure of the native enzyme and determines the catalytic activity. Deactivation of the enzyme at elevated temperature is partly ascribed to the loss of hydrogen bonding, which led to the collapse of the structure that had been stabilized by hydrogen bonding. As with the deactivation of proteins, ZAF(Ser) failed to perform hydrogen bonding-mediated amide cleavage at high temperature due to the

breakage of the hydrogen bond. Benefitting from the reversibility of hydrogen bonding, when ZAF(Ser) was heated at 80 °C and then cooled down to room temperature and exposed to enzymatic assay, it restored all its catalytic activity (Supplementary Fig. 37), which maintained intact structure and high thermal stability (Supplementary Fig. 38). This result verified the presence of hydrogen bonding from another aspect. Another experiment of hydrogen bonding disruption was further carried out by exposing ZAF(Ser) and ZAF to guanidine hydrochloride (GdnHCl) treatment. With increasing concentrations of GdnHCl, the rate for hydrolysis of HPPA catalyzed by ZAF(Ser) decreased gradually to 3.1 μM min$^{-1}$, which was identical to that of pure ZAF (Fig. 3i). ZAF showed similar catalytic activities despite the presence of GdnHCl.

The effect of the serine in ZAF(Ser) on the catalytic activity was investigated further. Control experiments were carried out by physically mixing varying concentrations of serine with defined amounts of ZAF or ZAF(Ser) for enzymatic assays (Fig. 3j), which confirmed that the accelerated hydrolysis was attributable to coordinated serine but not to free serine physically mixed with the catalyst. The possibility that a combination of ZAF and ZnSer (denoted as pseudo-ZAF(Ser)) that led to activity enhancement was also ruled out. The pseudo-ZAF(Ser) with molar ratio of ZAF to ZnSer of 9: 1 was first prepared by physically mixing the powders of ZAF and ZnSer based on the result of ICP-OES and elemental analysis of ZAF(Ser). Both the morphologies of ZAF and ZnSer (Supplementary Fig. 39) were observed in SEM images of pseudo-ZAF(Ser), which were completely different from ZAF(Ser) (Fig. 2a). Activity measurement showed that hydrolysis rate of of pseudo-ZAF(Ser) was ~4 times lower than that of ZAF(Ser) (6.6 μM min$^{-1}$) under identical zinc amount (Supplementary Fig. 40). Furthermore, when this pseudo-ZAF(Ser) was used under a high reaction temperature, no obvious activity change was observed (Supplementary Fig. 41), which is quite different from that of ZAF(Ser) (Fig. 3h), further implying the absence of hydrogen bond-mediated active sites in the pseudo-ZAF(Ser). Furthermore, to eliminate the possibility that ZAF(Ser) presented in the form of ZAF adsorbed serine (denoted as xSer/ZAF). Thus xSer/ZAF with various amounts of serine as the adsorbate and ZAF as the adsorbent was prepared (Supplementary Figs. 42 and 43). Catalytic activity showed that even with serine adsorption capacity of 75.1 mg g$^{-1}$, the catalytic activity of xSer/ZAF was identical to pure ZAF (Supplementary Fig. 42b), indicating that the enhanced catalytic activity of ZAF(Ser) compared with both ZAF and ZnSer could not be ascribed to simple mixture of ZAF and ZnSer particles or ZAF with serine adsorbed.

To quantify the contribution of the hydrogen bonding to the catalytic activity of ZAF(Ser), a series of xSer@ZAF analogs with varying serine contents was prepared via postsynthetic modification, and the subscript x stands for the molar ratio of serine to zinc ions during the synthetic process. PXRD (Supplementary Note 3), FT-IR and TGA characterization revealed that ZAF(Ser) obtained via the one-step self-assembly process and two-step postsynthetic modification method shared the same structure (Supplementary Figs. 44–46). Notably, xSer@ZAF featured a lower surface area than that of ZAF (Supplementary Fig. 47 and Supplementary Table 9), but increasing the serine content in the composite led to proportional activity enhancements from xSer@ZAF (Fig. 3j, Supplementary Tables 9 and 10), which corroborated the vital role of serine in hydrolysis. Careful calculations for ZAF(Ser) with varying contents of serine showed a linear relationship between the incremental activity and the amount of serine incorporated (Supplementary Fig. 48). The linear relationship between the serine content coordinated with zinc and the catalytic activity suggested that serine was involved in hydrogen bonding, which generated an entirely new catalytic active site independent of the Lewis acidic active site in the framework. This pathway is similar to that of the natural hydrolytic enzyme protease[34], but this hydrogen bond-mediated hydrolytic pathway[35] has rarely been reported for artificial

enzymes. In this case, the uncoordinated hydroxyl group in serine was anticipated to form hydrogen bonds with the vacant nitrogen dangling from the BTA ligand in the ZAF secondary coordination sphere. The activated oxygen from serine acted as a nucleophile and attacked the carbonyl carbon of an amide bond, which accelerated amide bond cleavage.

In addition, the role of hydrogen bonding was further confirmed by using serine with OH shielded by acetyl and t-butyl for the synthesis of ZAF(O-Ac-ser) and ZAF(O-tBu-ser) (Supplementary Figs. 49 and 50). ZAF(O-tBu-Ser) and ZAF(O-Ac-Ser) also showed a 3.1-fold and 3.4-fold higher surface area compared with ZAF (Supplementary Fig. 51 and Supplementary Table 4) but featured almost identical activity with ZAF under identical zinc amount (Supplementary Fig. 52). On the contrary, ZAF(Thr) with threonine (Thr) bearing -OH groups (Supplementary Fig. 53) showed almost identical surface area with ZAF, but exhibited a 232% catalytic activity relative to ZAF. Furthermore, the catalytic results of xCys@ZAF obtained following the postsynthetic modification were similar to xSer@ZAF, showing that even under high content of cysteine, xCys@ZAF displayed almost identical catalytic activity to that of ZAF under identical amount of zinc (Supplementary Fig. 54 and Supplementary Table 11). This finding further corroborated the function of the OH group and the vital role of hydrogen bonding in the catalysis.

## Theoretical investigation of the catalytic mechanism for ZAF(Ser)

Inspired by the catalytic mechanisms of metallohydrolases (Supplementary Fig. 55) and the His-Asp-Ser catalytic triad (Supplementary Fig. 56), we speculated that the high hydrolytic activity of ZAF(Ser) for amide hydrolysis was attributable to the presence of two catalytically active sites: the Lewis acidic Zn(II) active site and the hydrogen bonding involved oxygen anion active site, which were incorporated in a single scaffold. Thus, theoretical investigation was performed with density functional theory (DFT) calculations. The configurations of the proposed Lewis acidic Zn(II) active sites of ZAF (Fig. 4a) and ZAF(Ser) (Fig. 4b) and the hydrogen bonding active sites (Fig. 4c) were first optimized with DFT calculations using the PBE0 functional and def2SVP basis set. In the proposed metallohydrolase-like hydrolytic process (Fig. 4f), the amide substrate coordinates to one Zn(II) ion, which is activated by water molecules interacting with the carbonyl oxygen (i, ii). Next, the hydroxide formed by deprotonation of a Zn(II)-bound water (Zn-OH⁻) acts as a nucleophile and attacks the carbonyl carbon to form an intermediate (iii), followed by elimination of the leaving amine and subsequent displacement of the hydrolyzed product by water (iv and v), respectively. In the native metallohydrolase (Supplementary Fig. 41 and Supplementary Note 4), the zinc-bound water is activated by Glu143 and serves as a general base[36], while in ZAF and ZAF(Ser), activation was performed by the triazole groups (Supplementary Figs. 57 and 58 and Supplementary Note 5). The rate-determining step for the metallohydrolase is the breakage of the C-N bond[37]. Notably, in ZAF(Ser), the coordinated serine was involved in the rate-determining step in which the Zn-OH initiates nucleophilic attack of the carbonyl carbon and reduces the free energy below that of ZAF (Fig. 4d). The coordinated serine forms a hydrogen bond-mediated active site to reduce the energy barrier thus accelerating the reaction[37,38], which is consistent with the EXAFS result (Fig. 2h) and activity result of a series of ZAF based composites. Although the detailed structures of different native metallohydrolases vary considerably regarding the specific organization of the peptide sequences and organization of the peptides, their catalytic activities are dependent on the metal/hydroxy species that serve as Lewis acids. The carbonyl oxygen of an amide substrate binds to the Lewis acidic Zn(II) center; this makes the carbonyl carbon atom more susceptible to nucleophilic attack, which enables more rapid hydrolysis of the substrate amide compared with non-catalyzed amide hydrolysis. Serine

coordination with the zinc ion changes the serine electronic structure and regulates the Lewis acidity of the active site[39].

To better elucidate the contribution of the second coordination sphere to the catalytic activity, we considered a second catalytic pathway in which the hydrogen bond deprotonated the hydroxyl group of serine to generate an oxygen anion; this is reminiscent of the His-Asp-Ser triad in the enzyme (Supplementary Fig. 56 and Supplementary Note 6). As with the catalytic triad-enabled hydrolytic process of the enzyme (Fig. 4f, Supplementary Fig. 59 and Supplementary Note 7), nucleophilic attack of the activated oxygen atom on the side chain of serine near the scissile amide carbonyl of the substrate occurs. Breakage of the C-N bond leads to the release of the first product (amine). Water molecules from the solution act as the nucleophiles, and the BTA triazole acts as the general base to regenerate the catalyst and liberate the acid product[33,38].

The Gibbs free energy profile for Zn(II)-mediated hydrolysis with ZAF was calculated, which revealed that attack of the N-H in HPPA by free water molecules constituted the rate-determining step with a free energy increase ($\Delta G_{TS-2}$) of 42.02 kcal mol⁻¹ (Fig. 4d). An energy as low as 7.5 kcal mol⁻¹ was required on going from transition state 1 (TS-1) to transition state 2 (TS-2), which involved hydrolysis by $H_2O$ and formation of intermediate 2 ($\Delta G_{INT-2} = 13.2$ kcal mol⁻¹). Finally, the carboxylic acid-ligated Zn(II) dissociates to release the carboxylic acid product. When serine was introduced into ZAF(Ser), we discerned an alternative conformer for the active site. The small serine caused the rearrangement of BTA coordination with zinc and changed the accessibility of the active site and the Lewis acidity of Zn(II). DFT calculations for the entire reaction pathway showed that the rate-determining step was attack of the hydroxyl group on the carbonyl group, which exhibited a free energy of 39.1 kcal mol⁻¹ (TS-1) (Fig. 4d and Supplementary Fig. 58). Compared with ZAF, the decrease in energy on going from INT-2 to TS-2 suggested that amine release was more preferable with ZAF(Ser). Moreover, the lower energy barrier for ZAF(Ser) relative to ZAF suggested that the Lewis acid-mediated catalytic process in ZAF(Ser) facilitated hydrolysis.

The hydrogen bonding-mediated pathway is the energetically favored catalytic process, and it leads to a substantial decrease in the energy barrier (28.1 kcal mol⁻¹) for the rate-determining step compared with that for the Lewis acid-mediated pathway (42.0 kcal mol⁻¹ for ZAF and 39.1 kcal mol⁻¹ for ZAF(Ser)) (Fig. 4d, e). DFT calculations of several studies[40–43] also confirm the rationality and feasibility of energy calculations in hydrogen bond-mediated catalytic reaction pathways. The adsorption and activation of HPPA occur spontaneously, with the Gibbs free energy decreased by 10 kcal mol⁻¹. Therefore, ZAF follows a single Lewis acid-mediated pathway during the reaction and exhibits relatively poorer activity. In contrast, the serine incorporated in ZAF endowed the resultant ZAF(Ser) with dual catalytic active sites. The hybrid coordination of serine and BTA with zinc in the primary coordination creates a more accessible Lewis acidic Zn(II) active site, which follows the metallohydrolase-like catalytic pathway. More importantly, the orientation of the hydroxy group (OH) in the side chain of coordinated serine and the adjacent nitrogen (N) from coordination of the BTA with zinc leads to the formation of hydrogen bonds in the secondary coordination sphere, which enables ZAF(Ser) to function with high catalytic activity and follow the oxygen anion catalytic process. The hydrogen bond-mediated oxyanions in the secondary coordination sphere are reminiscent of the Lys-Ser catalytic dyad and Asp-His-Ser catalytic triad found in a range of hydrolases, which create distinct catalytically active sites in artificial enzymes. For comparison, the binding energy of typical catalytic triad (Asp-His-Ser) in Asp-His-Ser native enzyme and its mutations of Asp-His-Cys and Asp-His-Ala with the substrates was calculated by molecular docking simulation (Supplementary Fig. 60). The calculation showed that that Asp-His-Ser containing serine and formation of hydrogen bonding is more energy preferred, which is consistent with the experiments obtained in

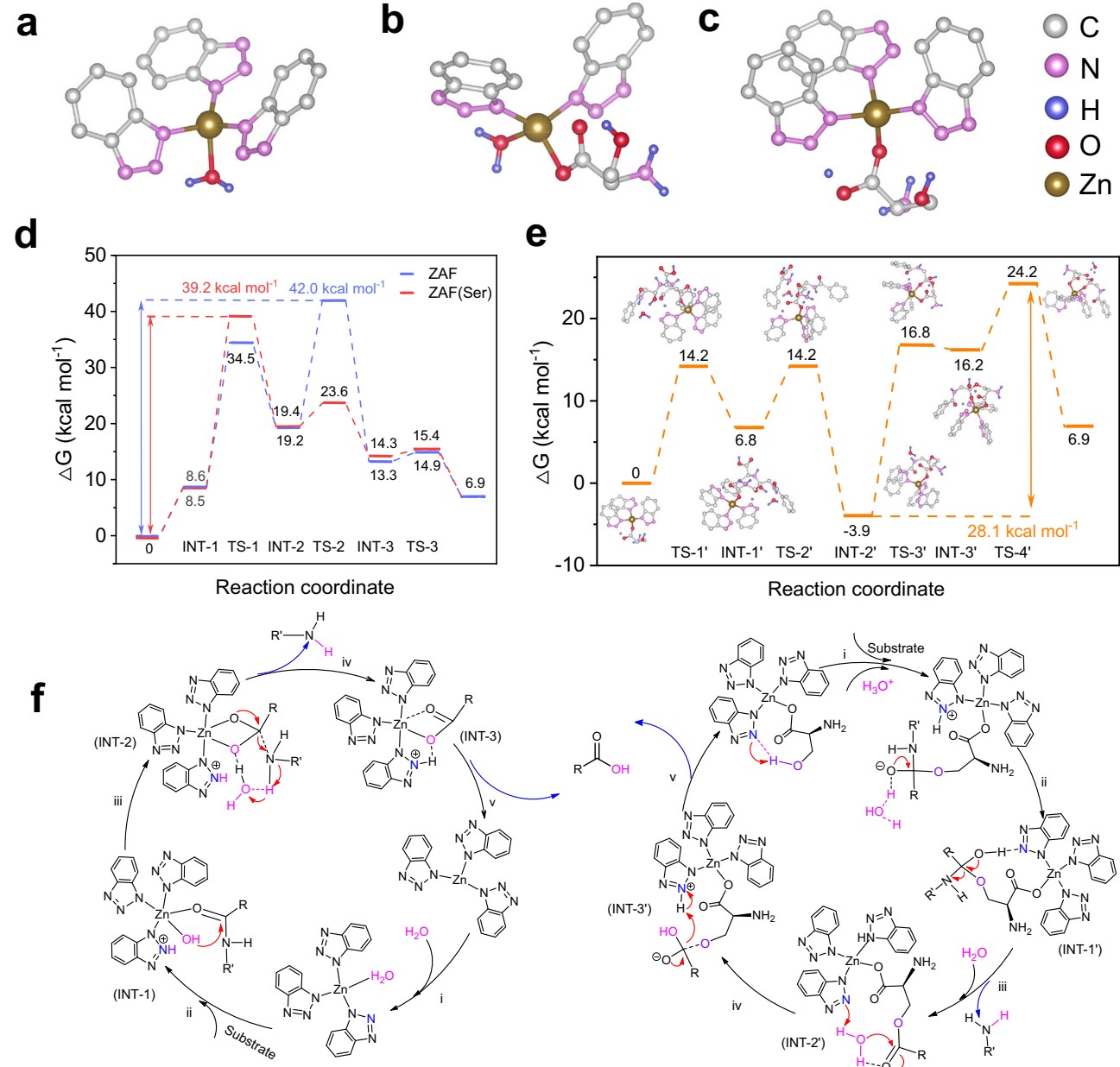

**Fig. 4 | Proposed catalytic reaction pathways of ZAF(Ser).** Schematic structure model of the Lewis acidic Zn(II) active site in ZAF (**a**), ZAF(Ser) (**b**) and the hydrogen bonds mediated active site in ZAF(Ser) (**c**). **d** Comparison of Gibbs free energy profiles for the catalytic processes of ZAF (blue dotted line) and ZAF(Ser) (red dotted line), which are analogous to those of metallohydrolases with Lewis acidic Zn(II)-mediated catalytic pathways. **e** Gibbs free energy profile for the catalytic process of ZAF(Ser) proceeding via the hydrogen bonding-mediated catalytic pathway. **f** Proposed catalytic mechanism of ZAF(Ser) involving the Lewis active site-mediated pathway (left) and hydrogen bonding-mediated pathway (right).

ZAF(Ser), ZAF(Cys) and ZAF(Ala) (Fig. 3d, f, g). Furthermore, the porous structure of ZAF(Ser) facilitates mass transfer during the catalytic process. Simultaneous regulation of the primary and secondary coordination spheres of the metal-organic frameworks generates two catalytic pathways, which provide insights into the enzyme-inspired design of artificial enzymes.

## Catalytic performance of ZAF(Ser)
The catalytic performance of ZAF(Ser) was investigated systematically under various working conditions. ZAF(Ser) exhibited the highest catalytic activity at pH 6.0, similar to the native hydrolase (carboxypeptidase A, CPA). ZAF(Ser) was less sensitive to pH (Supplementary Fig. 61) and maintained over 75% of the highest activity with pH values ranging from 3 to 9, while native hydrolase exhibited

less than 25% of the optimal activity at extreme pH values (pH 3.0 and pH 9.0). When the hydrolytic reaction temperature exceeded 40 °C, the catalytic activity of native hydrolase began to decrease, and less than 20% of the initial activity was maintained once the temperature exceeded 60 °C (Supplementary Fig. 62). For ZAF(Ser), over 80% of the initial activity was retained when the reaction temperature was lower than 60 °C, verifying the high stability of ZAF(Ser) at high temperatures. When the reaction temperature exceeded 70 °C, the absence of the second active site caused by hydrogen bond breakage in the secondary coordination sphere of ZAF(Ser) led to a dramatic decrease in activity.

Maintaining enzyme activity during storage and operation is crucial for the application of artificial enzymes. The catalytic activity of ZAF(Ser) was systematically investigated while exposing it to hostile

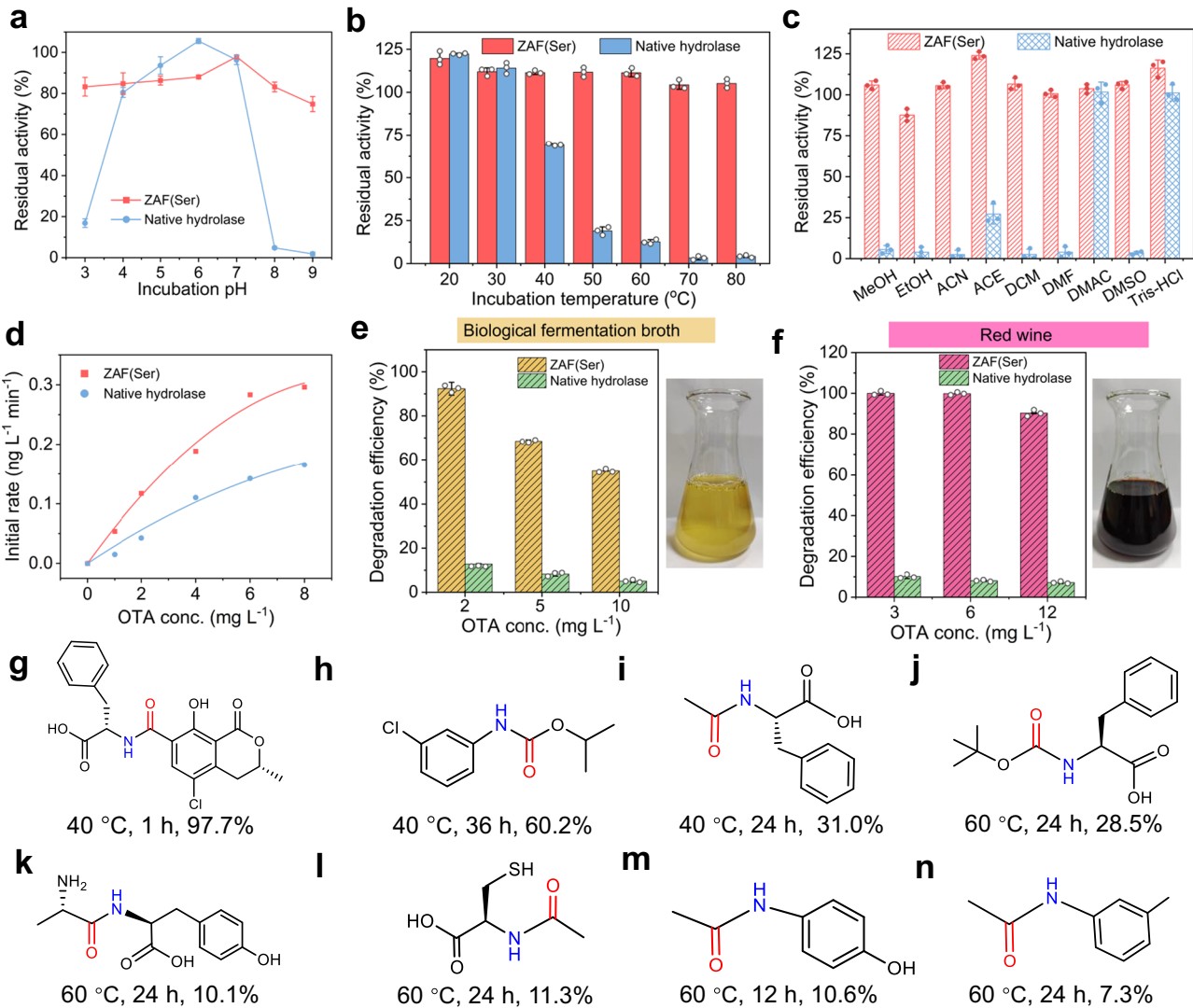

**Fig. 5 | Catalytic stability and substrate universality of ZAF(Ser). a–c** Residual catalytic activity of ZAF(Ser) and native hydrolase at various pH conditions (**a**), incubation temperatures (**b**) and with organic solvents (**c**). Data were represented as mean ± SD ($n = 3$). **d** Comparison of catalytic activities of ZAF(Ser) and native hydrolase for degradation of OTA. **e**, **f** Degradation efficiency of OTA by ZAF(Ser) in actual samples, including biological fermentation broth (**e**) and red wine (**f**) in the left panels, the right panels showing photos of samples. Data were represented as mean ± SD ($n = 3$). **g–n** Chemical structure diagram for amide-containing chemicals and corresponding degradation efficiency, **g** OTA; **h** chlorpropham; **i** N-acetyl-L-phenylalanine; **j** N-Boc-L-phenylalanine; **k** L-alanyl-L-tyrosine; **l** N-acetyl-L-cysteine; **m** paracetamol; **n** N-acetyl m-toluidine.

conditions, including various pH values, temperatures, ionic strengths and organic solvents. When incubated at pH values ranging from 3 to 9 for 7 h, ZAF(Ser) retained over 70% of its initial activity. However, its native counterpart retained less than 20% of its initial activity when the pH was 3.0, 8.0 or 9.0 (Fig. 5a). The native hydrolase was incubated at temperatures varying from 20 to 80 °C and removed for enzymatic assay, which showed that the native CPA lost over 80% of its initial activity when the temperature was higher than 50 °C, while ZAF(Ser) preserved more than 90% of its initial activity under the same conditions (Fig. 5b). In addition, the tolerance of ZAF(Ser) toward common organic solvents was assessed, which included methanol (MeOH), ethanol (EtOH), acetonitrile (ACN), acetone (ACE), dichloromethane (DCM), N, N-dimethylformamide (DMF) and N, N-dimethylacetamide (DMAC), several of which are detrimental to native enzyme activity. The rigid framework endowed ZTF(Ser) with satisfactory stability under the tested conditions, and more than 80% of the initial activity was preserved (Fig. 5c). In contrast, native hydrolase was almost completely deactivated, and less than 5% of the initial activity was retained, with the exception of 20% activity retention by ACE (Fig. 5c).

Moreover, the effect of ionic strength on the enzymatic activity was examined (Supplementary Fig. 63). High ionic strength affects the charge distributions and spatial structures of enzymes, which tends to cause activity loss with the native enzyme[44]. Native hydrolase exhibited a drastic decrease in activity (more than 80%) when incubated in NaCl solutions with concentrations higher than 50 mM. In contrast, even when exposed to a 300 mM NaCl solution, ZAF(Ser) maintained more than 80% of its initial activity, implying the high stability of ZAF(Ser). Moreover, ZAF(Ser) exhibited good storage performance, with 90% of the original activity retained even when stored as a dry powder for 75 days (Supplementary Fig. 64). Encouraged by the good enzymatic activity and high stability, the recyclability of ZAF(Ser) was investigated to assess practical application potential (Supplementary Fig. 65). The catalytic activity of ZAF(Ser) exhibited no significant decrease after 4 cycles, and over 60% of the initial activity was maintained after 8 cycles. Further examination of ZAF(Ser) confirmed that the structure of ZAF remained intact with negligible leakage of zinc ions after recycling experiments, further validating its potential for future application (Supplementary Figs. 66–69).

Ochratoxin A (OTA) is a widespread mycotoxin found worldwide, which possesses carcinogenic, liver damage and kidney defect properties[45]. Biological detoxification of OTA via enzymatic hydrolysis of its amide bond to generate the remarkably less toxic OTα is one of the most promising methods for OTA degradation[46]. Therefore, we examined the catalytic performance of ZAF(Ser) in the degradation of OTA. ZAF(Ser) catalyzed the hydrolysis of OTA with the synergistic effect of the dual active sites, which led to the generation of L-phenylalanine and OTα (Supplementary Fig. 70). With identical mass concentrations for the catalysts, the rate for hydrolysis of OTA catalyzed by ZAF(Ser) was 1.8 times higher than that for native hydrolase (Fig. 5d), suggesting the excellent catalytic performance of ZAF(Ser). Subsequently, the effects of different reaction parameters, including pH, enzyme concentration and reaction time, on OTA degradation catalyzed by ZAF(Ser) were investigated (Supplementary Figs. 71–73). The degradation efficiency of OTA reached 97% after 30 min of reaction with a ZAF(Ser) concentration of $4\,mg\,mL^{-1}$ (Supplementary Fig. 58), indicating that ZAF(Ser) catalyzed rapid detoxification of OTA under mild reaction conditions. Compared with other OTA detoxification methods (Supplementary Table 12), such as adsorption, ozone treatment and electron beam irradiation, the catalytic degradation of the mycotoxin OTA by ZAF(Ser) exhibited great potential in terms of reaction time and degradation efficiency.

The above exciting results inspired us to further investigate the potential of ZAF(Ser) in real-world applications. Thus, the catalytic degradation of OTA in actual samples, including biological fermentation broth from our lab, which contains high salt concentration and commercialized red wine from a local supermarket which contains a higher ethanol concentration than the corresponding wastewater, were carried out. As can be seen, the degradation efficiency of OTA reached 92% and over 95% in biological fermentation broth with initial OTA of $2\,mg\,L^{-1}$ OTA (Fig. 5d, Supplementary Fig. 74) and in acidic red wine with initial concentration of OTA ranging from 3 to $12\,mg\,L^{-1}$ (Fig. 5f, Supplementary Fig. 75). On the contrary, the natural hydrolase (CPA) showed extremely low degradation efficiency of less than 15% in both samples. These experimental results demonstrate the advantages of ZAF(Ser) in practical applications involving samples with complex compositions and extreme environments.

Furthermore, the catalytic performance of ZAF(Ser) for detoxification of other amide-containing pollutants was examined. Chlorpropham, a widely used herbicide that is soluble in water and prone to accumulate in water sources, results in environmental risk. The efficiency of chlorpropham degradation catalyzed by ZAF(Ser) was 60.2% after 36 h (Fig. 5h and Supplementary Fig. 76). Additionally, ZAF(Ser) catalyzed hydrolysis of chlorpropham (Fig. 5h), N-acetyl-L-phenylalanine (Fig. 5i, Supplementary Fig. 77a), N-Boc-L-phenylalanine (Fig. 5j, Supplementary Fig. 77b), L-alanyl-L-tyrosine (Fig. 5k), N-acetyl-L-cysteine (Fig. 5l), paracetamol (Fig. 5m), N-acetyl m-toluidide (Fig. 5n), N-acetyl-L-tyrosine (Supplementary Fig. 78e), and N-acetyl-L-histidine (Supplementary Fig. 78f), which indicates the wide substrate range and potential for practical application (Supplementary Figs. 76–78).

## Discussion

Artificial enzymes are designed to overcome the challenges of low stability and limited operating range seen with native enzymes. Despite the fact that regulation of Lewis acidity via the primary coordination sphere led to enhanced hydrolytic activity of the artificial enzyme, hydrogen bonding in the second coordination sphere, which is common in nature enzyme, is involved in attack on the carbonyl group, which plays a key role in the hydrolysis process and has not received enough attention. Yet, the precise organization of functional groups to generate suitable reaction microenvironments mimicking the active sites of native enzymes is challenging. Additionally, the

coupling of the Lewis acidic active site and the hydrogen bonding active site in a single scaffold has not yet been realized.

In this study, the obtained ZAF(Ser) with dual catalytic active sites incorporated was realized via a fine-tuned organization of ligands and amino acids around metal ions. In addition to Lewis acidity regulation of the metal ion in the primary coordination environment, the formation of hydrogen bonds in the second coordination sphere enabled a more energy-favorable catalytic profile, which endowed the obtained MOFs with higher activities than their counterparts installed with a single Lewis acidic active site. The developed ZAF(Ser) displayed high stability, recyclability, and a wide substrate range and exhibited great potential in the field of environmental remediation, including biological fermentation wastewater and in healthcare-related applications. Given the abundance and structural versatility of amino acids and peptides, interactions among the biological moieties, organic ligands and metal ions are anticipated to provide a sophisticated and elaborate microenvironment resembling that of native enzyme. Thus, this study provides a rational method for the design of artificial enzymes and enzyme engineering, which hold promise for environmental and healthcare applications.

## Methods

### Synthesis of ZAF(X) and ZAF

$Zn(NO_3)_2 \cdot 6H_2O$ (1 mmol, 297.5 mg) and benzotriazole (1 mmol, 117.4 mg) in 30 mL of DMF were ultrasonically dissolved in a vial, and an aqueous solution of serine (100 mM, 10 mL) was added. The mixture was heated at 140 °C for 24 h. After cooling to room temperature, a pale white powder was harvested by centrifugation and washed three times with DMF and methanol. The collected precipitate was incubated in methanol overnight and then dried at 120 °C for 12 h under vacuum; it was denoted as ZAF(Ser). ZAF(X) was fabricated by following the exact procedure for ZAF(Ser) but replacing the serine with another amino acid (X). ZAF was synthesized according to the literature method with modification[27]. Typically, $Zn(NO_3)_2 \cdot 6H_2O$ (1 mmol, 297.5 mg) and benzotriazole (1 mmol, 117.35 mg) were ultrasonically dissolved in 20 mL of DMF/H$_2$O mixture solution (volume ratio of 3:1). The mixture was then transferred to a Teflon reactor and heated at 140 °C for 24 h. After cooling to room temperature, the solids were collected by filtration. The powders were washed three times with DMF and methanol, then dried at 60 °C under vacuum.

ZnSer was obtained via the self-assembly of zinc ions with serine. First, $Zn(NO_3)_2 \cdot 6H_2O$ (1 mmol, 297.4 mg) was dissolved in 15 mL of DMF and serine (1 mmol, 105.1 mg) was dissolved in 5 mL of deionized water. The mixture of $Zn(NO_3)_2$ solution and serine solution was exposed to 5 min of ultrasonication and then transferred to an autoclave and heated at 130 °C for 24 h. After cooling to room temperature, a pale-yellow precipitate was obtained by centrifugation. After washing with methanol, the solid was dried at 60 °C under vacuum.

### Enzymatic activity assays of ZAF(X) and native enzyme

The hydrolytic activities of ZAF(Ser) and carboxypeptidase A were determined by using hippuryl-L-phenylalanine (HPPA) as the substrate, and the hydrolysis product hippuric acid (HA) was monitored by HPLC. In brief, HPPA was dissolved in Tris-HCl (50 mM, pH 6.0) to give a final concentration of 1 mM. 6 mg ZAF(Ser) or natural enzyme of identical mass was added to 3 mL of the substrate solution to initiate the reaction. After 40 min of reaction at 40 °C, the reaction mixture was filtered, and the hydrolysis product HA in the filter liquor was removed for HPLC analysis. A blank control was set without the addition of a catalyst to rule out the self-hydrolysis of the substrate.

### Preparation of xSer@ZAF

Presynthesized ZAF powders (100 mg) were dispersed in 15 mL of DMF, into which different amounts of the aqueous solution of serine were added (0.1, 0.2, 0.3, 0.5, 1, 1.5 mmol). After sonication for 3 min,

the mixture solution was transferred to a 50 mL Teflon autoclave and reacted at 120 °C for 24 h. After cooling to room temperature, the resulting solids were collected via filtration and washed three times with DMF and methanol. The obtained sample was denoted xSer@-ZAF, where x refers to x mmol of Ser initially added to 1 g of ZAF during the synthetic process.

### EXAFS data collection and analysis

The X-ray absorption data at the Zn K-edge of the samples were recorded at room temperature in transmission mode using ion chambers at beamline 1W1B of the Beijing Synchrotron Radiation Facility (BSRF), China. The station was operated with a Si (211) double crystal monochromator. During the measurements, the synchrotron was operated at an energy of 3.5 GeV and with a current ranging between 150 and 210 mA. The photon energy was calibrated with the first inflection point of the Zn K-edge in Zn foil.

### Computational details

First-principles calculations were performed by using the SIESTA package[31] (SIESTA 4.0 version) to optimize the structures of serine, cysteine, alanine, and homoserine coordinated ZAF models, which were referred to as ZAF(Ser), ZAF(Cys), ZAF(Ala) and ZAF(Hser), respectively. DFT-D calculations were performed using a vdW-DF. Norm-conserving pseudopotentials of Troullier and Martins were used with valence electron configurations. A uniform real space grid of points equivalent to a plane wave cut-off of 250 Ry was used for the evaluation of the Hartree and exchange-correlation energies. The basis sets in SIESTA are numerical ones, consisting of the exact solutions of the pseudopotential for the atomic state, except that radial confinement is included to localize the orbital corresponding to an energy shift of 0.001 Ry.

For calculation of the catalytic mechanism, the cluster configurations of ZAF(Ser) and ZAF were extracted from the crystal structure files and then optimized under the framework of density of functional theory (DFT) with the PBE0 functional and def2SVP basis set[47]. To describe the restraints of these ligands in our real crystals, the nitrogen atoms coordinated with adjacent zinc ions were frozen in the calculations. The SMD (Solvation Model Based on Density) implicit solvent model was used to describe the solvation effect of water. The structures of the transition states and intermediates in the calculated reaction paths were optimized by using the same method. Vibrational frequency analysis was carried out for the optimized structure with the same functional and basis set. Thermodynamic correction terms of these structures at 298.15 K were obtained using the Shermo program. To obtain the electron energies with higher accuracy, which has a major impact on the accuracy of the Gibbs free energy, single point calculations for these optimized structures with the PBE0 functional and def2TZVP basis set[44] were performed. Finally, the single point energy was added to the free energy corrections calculated before to obtain the Gibbs free energies of all reactants, transition states, intermediates and products. The DFT calculations were performed using the Gaussian 16 B.01 program suite.

### Catalytic degradation of ochratoxin A by ZAF(Ser)

A methanol solution of ochratoxin A (OTA) (250 μg mL$^{-1}$, 40 μL) was added slowly into 1.96 mL of Tris·HCl solution (pH 6.0) with stirring, giving a final concentration of OTA 5 μg mL$^{-1}$. The desired amount of ZAF(Ser) a was added to 2 mL of the substrate solution and then reacted in the dark at 40 °C under stirring. After the reaction, an equivalent volume of methanol was added immediately to the substrate solution, followed by centrifugation and filtration to remove the catalyst and stop the reaction. The concentration of OTA left in the solution was determined with a fluorescence spectrometer. The degradation efficiency of OTA was calculated. The blank group without ZAF(Ser) was used as the control. The effect of ZAF(Ser) concentration on the OTA degradation efficiency was determined by varying the concentration of ZAF(Ser) (1, 2, 3, 4, 6 mg mL$^{-1}$) with an initial OTA concentration of 5 μg mL$^{-1}$. The effect of pH on the catalytic efficiency was examined by adjusting the pH of the reaction solution from 3.0 to 9.0 with the other experimental conditions unchanged. The degradation efficiency was expressed as the ratio between the difference in the OTA concentration before and after the reaction and the initial OTA concentration.

### Assessment of catalytic stability

The pH stability of ZAF(Ser) was examined by incubation at various pH values for 7 h and an enzymatic activity assay. The activity of ZAF(Ser) at pH 6.0 without incubation was set as a 100% reference. To investigate the thermal stability of the catalyst, ZAF(Ser) and natural enzyme were dispersed in Tris·HCl buffer and incubated at different temperatures (20–80 °C) for 1 h before the catalytic reaction. The catalytic activity of ZAF(Ser) at 40 °C without treatment was set as the 100% reference. The effect of ionic strength on the catalytic activity was investigated by adding different concentrations of NaCl (0–300 mM) to the reaction mixtures. The relative activity of ZAF(Ser) in the absence of NaCl was regarded as a 100% reference. The long-term stability of ZAF(Ser) was assessed by measuring the residual activity after storage at room temperature for periods of 0–35 days. The activity on the first day was taken as the 100% reference. To examine the tolerance of the catalyst toward organic solvents, powders of ZAF(Ser) or the natural enzyme were incubated for 7 h in various organic solvents, including methanol (MeOH), ethanol (EtOH), acetonitrile (ACN), acetone (ACE), dichloromethane (DCM), N,N-dimethylformamide (DMF) and N,N-dimethylacetamide (DMAC), removed for centrifugation and subjected to an enzymatic activity assay. The activity of ZAF(Ser) in Tris·HCl solution (pH 6.0) without treatment was taken as the 100% reference.

### Reporting summary

Further information on research design is available in the Nature Portfolio Reporting Summary linked to this article.

## Data availability

Data supporting the findings of this work are available within the paper and its Supplementary Information files. The datasets generated and analyzed during the current study are available from the corresponding author upon request. The source data for Figs. 2c–i, 3a–c, 3h–j, 4d, e, 5a–f and Supplementary Figs. 1a, b, 2–4, 6, 8, 11–15, 17–19, 21a–d, 24–29, 32, 37a, b, 38, 40–48, 51a–d, 52, 54, 61–65, 67–78 are provided as a Source Data file. Crystallographic data for the structures reported in this article have been deposited at the Cambridge Crystallographic Data Centre under deposition numbers CCDC 2292078 (ZAF). Copies of the data can be obtained free of charge via https://www.ccdc.cam.ac.uk/structures/. Source data are provided with this paper.

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

## Acknowledgements

This work was supported by the National Key Research and Development Program of China (2021YFC2102800), the National Natural Science Foundation of China (22278161) and the Guangdong Basic and Applied Basic Research Foundation (2022B1515020013).

## Author contributions

W.L., X.W. and J.G. supervised the project. W.L. and X.W. conceived the idea. X.Y. and X.W. performed the experiments and theoretical calculations with technical help from J.X. and S.L. B.Y. performed the EXAFS experiment. R.G. performed the MD simulations and helped with experiment result analysis. M.Z. put forward some constructive comments on the paper. All the authors discussed the results and commented on the manuscript.

## Competing interests

The authors declare no competing interests.
