## [Peer Review File · Nature Communications]

Reviewers' Comments:

Reviewer #1:

Remarks to the Author:

The authors report the benefit of using second shell coordination to enhance catalysis with Zn-MOFs, through enhanced interaction with substrate close to the active metal centers of the MOFs. The key novelty that deserves consideration at the level of Nature Comm is the idea that engineering second coordination sphere could be used to construct biomimetic active sites. In many ways, this has been done before by decorating the pores with H-bonding donating/acceptor moieties by ligand functionalization (ex: *J. Catal.* 2015, 331, 1 ; *Angew. Chem. Int. Ed.* 2018, 57, 1949). Even serine has been already used to enhance the hydrolase-like reactivity of MOFs (*Nat. Commun.* 2020, 11, 3080). Yet, I believe the concept of engineering the second coordination sphere to mimic that of a natural hydrolase is quite original for MOFs, and should inspire future catalyst design. As such, it could be published in Nature Comm, given the potential and need of robust artificial hydrolases.

To validate such concept, authors used experiments and theory. Notably, the theory is coherent with the proposed model, assigning a lower energy barrier for the Ser-containing materials, but showing also that Lewis acid catalysis and the putative H-bonding mediated pathway are too close in energy to clearly claim a dominant mechanism. Essentially, both mechanisms could be operating. Note that the so-called second coordination sphere has been also proposed to have diverse influence on the reactivity of metal centers, such as the electronics of the metals (ref 34 of manuscript), or the acidity inside MOF pores (*Catal. Sci. Technol.* 2020, 10, 4002). Therefore, the experiments are key to judge whether the second coordination sphere effect is truly operative, or if the serine is only changing the physicochemical features of the materials.

An overwhelming amount of catalytic experiments, and different materials has been provided. Generally, good indications that the proposed H-bonding effect could be operating are given. Particularly:

1. the lower rates of protected Ser-containing, and Ala-containing materials;
2. the slight decrease in rate for Thr-containing material compared to Ser-containing one.
3. the lower rate of Cys-containing material showing the relevance of the functional group involved in the presumed H-bonding interaction.
4. the rate increase positively correlating to the amount of serine when serine is incorporated post-synthetically (Fig 3j)

Even though, as mentioned above, the changes in structure could affect the MOF reactivity in different ways, I could give the authors the benefit of the doubt considering points 1-4, and in principle agree with their proposition, since studying molecular mechanistic features in heterogeneous catalysis is very challenging. However, the comparisons mentioned above ignored the fact that these MOFs are entirely different structures, and authors fail to show they are comparing comparable materials. For example:

5. Sup Fig. 21 clearly shows the MOFs prepared Cys, and Ala are clearly not crystalline materials. Surface area of ZAF(Ala) is also not given, giving no way to assess any eventual porosity of the material.

6. With respect to point 4, Sup. Table 8 shows the amount of serine incorporated by the post-synthetic method is virtually the same for all materials. Thus, a growing amount of serine cannot explain the results alone. Once again, surface area of the materials is not given, casting doubts on the assumption that H-bonding ability of serine is solely responsible to the increase in rate. It could be simply a greater access of the active sites, as the parent ZAF has a very low surface area.

7. Larger surface area has been long established to enhance reaction rates. However, authors only provide surface area for a few materials, leaving little room to analyze its effect in the reactivity. It could be that materials mentioned in points 1-4 simply have much lower surface areas than the ZAF(Ser) material, which would explain the differences in rate. One clear evidence of this potential effect is the 10-fold difference in surface area between ZAF and ZAF(Ser) - which curiously match the 11-fold higher hydrolytic activity claimed.

8. The X-ray diffraction analysis revealed that Serine could not be detected in ZAF(Ser) material, casting the doubt on the structure and the identity of the resulting material. They say that the amount of serine embedded in a single crystal of ZAF(Ser) was at such a low concentration as to

escape detection by SCXRD, but then the question is why they see such large effect in the reactivity of ZAF(Ser) compared to ZAF?

9. It is not clear on which basis the authors obtained the structures presented in in Supplementary Figure 22. How many Zn sites have Ser bound to them (keeping in mind the statement that " the amount of serine embedded in a single crystal of ZAF(Ser) was at such a low concentration as to escape detection by SCXRD" ?

10. There is not much detail related to the sample preparations and analysis methods used – how were the materials prepared to ensure that all O in the EDS analysis came from serine only and not H₂O adsorbed from the air?

11. The surface areas presented in sup. Table 4 are very small, and no pores are present in some MOF materials (the insert in sup figure 17 has no axis), as such, is a porous MOF structure really formed? Even with analogues ZIF materials, the surface area is expected to be around 1500 m²/g with well defined pores, and clear internal and external surface areas detected. If surface area is going to be used to explain differences in reactivity, it should be presented for every material. PXRD should be presented along with the calculated pattern of the MOF structure to ensure the expected MOF is formed.

12. The authors show that by heating the materials up, h-bonding is disrupted. When cooled down, is the activity restored? Or is the material structurally damaged at slightly elevated temperature when in the reaction conditions.

13. How does the presence of amino acids affect the connectivity of the MOF – is it possible that the use of serine for example, increases missing linkers, and the additional h-bonding and reactivity is contributed to by residual solvent molecules such as methoxy groups bound to the cluster rather than the amino acid – this would also give the O signal in EDX.

14. Comparing sup. Figures 3 and 4 the particle size of ZAF(ser) seems very different. Why is this?

15. It is unusual to give TGA conditions in terms of K/min-1, °C/min is more commonly used. Why did the authors choose such a high rate of heating for the TGA experiments? And in sup Figure 9, the weight % increases above 100% under 100 °C – how did this happen? A slower temperature increase may have prevented that from occurring.

16. The authors should make a bigger effort regarding the XAFS analysis. There are no details given and an actual discussion about the XAFS fits/spectra is missing. For example, in Figure 2g the authors affirm that ZAF and ZAF(Ser) have similar XANES spectra to ZnO. However, while ZAF appears similar to ZnO, ZAF(Ser) has some additional features. The white line is indeed quite broad.

17. Figure 2h: The authors should address why ZAF(Ser) has no second-shell contribution, differently from ZAF. Moreover, why do these FTs look different from Figure S12? In the latter, the first peak for ZAF and ZAF(Ser) are at a similar radial distance; in contrast, the peaks' position in the main figure is very different.

18. Errors should be expressed for all structural results (bond lengths, path degeneration, etc). The Experimental should clearly state the adopted EXAFS model, analysis protocol, and software. I suggest including a table containing all structural results derived from the fit, including the respective errors, to draw a clear comparison between the samples.

19. There is very little detail given on the DFT calculations given. It should be clearly explained how these calculations were performed. In particular, Figures S44 and S45 depicting the proposed mechanism are very unclear. They should be presented using chemical structures (similar to what is presented in figures S42 and S43) and not ball and stick model.

In this context, I would recommend the paper to be rejected or, at least, to undergo major revisions, since key evidence is not supporting the conclusions. In revising the paper, the authors should also:

20. Improve the scholarly presentation of the paper. An overwhelming amount of data not essential to demonstrate the point is included, making the paper very difficult to read. Authors should select the materials that could be clearly demonstrated they are MOFs, and that actually contribute to prove their point. For example, statements such as "The generality of this design was confirmed by using different kinds ofazole derivatives,..." (from line 358) are clearly not fully supported by PXRD patterns on Sup Fig. 39, and as such just makes the manuscript more convoluted.

21. Provide surface area of all materials, and a clear comparison of reaction rates considering this property should be given.

22. Avoid comparisons with ZAF parent material. These are not relevant, given the large difference in surface area. Moreover, ZAF and ZAF(Ser) are materials with a very different nature, and

claiming benefits to the design by comparing them are largely misleading.

23. Clarify the amounts of MOF material used in every catalytic reaction. Currently, experimental protocols are vague, and figures or text do not contain the information.

24. Provide clearer details on how the catalytic results are presented, and discuss the limitations of their analysis. Several results are given based on the ratio between Ser and Zn (Fig 3b), while others are expressed in terms of an intriguing rate increment (3c), although the actual rate change is only 2- to 3-fold (Sup Table 5). In general, I appreciate the deep analysis of catalytic activity with respect to changes in structure. However, authors should give more details how these calculations have been done (no protocol is given). Without clearer explanations/details, it is very difficult to understand for example how/why the numbers in Fig 3b and 3c are different, and the moles of what the normalization is being based on.

25. Consider removing comparisons with natural enzyme from the main text. Comparison with natural enzyme, though interesting, is not exactly surprising or highly relevant. It has been demonstrated before for other MOF materials designed to environmental remediation. If authors wish to keep it in the main text, equivalence of catalyst concentration should be not done by using equal mass amounts of both. Instead, the number of catalytic active sites in order to claim higher MOF activity should be used, as an enzyme has only one catalytic site, while the same mass of MOF could bear multiple ones.

26. Improve the discussion to reflect the advances of the paper. Currently, it stands as a paragraph more suited to the introduction, and offers few critical insights with respect to the paper's findings and concept.

27. Improve graphical displays, as they are generally too small, and difficult to read. A two-column format should be used, at least.

Reviewer #2:

Remarks to the Author:

This report by Wen-Yong Lou and co-workers reports some interesting catalytic activity from a mixed ligand preparation of Zn using benzotriazole and various amino acids, especially serine. Generally a lot of work has been done here, and the subject of the work is likely to be of interest to the community. However, in my view the authors have based a lot of their analysis on the basis of a somewhat flimsy hypothesis as to what the active material actually is here. From the evidence produced, I suspect the actual composition of "ZAF(Ser)" is more likely just a mixture of particles of a (known) non-porous coordination polymer Zn(btz) and a more active zinc serine complex.

The authors have fundamentally misinterpreted their PXRD and single crystal data - the PXRD in figure 2(c) does not "indicat(e) coexistence of the coordination structures of ZAF and ZnSer in the framework of ZAF(Ser)", it indicates coexistence of separate particles of ZnSer and ZAF(Ser)! PXRD does not work like that - if the active species had the structure that the authors propose (with partial replacement of a btz ligand with a Ser ligand within the same ZAF structure), the PXRD pattern of the bulk would either resemble that of pure ZAF or would take on an entirely new pattern based on another crystal structure with different symmetry.

The authors' own SC-XRD data show that the ZAF structure is essentially unchanged. Taking this hypothesis forwards it also explains the gas adsorption results - presumably the ZnSer and ZnCys structures have some porosity, while the ZAF doesn't (this is clear from the crystal structure - it is a close packed, non porous substance, and the "mesoporosity" of <30 m²/g is essentially negligible in MOF terms). Much of the remaining data then carries this hypothesis forwards but does nothing to disprove the alternative explanation - IR, elemental analysis etc are all done on bulk samples and would look the same with a hypothetical "ZAF(Ser) as they would with a mixture of ZnSer and ZAF particles in various ratios.

The catalytic activity seems to be higher in the mixture than with the individual components - this is an interesting element, but it is again not fully explained by the hypothesis put forward by the authors. The modelling of the active site looks nice, but neglects the extended structure of the material. We have already seen that ZAF is non-porous (this phase has been known since 1981), so presumably this is all happening on the particle surfaces? If so, the real active site could take any number of configurations.

Regardless of the potentially interesting catalytic activity of this mixture, the chemistry is not sufficiently characterised here and the authors have not sufficiently proven the identity of the active species, and so in my view this work is not yet ready for publication.

Reviewer #3:

Remarks to the Author:

The authors present the rational design of a new MOF with post-synthesis modifications at the second coordination sphere that enhance the catalytic degradation of C-N bond via hydrolysis. The authors are inspired by enzymatic analogues where specific functional groups (-OH) of amino acids increase the hydrolytic activity. After the presentation of the synthesis and a detailed characterization of the material, the authors are presenting mechanistic insights from computations. For the computed reaction mechanism, the authors have nicely frozen key atoms of the ZAF material to introduce the "stiffness" of the MOF in the geometry optimizations. Finally, they provide results on the degradation of ochratoxin A via hydrolysis promoted by ZAF(Ser). Overall, I believe that the experimental part of the article is excellent, but I do have concerns with respect to the computational work. For that reason, the authors should consider the following revisions and provide additional clarifications on the computational aspects of their work:

In Figure 4, we see in (a) a $\text{Zn}(\text{triazole})_3(\text{H}_2\text{O})$, (b) a $\text{Zn}(\text{triazole})_2(\text{serine})(\text{H}_2\text{O})$, and in (c) a $\text{Zn}(\text{triazole})_3(\text{serine})$. I do not understand how Zn lost a triazole linker in b. Is this a demonstration of a defect? It is also unclear how the reaction profiles of Figures 4d and e are connected to the reaction cycles of Figures 4f and g, since there are no common labels between profiles and cycles. Please add the labels of the reaction intermediates in the cycle. It is also not clear why the authors decide to add a H_3O^+ on the second mechanism and a H_2O on the first. After all, the catalytic reaction with ZAF(Ser) is independent of the pH (Supplementary Figure 46.).

Most importantly, both catalytic cycles are endothermic, and both involve reaction steps with very large energy barriers. In the first case, the reaction barriers of 39.2 and 42.0 kcal/mol are prohibitively large and in the second case, the reaction is be trapped on INT-2' since this is the more thermodynamically stable intermediate (more stable than reactants/products). In addition, experimentally, the authors report (low) product formation for ZAF, but the proposed mechanism with the 42.0 kcal/mol barrier is again prohibitively large.

It would be nice if there is a direct comparison of the hydrogen bonding-mediated catalytic pathway with the biological equivalent reaction mechanisms of Supplementary Figure 42.

Finally, the statement "reduction of the energy barrier thus accelerates the reaction which is consistent with the EXAFS result" needs additional justification. This will help the authors to connect computational findings with experimental results.

I believe that the title is too general and does not fully reflect the scope of the work presented in this manuscript.

Please change the section title "Creation and structural..." to "Synthesis ..." or "Formation...".

Responses to editor's and reviewers' comments and changes we have made

Thank you very much for all the comments and suggestions from three reviewers, which greatly improved our manuscript. We provided a point-to-point response to all comments and listed all new experiments and revisions in blue font.

REVIEWER COMMENTS

Reviewer #1 (Remarks to the Author):

Comment 1-5 *The authors report the benefit of using second shell coordination to enhance catalysis with Zn-MOFs, through enhanced interaction with substrate close to the active metal centers of the MOFs. The key novelty that deserves consideration at the level of Nature Comm is the idea that engineering second coordination sphere could be used to construct biomimetic active sites. In many ways, this has been done before by decorating the pores with H-bonding donating/acceptor moieties by ligand functionalization (ex: J. Catal. 2015, 331, 1; Angew. Chem. Int. Ed. 2018, 57, 1949). Even serine has been already used to enhance the hydrolase-like reactivity of MOFs (Nat. Commun. 2020, 11, 3080). Yet, I believe the concept of engineering the second coordination sphere to mimic that of a natural hydrolase is quite original for MOFs, and should inspire future catalyst design. As such, it could be published in Nature Comm, given the potential and need of robust artificial hydrolases.*

To validate such concept, authors used experiments and theory. Notably, the theory is coherent with the proposed model, assigning a lower energy barrier for the Ser-containing materials, but showing also that Lewis acid catalysis and the putative H-bonding mediated pathway are too close in energy to clearly claim a dominant mechanism. Essentially, both mechanisms could be operating. Note that the so-called second coordination sphere has been also proposed to have diverse influence on the reactivity of metal centers, such as the electronics of the metals (ref 34 of manuscript), or the acidity inside MOF pores (Catal. Sci. Technol. 2020, 10, 4002). Therefore, the experiments are key to judge whether the second coordination sphere effect is truly operative, or if the serine is only changing the physicochemical features of the materials.

An overwhelming amount of catalytic experiments, and different materials has been provided. Generally, good indications that the proposed H-bonding effect could be operating are given. Particularly:

- 1. the lower rates of protected Ser-containing, and Ala-containing materials;*
- 2. the slight decrease in rate for Thr-containing material compared to Ser-containing one.*
- 3. the lower rate of Cys-containing material showing the relevance of the functional group involved in the presumed H-bonding interaction.*
- 4. the rate increase positively correlating to the amount of serine when serine is incorporated post-synthetically (Fig 3j)*

Even though, as mentioned above, the changes in structure could affect the MOF reactivity in different ways, I could give the authors the benefit of the doubt considering

points 1-4, and in principle agree with their proposition, since studying molecular mechanistic features in heterogeneous catalysis is very challenging. However, the comparisons mentioned above ignored the fact that these MOFs are entirely different structures, and authors fail to show they are comparing comparable materials. For example:

5. *Sup Fig. 21 clearly shows the MOFs prepared Cys, and Ala are clearly not crystalline materials. Surface area of ZAF(Ala) is also not given, giving no way to assess any eventual porosity of the material.*

Response: We deeply thank you for your appreciation on our study and also the positive and professional comments. And as you mentioned, “studying molecular mechanistic features in heterogeneous catalysis is very challenging”. Thus, we have tried our utmost to conduct a large amount of experiments and also use theory calculations to support our assumption and conclusion. **Following your helpful suggestion, we have carried out new experiments to include the surface area of all the prepared materials mentioned in this manuscript.** We supplemented the surface area and pore size distribution of ZAF(Cys), ZAF(Ala), ZAF(His), ZAF(Asp), and ZAF(Glu) (Supplementary Fig. 21 and Supplementary Table 4). In addition, the surface area and pore size distribution of ZAF(HSer), ZAF(Thr), ZAF(O-Ac-Ser) and ZAF(O-Btu-Ser) was supplemented (Supplementary Fig. 51, Supplementary Table 4). These data enable a better comparison of the catalytic activities among different ZAF based catalysts. Also, as high scanning speeds led to poor XRD signals, we re-carried out the XRD analysis of ZAF(Cys) and ZAF(Ala) (revised Supplementary Fig. 24, Page S25). Despite the poor crystallinity of ZAF(Cys) and ZAF(Ala), both maintained the main characteristic peaks of ZAF (9.0°, 18.1° and 28.3°).

Nitrogen sorption isotherms analysis showed that BET surface area of ZAF(Cys) (Entry 2, Supplementary Table 4) was 2.7 times higher than that of ZAF(Ala) (Entry 3, Supplementary Table 4), but showed almost identical activity. Though the surface area of ZAF(Ser) was higher (Entry 1, Supplementary Table 4) than that of ZAF(Ala) and lower than that of ZAF(Cys), ZAF(Ser) featured 3 times higher catalytic activity. This result indicated that specific surface area can not explain why ZAF(Ser) displayed higher catalytic activity. We have included more discussion about the effect of surface area on the catalytic activity in the revised manuscript according to your suggestion.

Revision:

Manuscript, Page 12-13, Line 229-258:

“Furthermore, ZAF(Cys) with cysteine substituted for serine was fabricated by following the same procedure (Fig 3b, Supplementary Fig. 19, Fig. 2d), which featured a 16 times larger surface area than that of ZAF but exhibited no evident difference in activity compared with ZAF under identical amount of zinc. Thus, increased surface area did not lead to activity enhancement, which was further confirmed by the case of alanine (Ala), where ZAF(Ala) bearing 5 times higher surface area exhibited almost identical catalytical activity to that of ZAF (Supplementary Fig. 21, Supplementary

Table 4). Thus, the enhanced catalytic activity of ZAF(Ser) compared with ZAF can not be simply ascribed to the increased surface area. Furthermore, MOFs with missing linker or with electron-withdrawing group on the linker has been reported to increase the Lewis acidity of metal sites³¹. Thus, ZAF(x) with amino acids bearing various side chain groups other than OH was prepared (Supplementary Figs. 22-24). ZAF(Asp) and ZAF(Glu) with carboxylate groups showed activities similar to that of ZAF at identical Zn content, although the surface area was ~3 times higher than that of ZAF (Supplementary Table 4 and Supplementary Fig. 21). Moreover, by normalizing the reaction rate with specific surface area, the catalytic activity of ZAF(Asp), ZAF(Glu) is almost identical to that of ZAF(Ser) (Supplementary Table 5), implying that change of amino acid with different electron-withdrawing capability and the resultant enhanced Lewis acidity did not lead to activity change of the obtained composite. This was further confirmed by the almost unchanged activity of ZAF(Cys) with varying ratio of cysteine to BTA during synthetic process to regulated the Lewis acidity of Zn in the final composite (Supplementary Fig. 25). Thus, the possibility of enhanced Lewis acidity after the hybrid coordination of amino acid that led to activity increase is ruled out. These results together indicated that the remarkably higher activity of ZAF(Ser) cannot simply be ascribed to the defects and increased surface area due to incorporation of the small amino acids or the regulated Lewis acidity of the zinc ions in the primary coordination sphere but was generated by the structure of ZAF(Ser), which points to the presence of hydrogen bonds involving OH groups in serine and dangling N atoms from BTA, as inspired by the native enzyme.”

Supplementary Information, Page S22:

Supplementary Fig. 21. Nitrogen sorption isotherms and pore size distribution of (a) ZAF(Cys), (b) ZAF(Ala), (c) ZAF(His), (d) ZAF(Asp) and (e) ZAF(Glu).

Supplementary Information, Page S25:

Supplementary Figure 24. PXRD patterns of ZAF(x) with different amino acids as the substitute of serine.

Supplementary Information, Page S26:

Supplementary Fig. 25. Catalytic activity of ZAF(Cys) obtained with varying the molar ratios of BTA to Cys during the synthetic process.

Supplementary Information, Page S52:

Supplementary Fig. 51. Nitrogen sorption isotherms and pore size distribution of (a) ZAF(HSer), (b) ZAF(Thr), (c) ZAF(O-Ac-Ser) and (d) ZAF(O-tBu-Ser).

Supplementary Information, Page S83:

Supplementary Table 4. BET surface area, pore volume of sample, zinc content and corresponding reaction rate of different ZAF composites.

Entry	Sample	Zn content (wt %)	BET surface area (m ² g ⁻¹)	Total pore volume (cm ³ g ⁻¹) calculated by DFT method	Reaction rate under identical amount of Zn (μM min ⁻¹)
0	ZAF	32.1	21.9	0.053	2.1 ± 0.04
1	ZAF(Ser)	18.6	174.8	0.256	6.6 ± 0.21
2	ZAF(Cys)	17.9	272.3	0.335	2.4 ± 0.44
3	ZAF(Ala)	19.8	99.94	0.161	1.8 ± 0.09
4	ZAF(His)	29.6	97.73	0.363	3.9 ± 0.23
5	ZAF(Glu)	20.1	73.00	0.0689	2.5 ± 0.34
6	ZAF(Asp)	29.1	62.51	0.2796	2.4 ± 0.03
7	ZAF(HSer)	31.5	97.40	0.3433	6.8 ± 0.66
8	ZAF(Thr)	29.3	23.69	0.0364	4.6 ± 0.08
9	ZAF(O-Ac-Ser)	33.3	73.51	0.3189	2.1 ± 0.03
10	ZAF(O-tBu-Ser)	29.9	67.80	0.3189	2.5 ± 0.23

Supplementary Information, Page S83:

Supplementary Table 5. BET surface area and Surface-area-normalized reaction rate of ZAF composites under identical amount of zinc.

Entry	Sample	BET surface area (m ² g ⁻¹)	Surface-area-normalized reaction rate under identical amount of Zn (g nM min ⁻¹ m ⁻²)
1	ZAF	21.9	91.3
2	ZAF(Ser)	174.8	37.7
3	ZAF(Cys)	272.3	8.8
4	ZAF(Ala)	99.94	18.0
5	ZAF(His)	97.73	39.9
6	ZAF(Glu)	73.00	34.2
7	ZAF(Asp)	62.51	38.4

Comment 6 6. *With respect to point 4, Sup. Table 8 shows the amount of serine incorporated by the post-synthetic method is virtually the same for all materials. Thus, a growing amount of serine cannot explain the results alone. Once again, surface area of the materials is not given, casting doubts on the assumption that H-bonding ability of serine is solely responsible to the increase in rate. It could be simply a greater access of the active sites, as the parent ZAF has a very low surface area.*

Response: Thank you for your valuable suggestion. **We have carried out new experiments to measure the nitrogen sorption isotherms of all the obtained $x\text{Ser@ZAF}$ listed in original Sup. Table 8** (Supplementary Fig. 47, revised Supplementary Table 9), which showed decreased surface area compared with parent ZAF when increased amount of the serine was incorporated, demonstrating that surface area did not account for reaction rate enhancement. Therefore, the positive correlation between the incorporated serine and activity under identical amount of zinc suggested the important role of serine for catalytic activity. And further experiments in our studies pointed to the conclusion that hydrogen bonding inside accounted for the activity enhancement of ZAF(Ser).

Despite the amount of serine incorporated per unit of ZAF (amount of serine incorporated in ZAF (mmol g^{-1})) by the post-synthetic method is almost the same for $x\text{Ser@ZAF}$ (as the reviewer mentioned), **the ratio of serine to zinc in $x\text{Ser@ZAF}$ varied.** Similar to other post-modification processes of MOFs, the competitive coordination between BTA and serine led to partial detachment of BTA from the parent ZAF. Therefore, more serine added during the post-synthetic process led to increased ratio of serine to zinc, both of which are involved with the number of active site and thus determined the final catalytic activity.

To avoid misunderstanding, **we have added the calculation result of the ratio of serine to zinc in $x\text{Ser@ZAF}$ in revised Supplementary Table 10.** Also, we revised Fig. 3j and Supplementary Fig. 48, which showed activity comparison of $x\text{Ser@ZAF}$ in terms of ratio to zinc under identical amount of zinc, highlighting the role of serine in reaction rate enhancement.

Revision:

Manuscript, Page 19, Line 381-383:

“Notably, $x\text{Ser@ZAF}$ featured lower surface area than that of ZAF (Supplementary Fig. 47 and Supplementary Table 9), but increasing the serine content in the composite led to proportional activity enhancements from Ser@ZAF (Fig. 3j, Supplementary Table 9-10).”

Supplementary Figure 47. Nitrogen sorption isotherms and pore size distribution for xSer@ZAF obtained via post-synthetic modification.

Supplementary Information, Page S88:

Supplementary Table 9. The specific surface area and pore volume of x Ser@ZAF with amounts of serine doping in ZAF.

Sample	Amount of added serine (mmol g ⁻¹ ZAF)	BET surface area (m ² g ⁻¹)	Total pore volume (cm ³ g ⁻¹) calculated by DFT method	Reaction rate under identical amount of Zn (μM min ⁻¹)
ZAF	0	21.9	0.053	2.0 ± 0.06
1 Ser@ZAF	1	11.4	0.047	2.1 ± 0.02
2 Ser@ZAF	2	10.1	0.025	2.7 ± 0.08
3 Ser@ZAF	3	7.4	0.021	3.7 ± 0.10
5 Ser@ZAF	5	5.4	0.024	5.9 ± 0.12
10 Ser@ZAF	10	6.9	0.018	7.6 ± 0.40
15 Ser@ZAF	15	3.4	0.018	8.2 ± 0.46

Supplementary Information, Page S89:

Supplementary Table 10. Elemental analysis of x Ser@ZAF obtained via post-synthetic modification.

Sample	Amount of added Ser (mmol g ⁻¹ ZAF)	Zn (%)	N (%)	C (%)	H (%)	Amount of Ser incorporated in ZAF (mol g ⁻¹)	Molar ratio of Ser to Zn
ZAF	0	32.1	24.8	39.9	2.3	-	-
1 Ser@ZAF	1	31.9	20.6	35.7	2.4	1.1×10 ⁻³	0.22
2 Ser@ZAF	2	31.5	20.6	36.4	2.5	1.2×10 ⁻³	0.25
3 Ser@ZAF	3	29.0	21.7	38.4	2.6	1.3×10 ⁻³	0.29
5 Ser@ZAF	5	24.4	22.7	40.7	2.9	1.4×10 ⁻³	0.37
10 Ser@ZAF	10	21.5	23.9	43.6	3.2	1.4×10 ⁻³	0.42
15 Ser@ZAF	15	19.8	25.6	45.9	3.1	1.5×10 ⁻³	0.49

Comment 7 *7. Larger surface area has been long established to enhance reaction rates. However, authors only provide surface area for a few materials, leaving little room to analyze its effect in the reactivity. It could be that materials mentioned in points 1-4 simply have much lower surface areas than the ZAF(Ser) material, which would explain the differences in rate. One clear evidence of this potential effect is the 10-fold difference in surface area between ZAF and ZAF(Ser) - which curiously match the 11-fold higher hydrolytic activity claimed.*

Response: Thank you for the helpful suggestion which helped improve the quality of our manuscript. **Following your suggestion, we have carried out new experiments to examine the surface area of the prepared materials mentioned in this study,** which provide the room for more discussion and also more useful experimental evidence for prove our assumption. A brief response is provided below.

We agree with the reviewer that larger surface area meant more accessible active sites during catalysis, which might lead to enhanced reaction rates, a strategy of which has also been widely adopted in many previous researches. However, in this study, surface area is not responsible for activity change, as evidenced by the surface area results. The surface area of the materials mentioned in points 1-4 has been supplemented in Supplementary Fig. 21, Supplementary Fig. 47, Supplementary Fig. 51, Supplementary Table 4 and Supplementary Table 9. Despite the fact that ZAF(Ala) showed 4.6-fold larger surface area than that of ZAF (Supplementary Table 4 and Supplementary Table 9), the activity of ZAF(Ala) is almost identical to that of ZAF under identical amount of zinc (**Point 1**), suggesting that larger surface area did not account for activity enhancement. This is further confirmed by the case with -OH group shielded (**Point 1**) or with -OH changed to -SH (**Point 3**). ZAF(O-Btu-Ser), ZAF(O-Ac-Ser) and ZAF(Cys) which also showed a 3.1-fold, 3.4-fold and even 12.5-fold higher surface area compared with ZAF, but featured almost identical activity with ZAF under identical zinc amount. On the contrary, ZAF(Thr) with Thr bearing -OH group showed almost identical surface area with ZAF, but exhibited 2.3-fold higher catalytic activity (**Point 2**), which suggested the key role of -OH group for activity enhancement. Furthermore, xSer@ZAF featured lower surface area than that of ZAF (Supplementary Table 9), but showed 1~4 fold higher catalytic activity than that of ZAF under identical amount of zinc, with activity increase proportional to the amount of incorporated serine (**Point 4**). **Thus, the materials mentioned in Point 1-4 demonstrated that introduction of amino acid bearing the specific hydroxyl group played a vital role in determining the catalytic activity of the final composite, but not the surface area.** It's purely coincidental that the activity difference of ZAF(Ser) and ZAF is 11-fold, similar to the surface area difference. Besides, other experiment results also supported our assumption. When using homoersine (HSer) containing -OH group instead of serine for catalyst synthesis, ZAF(HSer) exhibited 4.4-fold higher specific surface area but 13.8-fold higher catalytic reaction rate than that of ZAF (Supplementary Table 4). **We kindly ask the reviewer to refer to our response to Comment 1-5 and Comment 6.**

Furthermore, we carried out new experiments by preparing a series of cysteine (Cys) incorporated ZAF (x Cys@ZAF) via the post-synthetic method,

similar to that of x Ser@ZAF, which are followed characterizations and enzymatic assays (Supplementary Table 11, Supplementary Fig. 54). The structure difference between cysteine and serine lies in that cysteine bears -SH group and serine bears -OH group. Experiment results showed that even under high content of cysteine, x Cys@ZAF displayed almost identical catalytic activity to that of ZAF (Supplementary Fig. 54), again verified the crucial role of -OH group and resultant formation of hydrogen bonding in determining the catalytic activities.

Revision:

Manuscript, Page 20, Line 398-406:

“In addition, the role of hydrogen bonding was further confirmed by using serine with OH shielded by acetyl, and t-butyl was adopted for the synthesis of ZAF(O-Ac-ser) and ZAF(O-tBu-ser) (Supplementary Figs. 49-50). ZAF(O-tBu-Ser) and ZAF(O-Ac-Ser) also showed a 3.1-fold and 3.4-fold higher surface area compared with ZAF (Supplementary Fig. 51 and Supplementary Table 4), but featured almost identical activity with ZAF under identical zinc amount (Supplementary Fig. 52). On the contrary, ZAF(Thr) with Thr bearing -OH groups (Supplementary Fig. 53) showed almost identical surface area with ZAF, but exhibited a 232% catalytic activity relative to ZAF.”

Manuscript, Page 19, Line 381-383:

“Notably, x Ser@ZAF featured lower surface area than that of ZAF (Supplementary Fig. 47 and Supplementary Table 9), but increasing the serine content in the composite led to proportional activity enhancements from Ser@ZAF (Fig. 3j, Supplementary Table 9-10).”

Supplementary Information, Page S55:

Supplementary Fig. 54. Catalytic activity per mole of cysteine under identical amount of Zn in x Cys@ZAF.

Supplementary Information, Page S90:

Supplementary Table 11. Elemental analysis of x Cys@ZAF with amounts of serine doping in ZAF.

Sample	Amount of added Cys (mmol g ⁻¹ ZAF)	Zn (wt %)	N (wt %)	C (wt %)	S (wt %)	Molar ratio of Cys to Zn
1 Cys@ZAF	1	32.8	19.9	34.2	3.0	0.19
2 Cys@ZAF	2	32.4	17.1	31.8	5.5	0.34
3 Cys@ZAF	3	32.0	15.8	29.0	7.2	0.45
4 Cys@ZAF	4	31.8	13.5	26.6	9.9	0.64
6 Cys@ZAF	6	31.6	12.6	24.0	10.2	0.70

Comment 8 8. *The X-ray diffraction analysis revealed that Serine could not be detected in ZAF(Ser) material, casting the doubt on the structure and the identity of the resulting material. They say that the amount of serine embedded in a single crystal of ZAF(Ser) was at such a low concentration as to escape detection by SCXRD, but then the question is why they see such large effect in the reactivity of ZAF(Ser) compared to ZAF?*

Response: Thanks for your comments. Yes, it is the low content of serine incorporated and its large effect on catalytic activity in ZAF(Ser) that interested us and inspired us to investigate in-depth on the catalytic mechanism. The presence of coordinated serine and BTA in single crystal of ZAF(Ser) was first confirmed by using ¹H NMR, though with extremely low characteristic peak intensity of serine (Supplementary Fig. 31). The quantitative measurement of serine in ZAF(Ser) single crystal was calculated to be ~2.7 wt% based on the result of ICP-OES and elemental analysis (Supplementary Table 8). Such a low content of serine meant that only partial of the zinc randomly coordinated with serine inside the single crystal particle and generated much lower diffraction intensity compared with other facets intrinsic to ZAF. Therefore, in the single crystal, the poor signal from serine was considered as background noise during the data resolution process. Despite the low content of serine in the ZAF(Ser) single crystal, presence of tiny amount of hydrogen bonding due to the coordinated serine led to an enzyme-like catalytic active site, which showed high catalytic activity during the hydrolysis process. **If we attributed the difference of catalytic activity between ZAF and ZAF(Ser) to the number of the present hydrogen bonding, in other words, the amount of coordinated serine. And we considered that the activity of ZAF originated from the number of the Lewis acidic site (amount of zinc) and calculated the contribution of each kind of active site to the activity.** Calculation results showed that each mole of Lewis acidic active site exhibited activity of 1.2×10^5 $\mu\text{M min}^{-1} \text{mol}^{-1}$, while each mole of hydrogen bonding-involved active site exhibited

activity of $1.3 \times 10^6 \mu\text{M min}^{-1} \text{mol}^{-1}$, the latter of which is 11-fold higher than the former (Fig. 3c). The high catalytic activity of hydrogen bonding-involved active site can be attributed to the elaborately organized space inside the catalyst which facilitated the binding and activation of the substrates, mimicking the catalytic process of enzyme. The hydrogen bond formed between the hydroxyl group of the serine with the vacant nitrogen of the benzotriazole. The hydrogen bond deprotonated the hydroxyl group of serine to generate an oxygen anion, which efficiently hydrolyzes the C-N bond, as illustrated in Fig. 4f and Supplementary Fig. 56, Supplementary Fig. 59.

Revision:

Manuscript, Page 15, line 295-301:

“Despite our attempt to obtain the single crystal of ZAF(Ser), the quantitative measurement of serine in ZAF(Ser) single crystal was calculated to be ~2.7 wt% based on the result of ICP-OES and elemental analysis (Supplementary Table 8). Such a low content of serine implied that only partial zinc coordinated with serine inside the single crystal particle randomly and generated much lower diffraction intensity compared with other facets intrinsic to ZAF.”

Supplementary Information, Page S87:

Supplementary Table 8 Elemental analysis and ICP-OES of ZAF(Ser) powder and single crystal.

ZAF(Ser)	Zn (wt %)	N (wt %)	C (wt %)	Amount of Ser (wt %)	Molar ratio of Ser to Zn
Powder	18.6	28.1	47.3	6.1	0.2
Single crystal	22.2	26.6	46.5	2.7	0.091

Manuscript, Page 21, revised Fig. 3c:

Fig. 3c Catalytic activity contribution of Zn in ZAF and amino acid incorporated including Ser in ZAF(Ser), HSer in ZAF(HSer), Ala in ZAF(Ala) and Cys in ZAF(Cys) under identical amount of zinc.

Comment 9 9. It is not clear on which basis the authors obtained the structures presented in Supplementary Figure 22. How many Zn sites have Ser bound to them (keeping in mind the statement that “the amount of serine embedded in a single crystal of ZAF(Ser) was at such a low concentration as to escape detection by SCXRD”)?

Response: Thank you for your comment and we apologize for the confusion caused by Supplementary Figure 22 (Supplementary Fig. 30 in revised Supplementary Information). Original Supplemental Figure 22a is a ball and stick model structure of ZAF(Ser) and Supplemental Figure 22b shows the coordination environment of divalent zinc atoms, as derived from single crystal analysis. As the content of serine in the single crystal was low, we cannot directly obtain the specific coordination environment of the incorporated serine and the hydrogen bonding-involved active site. Thus, **the structure presented in original Supplementary Figure 22c was estimated based on the characterization including FTIR, XRD, ICP-OES, elemental analysis, XAFS, etc.** As the ratio of serine to zinc in ZAF(Ser) single crystal was 0.091 (Supplementary Table 8), which meant that less than 10% of zinc inside randomly coordinated with serine and BTA simultaneously. And the randomly distribution of the serine inside ZAF(Ser) led to almost undetectable XRD signals. According to your helpful suggestion and also to avoid confusion, we have revised the figure caption and deleted original Supplementary Figure 22c to make it more precise.

Revision:

Supplementary Information, Page S31:

Supplementary Fig. 30. (a) Ball and stick model structure of ZAF(Ser) in single crystal analysis. (b) Coordination environment of divalent zinc atoms.

Comment 10 10. There is not much detail related to the sample preparations and analysis methods used – how were the materials prepared to ensure that all O in the EDS analysis came from serine only and not H₂O adsorbed from the air?

Response: Thank you for your suggestion. We have made our best to avoid the long

time contact of the sample with the air before the TEM observation and EDS mapping. **The experiment details are added in the revised Supplementary Information according to your suggestion.** Yet, it is still difficult to ensure that all O element in EDS mapping is from serine only, as contact of materials with H₂O from the air can be not completely avoided. However, the ratio of O from H₂O in the air in the final EDS mapping image is extremely low. **Following your suggestion, we have also carried out new experiment by measuring the EDS mapping image of ZAF, which did not contain O inside.** By prolonged the scanning time, stronger signals of element distribution in ZAF and ZAF(Ser) was obtained (Supplementary Fig. 5), which suggested that maybe not 100%, but the majority of O in EDS mapping image came from the incorporated serine inside ZAF(Ser). In ZAF, even with much longer scanning time, very low signal intensity of O was collected, which can be reasonably considered as the background noise. Thus, the uniform distribution of O in the EDS mapping image of ZAF(Ser) implied the homogeneous distribution of serine inside. To be more precise, we have revised the expression in the revised manuscript.

Revision:

Manuscript, Page 7-8, Line 132-138:

“To eliminate that the possibility that O came from the adsorbed H₂O of the air, EDS mapping of ZAF and ZAF(Ser) by using much longer scanning time to obtain stronger signals was carried out, which suggested that without serine incorporated, the signal of O observed in ZAF was extremely low and can be considered as background noise (Supplementary Fig. 5). Thus, the uniform distribution of O in ZAF(Ser) suggested homogeneous dispersion of serine throughout the ZAF(Ser).”

Supplementary Information, Page S103-104:

“To ensure that all O in the EDS analysis came from serine only and not H₂O adsorbed from the air, we carried out experiments very carefully to avoid the long time contact of the sample with air before TEM observation and EDS mapping analysis.

For sample Storage, the freshly synthesized ZAF(Ser) were vacuum-dried overnight at 60 °C and then transferred to a sealed vacuum desiccator for storage at room temperature.

During the preparation of TEM samples, the aforementioned ZAF(Ser) powder was dispersed in anhydrous ethanol through a clean ultrasonic probe. Then, One drop containing ZAF(Ser) were placed on the carbon film and stayed for 15-20s and the excess ethanol was removed carefully using filter paper strip. After drying, the TEM sample was quickly transferred a vacuum desiccator for temporary storage to avoid adsorption of water from the air.

During the sample transfer process, the samples were removed from the vacuum desiccator and immediately transferred to the transmission electron microscope loading tank.”

Supplementary Information, Page S6:

Supplementary Fig. 5. HAADF-STEM images of (a) ZAF and (b) ZAF(Ser) and corresponding EDS elemental mapping images of O with prolonged scanning time 240s.

Comment 11 II. *The surface areas presented in sup. Table 4 are very small, and no pores are present in some MOF materials (the insert in sup figure 17 has no axis), as such, is a porous MOF structure really formed? Even with analogues ZIF materials, the surface area is expected to be around 1500 m²/g with well defined pores, and clear internal and external surface areas detected. If surface area is going to be used to explain differences in reactivity, it should be presented for every material. PXRD should be presented along with the calculated pattern of the MOF structure to ensure the expected MOF is formed.*

Response: Thank you for your suggestion. The relationship between surface areas and catalytic activity has been discussed in-depth, which concluded that **surface area was not able to explain the activity difference between difference ZAF composites. We kindly ask the reviewer to refer to our response to Comment 1-5 and Comment 6.** Despite this, the surface area of every material mentioned here has been added in the revised manuscript (Supplementary Fig. 21, Supplementary Fig. 47, Supplementary Fig. 51 and Supplementary Table 4). Also, PXRD patterns of different materials are added along with the calculated pattern of the MOFs as suggested (Supplementary Fig.24). The axis of the insertion image in Sup. Figure 17 was supplemented (Revised Supplementary Fig. 19).

We agree with the reviewer that some reported analogues ZIF materials exhibited much higher surface area with well defined pores, clear internal and external surface areas than ZAF reported here. However, there has not a strict definition on MOFs and the limit of surface area yet. In fact, the surface area of MOFs varied greatly depending on the ligand, synthetic methods, post-synthetic methods, crystallinity, etc. In this study,

BTA was chosen as the ligand due to its ability to leave the middle nitrogen atom empty, which facilitated the formation of hydrogen bond-mediated active sites. However, the obtained MOFs with p21/c cells synthesized by this class of triazole ligands exhibit essentially small surface areas, typically less than $300 \text{ m}^2/\text{g}^{1-4}$. Despite the low surface area, ZAF with clear composition and crystallinity exhibited good hydrolytic catalytic performance. Furthermore, engineering the second coordination inside ZAF led to remarkably enhanced catalytic activity and stability, which is anticipated to provide insights for biomimetic design of hydrolase and environmental remediation.

Reference

1. Loukopoulos, E. & Kostakis, G.E. Recent advances in the coordination chemistry of benzotriazole-based ligands. *Coordin. Chem. Rev.* **395**, 193-229 (2019).
2. Wang, X. L. et al. Bottom-up synthesis of porous coordination frameworks: apical substitution of a pentanuclear tetrahedral precursor. *Angew. Chem. Int. Ed.* **121**, 5395-5399 (2009).
3. Lata Singh, D., Mishra, V., Kumar Ghosh, T. & Rao, R. Hydrothermal Synthesis and Symmetrical Supercapacitor Study of 1D Ln-H₂PDA (Ln= La and Sm) Metal-Organic Frameworks. *ChemistrySelect* **7**, e202202076 (2022).
4. Li, Y.Z., et al. A separation MOF with O/N active sites in nonpolar pore for One-step C₂H₄ purification from C₂H₆ or C₃H₆ mixtures. *Chem. Eng. J.* **466**, 143056 (2023).

Revision:

Supplementary Information, Page S25:

Supplementary Fig. 24. PXRD patterns of ZAF(x) with varying amino acids as substitute of serine.

Supplementary Information, Page S20:

Supplementary Fig. 19. Pore size distribution of ZAF, ZAF(Ser), ZAF(Cys) and Ser@ZAF, respectively.

Comment 12 12. *The authors show that my heating the materials up, h-bonding is disrupted. When cooled down, is the activity restored? Or is the material structurally damaged at slightly elevated temperature when in the reaction conditions.*

Response: Thank you for your comment which helped further improve the quality of this study. For ZAF(Ser), when the temperature is over 70 °C, H-bonding is disrupted, leading to decreased catalytic activity (Fig. 3h). When cooled down, ZAF(Ser) is able to restore its catalytic activity due to the reversibility of hydrogen bonding (Supplementary Fig. 37), which again demonstrated the presence of hydrogen bonding and its vital role for activity. And **we carried out new experiment measuring the XRD patterns and TGA curves of the separated material after reaction under elevated temperature**, which showed the unchanged XRD pattern, indicating the structure integrity (Supplementary Fig. 37a) and higher thermal stability (Supplementary Fig. 37b). Brief description are shown below.

As shown in Fig. 3h, the catalytic activity of ZAF exhibited negligible variation when the reaction temperature was within the range of 20 to 80 °C, suggesting the high thermal stability of ZAF. However, in ZAF(Ser), when the reaction temperature was over 70 °C, it showed a drastic decrease of activity (70%) towards the hydrolysis of HPPA to 2.5 $\mu\text{M min}^{-1}$, almost identical to that of ZAF, indicating the disruption of hydrogen bonding under the elevated reaction temperature. That is to say, under this temperature, only Lewis acidic active site is present in ZAF(Ser), which is similar to the case of ZAF, thus almost identical catalytic activity between ZAF(Ser) and ZAF is observed.

Due to the reversibility of hydrogen bonding, when the heated ZAF(Ser) suspension cooled down, hydrogen bonding re-formed and catalytic activity of ZAF(Ser) restored. ZAF(Ser) was incubated in the aqueous solution without substrate at 80 °C for 1 h and then cooled down to room temperature. After cooling, precipitation was obtained by centrifugation and dried at 60 °C overnight and exposed to typical enzymatic assay with reaction temperature of 40 °C. The evaluation of

catalytic activity revealed that the catalytic activity of this treated ZAF(Ser) was identical to ZAF(Ser) without treatment (Supplementary Fig. 37), demonstrated the re-formation of the hydrogen bonding and the subsequent activity recovery.

After the high-temperature reaction, ZAF(Ser) was collected via centrifugation and drying for PXRD analysis. The results (Supplementary Fig. 38a) showed that ZAF(Ser) retained the same characteristic peak with high intensity as freshly prepared ones, indicating the intact structure of ZAF(Ser). In addition, TGA analysis showed that ZAF(Ser) began to decompose at 302 °C, implying its high thermal stability (Supplementary Fig. 38b).

Revision:

Manuscript, Page 16, Line 339-343:

“Benefitting from the reversibility of hydrogen bonding, when ZAF(Ser) was heated at 80 °C and then cooled down to room temperature and exposed to enzymatic assay, it restored all its catalytic activity (Supplementary Fig. 37), which maintained intact structure and high thermal stability (Supplementary Fig. 38). This result verified the presence of hydrogen bonding from another aspect.”

Manuscript, Page 21:

Fig. 3h. The presence and gradual disappearance of hydrogen bonds induced by temperature.

Supplementary Information, Page S38:

Supplementary Fig. 37. Catalytic activity of ZAF(Ser) at room temperature (100% reference), ZAF(Ser) with reaction temperature at 80 °C (reaction at 80 °C), and incubation of ZAF(Ser) at 80 °C and then cooled down to room temperature followed by enzymatic assay (incubated at 80 °C).

Supplementary Information, Page S39:

Supplementary Fig. 38. (a) PXRD pattern of cooled down ZAF(Ser) after incubation at 80 °C. (b) TGA analysis of cooled down ZAF(Ser) after incubation at 80 °C.

Comment 13 13. *How does the presence of amino acids affect the connectivity of the MOF – is it possible that the use of serine for example, increases missing linkers, and the additional h-bonding and reactivity is contributed to by residual solvent molecules such as methoxy groups bound to the cluster rather than the amino acid – this would also give the O signal in EDX.*

Response: Thank you for your valuable comments which helped improve the quality of this research. **We have added more discussion about how the presence of amino acids affected the connectivity of the MOFs as suggested**, as introduction of a second ligand during MOFs preparation inevitably lead to detachment of the first ligand. **According to your comment, we have also carried out new experiments, which excluded the possible effect of residual solvent molecules inside ZAF on the catalytic activity (Supplementary Figs. 26-29).** Combining all these experiments and calculations, it pointed to our assumption that the resultant activity enhancement of the obtained ZAF(Ser) originated from the hydrogen bonding involved active site, with the hydrogen bonding formed between the coordinated serine and BTA. A brief response is provided below.

During the synthetic and washing process of ZAF(Ser), deionized water (H₂O) and methanol (CH₃OH) which contained the O element were used. Thus, their possible residues inside the obtained ZAF(Ser) powder were examined separately.

First, deionized water (H₂O) used during the synthetic process of ZAF(Ser) was replaced with deuterium water (D₂O), which followed by washing with anhydrous methanol and drying. PXRD pattern and FT-IR spectrum (Supplementary Fig. 26) of the obtained ZAF(Ser) did not showed any change. No signal peak of D₂O at 4.79 ppm in the ¹H NMR spectra of the obtained ZAF(Ser) was observed (Supplementary Fig. 27), indicating that negligible presence of D₂O inside the obtained ZAF(Ser). This ZAF(Ser) did not show activity change (Supplementary Fig. 28). Moreover, freshly prepared ZAF without washing process exhibited identical activity to the one following regular washing process, further demonstrated that even with residual H₂O inside, it did not lead to activity enhancement. Thus, the possibility of residual solvent H₂O used during the preparation of ZAF that lead to activity enhancement was excluded.

Secondly, a freshly synthesized ZAF(Ser) was washed with deuterated methanol (CD₃OD) for 1 and 5 times respectively, instead of using methanol in our regular washing process and then dried overnight at 60 °C. The obtained ZAF(Ser) was acid-digested and then detected by ¹H NMR, which did not show any characteristic peak of CD₃OD at 3.31-3.34 ppm (Supplementary Fig. 29a), indicating the absence of solvent molecules inside the composite. PXRD patterns of ZAF(Ser) with CD₃OD as the washing solvent for 5 times further demonstrated the good structural integrity (Supplementary Fig. 29 b). Activity measurement showed that both samples exhibited almost identical activity to the one washed with 5 times of methanol (Supplementary Fig. 28). Thus, the possibility that the residual solvent CH₃OH inside ZAF that lead to activity enhancement due to possible formation of CH₃OH and BTA was also excluded.

Combining the negligible presence of solvent molecules inside and also the activity measurement, the possibility that activity increase of ZAF originated from the hydrogen bonding between the ligands and the residual solvent molecules but not amino acids was eliminated, which also verified that O in EDS mapping image is from the incorporated serine.

Revision:

Manuscript, Page 13-14, Line 258-266:

“Before going further, the possibility that formation of hydrogen bonding resulting from the residual solvent molecules but not the incorporated amino acids was eliminated by using deuterium water (D_2O) instead of water used during the synthesis of ZAF(Ser) and deuterated methanol (CD_3OD) instead of methanol used during the washing process (Supplementary Figs. 26-29). Results showed that even with residual solvent inside, the unwashed ZAF did not show any activity enhancement (Supplementary Fig. 29). And after washing process, the amount of solvent molecule left inside the catalyst is negligible (Supplementary Note 2).”

Supplementary Information, Supplementary Note 2, Page S93-94:

“Experiments was carried to exclude the possible effect of residual solvent molecules inside ZAF on the catalytic activity. First, deionized water (H_2O) used during the synthetic process of ZAF(Ser) was replaced with deuterium water (D_2O), which followed by washing with anhydrous methanol and drying. PXRD pattern and FT-IR spectrum (Supplementary Fig. 26) of the obtained ZAF(Ser) did not showed any change. No signal peak of D_2O at 4.79 ppm in the 1H NMR spectra of the obtained ZAF(Ser) was observed (Supplementary Fig. 27), indicating that negligible presence of D_2O inside the obtained ZAF(Ser). This ZAF(Ser) did not show activity change (Supplementary Fig. 28). Moreover, freshly prepared ZAF without washing process exhibited identical activity to the one following regular washing process, further demonstrated that even with residual H_2O inside, it did not lead to activity enhancement. Thus, the possibility of residual solvent H_2O used during the preparation of ZAF that lead to activity enhancement was excluded. Secondly, a freshly synthesized ZAF(Ser) was washed with deuterated methanol (CD_3OD) for 1 and 5 times respectively, instead of using methanol in our regular washing process and then dried overnight at 60 °C. The obtained ZAF(Ser) was acid-digested and then detected by 1H NMR, which did not show any characteristic peak of CD_3OD at 3.31-3.34 ppm (Supplementary Fig. 29a), indicating the absence of solvent molecules inside the composite. PXRD patterns of ZAF(Ser) with CD_3OD as the washing solvent for 5 times further demonstrated the good structural integrity (Supplementary Fig. 29 b). Activity measurement showed that both sample exhibited almost identical activity to the one washed with 3 times of methanol (Supplementary Fig. 28). Thus, the possibility that the residual solvent CH_3OH inside ZAF that lead to activity enhancement due to possible formation of CH_3OH and BTA was also excluded.

Combining the negligible presence of solvent molecules inside and also the activity measurement, the possibility that activity increase of ZAF originated from the hydrogen bonding between the ligands and the residual solvent molecules was eliminated.”

Supplementary Information, Page S27:

Supplementary Fig. 26. PXRD pattern (a) and FT-IR spectrum of ZAF(Ser) obtained with mixture of DMF and D_2O instead of H_2O as solvents during the synthetic process.

Supplementary Information, Page S28:

Supplementary Fig. 27. The ^1H NMR spectra of ZAF(Ser) obtained with mixture of DMF and D_2O instead of H_2O as solvents during the synthetic process.

Supplementary Information, Page S29:

Supplementary Fig. 28. (a) The ^1H NMR spectra of ZAF(Ser) obtained by washing with CD_3OD for 1 and 5 times. (b) PXRD pattern of ZAF(Ser) obtained by washing with CD_3OD for 5 times.

Supplementary Information, Page S30:

Supplementary Fig. 29. Catalytic activity of ZAF(Ser) via using CD_3OD as the washing solvent, ZAF(Ser) obtained by using D_2O and DMF as the solvent during the synthetic process, ZAF separated from the synthetic system without washing process.

Comment 14 14. Comparing *sup.* Figures 3 and 4 the particle size of ZAF(*ser*) seems very different. Why is this?

Response: Thank you for careful observation. This size difference came from the different sample preparation process. Samples for TEM observation was prepared by dropping suspension of ZAF(Ser) onto the carbon film, where larger sized particle aggregation during the drying process is inevitable. Also, irregular stacking state intrinsic to ZAF(Ser) also led to non-uniform size. While during the DLS measurement,

the sample was prepared by dispersing the ZAF(Ser) powders in the solution via the ultrasonic dispersion method using an ultrasonic probe, which achieved a relatively smaller size particles.

Comment 15 15. It is unusual to give TGA conditions in terms of $K \text{ min}^{-1}$, $^{\circ}\text{C min}^{-1}$ is more commonly used. Why did the authors choose such a high rate of heating for the TGA experiments? And in sup Figure 9, the weight % increases above 100% under $100 \text{ }^{\circ}\text{C min}^{-1}$ – how did this happen? A slower temperature increase may have prevented that from occurring.

Response: Thank you for your careful observation and the kind suggestion. As suggested by the reviewer, we have changed the unit of “ $K \text{ min}^{-1}$ ” to “ $^{\circ}\text{C min}^{-1}$ ”. Usually, a heating rate between $1 \text{ }^{\circ}\text{C min}^{-1}$ to $50 \text{ }^{\circ}\text{C min}^{-1}$ is chosen for TGA analysis. Therefore, a heating rate of $20 \text{ }^{\circ}\text{C min}^{-1}$ was chosen in the study, which however, led to weight increase possibly ascribed from the higher heating rate and the buoyancy effect due to air disturbance. Following your helpful suggestion, we have adopted a slower heating rate ($10 \text{ }^{\circ}\text{C min}^{-1}$) and re-done the TGA analysis to prevent this from occurring. The new DSC-TGA curves (revised Supplementary Fig. 11) has been added in the revised manuscript.

Revision:

Supplementary Information, Page S12:

Supplementary Fig. 11. TGA and DSC analysis of ZAF(Ser).

Comment 16 16. The authors should make a bigger effort regarding the XAFS analysis. There are no details given and an actual discussion about the XAFS fits/spectra is missing. For example, in Figure 2g the authors affirm that ZAF and ZAF(Ser) have similar XANES spectra to ZnO. However, while ZAF appears similar to ZnO, ZAF(Ser)

25 / 67

has some additional features. The white line is indeed quite broad.

Response: Thank you for your valuable suggestion. We have added more details about XAFS analysis, including the experiment procedure, data processing and the corresponding software according to your suggestion. Also, more discussion about the XAFS fitting results and the spectra is also added in the revised manuscript and Supplementary Information.

Compared with ZnO with a single coordination environment, ZAF(Ser) which contained both coordination structures of Zn-N and Zn-O led to different chemical environment of Zn with slightly different energy levels, resulting in a broad white line. As indicated by the K-edge XANES spectra (Figure 2g), ZAF and ZAF(Ser) exhibited absorption edge with energy lower than that of ZnO, suggesting a positive valence of less than +2. Compared with ZAF, The K-edge absorption edge of ZAF(Ser) is closer to that of ZnO with high energy, indicating a slightly higher average valence state of Zn in ZAF(Ser) than that in ZAF.

Revision:

Manuscript, Page 10, Line180-187:

“Compared with ZnO with a single coordination environment, ZAF(Ser) which contained both coordination structures of Zn-N and Zn-O led to different chemical environment of Zn with slightly different energy levels, resulting in a broad white line. As indicated by the K-edge XANES spectra (Fig. 2g), ZAF and ZAF(Ser) exhibited absorption edge with energy lower than that of ZnO, suggesting a positive valence of less than +2. Compared with ZAF, The K-edge absorption edge of ZAF(Ser) is closer to that of ZnO with high energy, indicating a slightly higher average valence state of Zn in ZAF(Ser) than that in ZAF.”

Supplementary Information, Page S106-107:

“EXAFS fitting was performed by the Artemis module, following the EXAFS equation below:

$$\chi(k) = \sum_j \frac{N_j S_0^2 F_j(k)}{k R_j^2} \exp[-2k^2 \sigma_j^2] \exp\left[\frac{-2R_j}{\lambda(k)}\right] \sin[2kR_j + \phi_j(k)]$$

where S_0^2 is the amplitude reduction factor, $F_j(k)$ is the effective curved-wave backscattering amplitude, N_j is the number of neighbors in the j th atomic shell, R_j is the distance between the X-ray absorbing central atom and the atoms in the j th atomic shell, λ is the mean free path in Å, $\phi_j(k)$ is the phase shift, σ_j^2 is the Debye-Waller parameter of the j th atomic shell (variation of distances around the average R_j).

The obtained XAFS data was processed in Athena (version 0.9.26) for background, pre-edge line and post-edge line calibrations. Then Fourier transformed fitting was carried out in Artemis (version 0.9.26). The k^3 weighting, k -range of 3-12 Å⁻¹ and R range of 1-3 Å were used for the fitting of ZnO; k -range of 3-10.5 Å⁻¹ and R range of

1-2 Å were used for the fitting of samples. The four parameters, coordination number, bond length, Debye-Waller factor and E_0 shift (CN, R, ΔE_0) were fitted without anyone was fixed, the σ^2 was set.

For Wavelet Transform analysis, the $\chi(k)$ exported from Athena was imported into the Hama Fortran code. The parameters were listed as follow: R range, 1 - 4 Å, k range, 0 - 11 Å⁻¹ for samples; k weight, 3; and Morlet function with $\kappa=10$, $\sigma=1$ was used as the mother wavelet to provide the overall distribution.”

Comment 17 17. *Figure 2h: The authors should address why ZAF(Ser) has second-shell contribution, differently from ZAF. Moreover, why do these FTs look different from Figure S12? In the latter, the first peak for ZAF and ZAF(Ser) are at a similar radial distance; in contrast, the peaks' position in the main figure is very different.*

Response: Thank you for your suggestion. As shown in Figure 2h, the contribution of the second-shell in ZAF(Ser) is very small, which possibly resulted from the charge transfer between the vacant nitrogen and zinc. The different peak position in the main figure was caused by the misalignment of axis between ZAF and ZAF(Ser) due to our carelessness. We felt deeply sorry for this and have corrected the figure in the revised manuscript. Similar to Supplementary Fig. 14 (original Supplementary Fig. 12), in the normalized Fourier transform k^3 -weighted EXAFS (FT-EXAFS) spectrum, ZAF exhibited an obvious peak located at 1.52 Å (Fig. 2h in manuscript), which was attributed to the Zn–N scattering path. While a shoulder peak located at 1.54 Å in ZAF(Ser) was associated with the Zn-N and Zn-O scattering paths. Despite the higher electronegativity of O than N, the scattering path of Zn-O due to coordination of incorporated serine and Zn, and the scattering path of Zn-N due to coordination between BTA and Zn inside ZAF(Ser) is difficult to distinguish in EXAFS analysis, which showed slightly higher radial distance than pure Zn-N in ZAF.

Revision:

Manuscript, Page 9, revised Fig. 2h:

Fig. 2h. Fourier transform of k^3 -weighted Zn K-edge EXAFS fitting curves of ZAF(Ser) and ZAF.

Comment 18 *18. Errors should be expressed for all structural results (bond lengths, path degeneration, etc). The Experimental should clearly state the adopted EXAFS model, analysis protocol, and software. I suggest including a table containing all structural results derived from the fit, including the respective errors, to draw a clear comparison between the samples.*

Response: Thank you for your valuable suggestions to improve the quality of the manuscript. We have added a table containing all the structural results derived from fit including the respective errors. Also, more comparison and discussion between samples was include as suggested.

Revision:

Manuscript, Page 10-11, Line 178-199:

“To distinguish the electronic and coordination states of the zinc species, X-ray absorption near-edge structure (XANES) and extended X-ray absorption fine structure (EXAFS) analyses were carried out. Compared with ZnO with a single coordination environment, ZAF(Ser) which contained both coordination structures of Zn-N and Zn-O led to different chemical environment of Zn with slightly different energy levels, resulting in a broad white line. As indicated by the K-edge XANES spectra (Fig. 2g), ZAF and ZAF(Ser) exhibited absorption edge with energy lower than that of ZnO, suggesting a positive valence of less than +2. Compared with ZAF, The K-edge absorption edge of ZAF(Ser) is closer to that of ZnO with high energy, indicating a slightly higher average valence state of Zn in ZAF(Ser) than that in ZAF. In the normalized Fourier transform k^3 -weighted EXAFS (FT-EXAFS) spectrum (Supplementary Fig. 14), ZAF exhibited an obvious peak located at 1.52 Å, which was attributed to the Zn-N scattering path, and small scattering peaks derived from Zn-Zn coordination were also observed²⁵. Moreover, a shoulder peak located at 1.54 Å in ZAF(Ser) (Fig. 2h) was associated with the Zn-N and Zn-O scattering paths. The Zn-N/O coordination number in ZAF(Ser) was increased to 3.5, compared with 3.0 for ZAF (Fig. 2h, Supplementary Table 3), which was in good agreement with the XPS results. Furthermore, the higher intensity maximum at 4.95 Å⁻¹ in the k space associated with the Zn-N/O scattering path for ZAF(Ser) was observed in the EXAFS wavelet transforms (Fig. 2i and Supplementary Fig. 15). The contour intensity maximum for ZAF(Ser) was shifted slightly compared with that of ZAF due to the introduction of serine.”

Supplementary Information, Page S107-108:**Supplementary Table 3.** EXAFS fitting parameters at the Zn K-edge for various samples ($S_0^2=0.88$).

Sample	path	CN	R (Å)	σ^2 (Å ²)	ΔE_0 (eV)	R factor
ZnO	Zn-O	4*	1.96 ± 0.01	0.0042	3.5 ± 0.8	0.0020
ZAF(Ser)	Zn-N(O)	3.5 ± 0.3	1.98 ± 0.01	0.0089	-1.2	± 0.0145
					1.9	
	Zn-N(O)	3.0 ± 0.2	1.96 ± 0.01	0.0045		
ZAF	Zn-Zn	2.7 ± 0.4	3.22 ± 0.01	0.0072	3.9 ± 1.0	0.0119
	Zn-	8.7 ± 1.1	3.75 ± 0.02	0.0061		
	N(O)1					

CN: coordination number for the absorber-backscatter pair. R: the average absorber-backscatter distance. σ^2 : the Debye-Waller factor. ΔE_0 : the inner potential correction. R factor: goodness-of-fit.

Comment 19 **19.** *There is very little detail given on the DFT calculations given. It should be clearly explained how these calculations were performed. In particular, Figures S44 and S45 depicting the proposed mechanism are very unclear. They should be presented using chemical structures (similar to what is presented in figures S42 and S43) and not ball and stick model.*

In this context, I would recommend the paper to be rejected or, at least, to undergo major revisions, since key evidence is not supporting the conclusions.

Response: Thank you for your helpful suggestion. We have included the details about the DFT calculations in the revised Supplementary Information (Page S105-106) to clearly show how these calculations were performed according to your suggestion. We have also used chemical structure instead of ball and stick model in original Figures S44 and S45 to show the catalytic mechanism as suggested. (revised Supplementary Figure 58, 59).

Revision:**Supplementary Information, Page S59:**

Supplementary Fig. 58. Detailed calculation path diagram of ZAF(Ser) with Lewis acid mediated active site (Zn-OH active site with serine).

Supplementary Information, Page S60:

Supplementary Fig. 59. Detailed calculation path diagram of ZAF(Ser) artificial enzyme with hydrogen bonding mediated active site (Ser-O-H-N active site).

Manuscript, Page 36, Line 732-748:

“For calculation of the catalytic mechanism, the cluster configurations of ZAF(Ser) and ZAF were extracted from the crystal structure files and then optimized under the framework of density of functional theory (DFT) with the PBE0 functional and def2SVP basis set⁴⁰. To describe the restraints of these ligands in our real crystals, the nitrogen atoms coordinated with adjacent zinc ions were frozen in the calculations. The SMD (Solvation Model Based on Density) implicit solvent model was used to describe the solvation effect of water. The structures of the transition states and intermediates in the calculated reaction paths were optimized by using the same method. Vibrational frequency analysis was carried out for the optimized structure with the same functional and basis set. Thermodynamic correction terms of these structures at 298.15 K were obtained using the Shermo program. To obtain the electron energies with higher accuracy, which has a major impact on the accuracy of the Gibbs free energy, single point calculations for these optimized structures with the PBE0 functional and def2TZVP basis set³⁹ were performed. Finally, the single point energy was added to the free energy corrections calculated before to obtain the Gibbs free energies of all reactants, transition states, intermediates and products. The DFT calculations were performed using the Gaussian 16 B.01 program suite.

Supplementary Information, Page S107-108:

“The cluster configurations of the MOF *** were extracted from the crystal structure files and then optimized under the framework of density of functional theory (DFT) with PBE0 functional and def2SVP basis set. The convergence criteria for Self-Consistent Field (SCF) is 10^{-10} Hartree for the energy change and 10^{-8} for the maximum element of the density matrix. The default convergence criteria for optimization are to reach a maximum force of 0.00045 a.u. or less, reach a root mean square (RMS) gradient of 0.00030 a.u. or less, reach a maximum displacement of 0.00180 or less and reach a root mean square (RMS) displacement of 0.00120 or less. This means that the optimization will continue until the change in the gradients of all atoms in the molecule is less than 0.00045 atomic units (a.u.). However, this value can be adjusted by the user depending on the specific optimization problem and desired level of accuracy. In order to describe the restraints of these ligands in our real crystals, the nitrogen atoms which are coordinated with adjacent zinc ions were frozen in our calculations.”

Comment 20 *20. Improve the scholarly presentation of the paper. An overwhelming amount of data not essential to demonstrate the point is included, making the paper very difficult to read. Authors should select the materials that could be clearly demonstrated they are MOFs, and that actually contribute to prove their point. For example, statements such as “The generality of this design was confirmed by using different kinds ofazole derivatives,...” (from line 358) are clearly not fully supported by PXRD patterns on Sup Fig. 39, and as such just makes the manuscript more convoluted.*

Response: Thank you for your suggestion. Following your professional and helpful suggestions which helped further improved the quality of this research, we have made substantial revision about the manuscript, including carrying out new essential experiments and adding more discussion, re-organization of part of the contents, etc., which will better highlight the idea of this study. We agree with you that deletion of the part about using azole derivatives to demonstrate the generality of this design will not contribute much to the idea, which however make the manuscript convoluted and obscure. Thus, this part has been removed in the revised manuscript. The key idea of the paper is not affected after the revision, but with a better presentation and more easy for reading and understanding.

Comment 21 *21. In revising the paper, the authors should also:*

Provide surface area of all materials, and a clear comparison of reaction rates considering this property should be given.

Response: Thank you for your valuable comments. The surface areas of all materials have been provided in Supplementary Figs. 21, 47, and 51. We have also added a comparison of reaction rate considering the surface area as suggested. By using the surface area of each material to normalize its reaction rate of ZAF(x), the effect of surface area on catalytic activity was taken into consideration, which showed that the specific surface did not dominate the catalytic activity enhancement. **For more detailed response, we kindly ask the reviewer to refer to our response to Comment 1-5 and 6-7.** A precise description about the effect of surface area on catalytic activity is provided below.

As shown in Supplementary Table 4, amino acids with chemical structures similar to serine but bearing different side chains were investigated, including serine (-OH) for ZAF(Ser), cysteine (-SH) for ZAF(Cys) and alanine (-H) for ZAF(Ala). By normalizing the reaction rate with specific surface area, the catalytic activity of ZAF(Ser) per unit of surface area is 4.3-fold and 2.1-fold higher than that of ZAF(Cys) and ZAF(Ala), which again highlighting the key role of hydroxyl group from amino acid and the resultant formation of hydrogen bonding inside ZAF(Ser).

Revision:

Manuscript, Page 12-13, Line 244-258:

“Moreover, by normalizing the reaction rate with specific surface area, the catalytic activity of ZAF(Asp), ZAF(Glu) is almost identical to that of ZAF(Ser) (Supplementary Table 5), implying that change of amino acid with different electron-withdrawing capability and the resultant enhanced Lewis acidity did not lead to activity change of the obtained composite. This was further confirmed by the almost unchanged activity of ZAF(Ser) with varying ratio of cysteine to BTA during synthetic process to regulated the Lewis acidity of Zn in the final composite (Supplementary Fig. 25). Thus, the possibility of enhanced Lewis acidity after the hybrid coordination of amino acid that led to activity increase is ruled out. These results together indicated that the remarkably

higher activity of ZAF(Ser) cannot simply be ascribed to the defects and increased surface area due to incorporation of the small amino acids or the regulated Lewis acidity of the zinc ions in the primary coordination sphere but was generated by the structure of ZAF(Ser), which points to the presence of hydrogen bonds involving OH groups in serine and dangling N atoms from BTA, as inspired by the native enzyme.”

Supplementary Information, Page S84:

Supplementary Table 5. BET surface area and Surface-area-normalized reaction rate of ZAF composites under identical amount of zinc.

Entry	Sample	BET surface area (m ² g ⁻¹)	Surface-area-normalized reaction rate under identical amount of Zn (g nM min ⁻¹ m ⁻²)
1	ZAF	21.9	91.3
2	ZAF(Ser)	174.8	37.7
3	ZAF(Cys)	272.3	8.8
4	ZAF(Ala)	99.94	18.0
5	ZAF(His)	97.73	39.9
6	ZAF(Glu)	73.00	34.2
7	ZAF(Asp)	62.51	38.4

Comment 22 *22. Avoid comparisons with ZAF parent material. These are not relevant, given the large difference in surface area. Moreover, ZAF and ZAF(Ser) are materials with a very different nature, and claiming benefits to the design by comparing them are largely misleading.*

Response: Thank you for your comment. From the perspective of surface area, ZAF and ZAF(Ser) is quite different, as ZAF(Ser) exhibited much higher surface area. Yet this effect has been eliminated by normalizing the reaction rate with surface area according to aforementioned helpful suggestions (Supplementary Table 5). Other than this, ZAF(Ser) and ZAF shared a lot of similarity from the perspective of the structure and function. First, ZAF(Ser) was designed and fabricated via incorporating serine into ZAF by learning from the nature, both of which shared the similar XRD patterns, similar coordination microenvironment and hydrolytic function. As evidenced by the large amount of experiments and calculations in this study, the activity between ZAF(Ser) and ZAF is attributed to the slight structure difference, the hydrogen bonding in the second coordination sphere, which we learnt from enzyme engineering. In enzyme engineering, the introduction of a new amino acid into the existing protein framework usually led to remarkably enhanced catalytic activity or improved stability via very slight structure change in the second coordination sphere. This is realized by strengthening the interaction between the substrate and the amino acids in the active

sites¹ (*J. Phys. Chem. B* 2021, 125: 10682-10691). **Despite the fact that this idea has been adopted in complex organic protein molecules, it has not been fully exploited and realized in the case of inorganic artificial enzymes, especially MOFs-based hydrolase.** Learning from the nature, ZAF with hydrolytic activity was first designed by mimicking the catalytically active site of natural hydrolases, with Lewis acidic active site inside. Based on this, we further learning from the idea of protein engineering by introducing a new amino acid into the framework to modulate the catalytic microenvironment in the second coordination sphere. Similar to native enzyme, introduction of tiny amount of serine and the formation of a small amount of hydrogen bonding inside ZAF(Ser) led to significantly enhanced catalytic activity. As we responded in Comment 8, each mole of active site with hydrogen bonding involved contributed a reaction rate of $1.3 \times 10^6 \mu\text{M min}^{-1} \text{mol}^{-1}$, which is 11-fold higher than that of each mole of Lewis acidic active site from Zn ($1.2 \times 10^5 \mu\text{M min}^{-1} \text{mol}^{-1}$), which emphasized that weak molecular interaction inside enzyme molecules will improve its catalytic performance and this principle to engineering the second coordination sphere of MOFs based artificial enzyme is also feasible. From this perspective, ZAF and ZAF(Ser) are highly comparable.

Reference

1. Yan, B., et al. Rate-perturbing single amino acid mutation for hydrolases: a statistical profiling. *J. Phys. Chem. B* **125**, 10682-10691 (2021).

Revision:

Manuscript, Page 12-13, Line 229-248:

“Furthermore, ZAF(Cys) with cysteine substituted for serine was fabricated by following the same procedure (Fig. 2d, Fig 3b, Supplementary Fig. 19), which featured a 16 times larger surface area than that of ZAF but exhibited no evident difference in activity compared with ZAF under identical amount of zinc. Thus, increased surface area did not lead to activity enhancement, which was further confirmed by the case of alanine (Ala), where ZAF(Ala) bearing 5 times higher surface area exhibited almost identical catalytical activity to that of ZAF (Supplementary Fig. 21, Supplementary Table 4). Thus, the enhanced catalytic activity of ZAF(Ser) compared with ZAF can not be simply ascribed to the increased surface area. Furthermore, MOFs with missing linker or with electron-withdrawing group on the linker has been reported to increase the Lewis acidity of metal sites³¹. Thus, ZAF(x) with amino acids bearing various side chain groups other than OH was prepared (Supplementary Figs. 22-24). ZAF(Asp) and ZAF(Glu) with carboxylate groups showed activities similar to that of ZAF at identical Zn content, although the surface area was ~3 times higher than that of ZAF (Supplementary Table 4 and Supplementary Fig. 21). Moreover, by normalizing the reaction rate with specific surface area, the catalytic activity of ZAF(Asp), ZAF(Glu) is almost identical to that of ZAF(Ser) (Supplementary Table 5), implying that change of amino acid with different electron-withdrawing capability and the resultant enhanced Lewis acidity did not lead to activity change of the obtained composite.”

Comment 23 *23. Clarify the amounts of MOF material used in every catalytic reaction. Currently, experimental protocols are vague, and figures or text do not contain the information.*

Response: Thank you for your suggestion. The amounts of MOF material used in every catalytic reaction has been added in the experimental procedures as suggested in the revised manuscript and supplementary information.

Revision:

Manuscript, Page 35, Line 693-695

“6 mg ZAF(Ser) or natural enzyme of identical mass was added to 3 mL of the substrate solution to initiate the reaction”

Supplementary Information, Page S102,

“Presynthesized ZAF or ZAF(Ser) powders (6 mg) were dispersed in 1 mL reaction solution containing 3 mM HPPA. Then, 2 mL of different concentration of the serine solution (Tris-HCl, pH = 6.0) were added (3, 6, 9, 15, 30, 45 mM). The final concentration of HPPA was 1 mM.”

Supplementary Information, Page S81, S83, S88,

“Zn content (1.72×10^{-5} mol) in 6mg of ZAF is used as a reference.”

Comment 24 *24. Provide clearer details on how the catalytic results are presented, and discuss the limitations of their analysis. Several results are given based on the ratio between Ser and Zn (Fig 3b), while others are expressed in terms of an intriguing rate increment (3c), although the actual rate change is only 2- to 3-fold (Sup Table 5). In general, I appreciate the deep analysis of catalytic activity with respect to changes in structure. However, authors should give more details how these calculations have been done (no protocol is given). Without clearer explanations/details, it is very difficult to understand for example how/why the numbers in Fig 3b and 3c are different, and the moles of what the normalization is being based on.*

Response: Thank you for your comment which helped improve the quality of this manuscript. A clearer details on the catalytic results of Figure 3b and 3c was added as suggested. A precise explanation is that, in Figure 3b, **the overall activity comparison of each catalyst is based on identical amount of zinc under the same reaction condition**, similar to other researches. While in Figure 3c, we further examined **the catalytic activity contribution of each active site** in different catalysts, which highlightd the role of hydrogen bonding in the active site. Calculation details are shown below in brief.

The hydrolytic activity of ZAF originated from the only one kind of active site: the Lewis acidic Zn. While ZAF(Ser) contains 2 kinds of catalytic active sites: Lewis

acidic Zn, and the hydrogen bond-mediated catalytic active site. Commonly, the comparison of the catalytic activity is based on the apparent catalytic reaction rate detected under identical Zn content, as shown in Fig. 3b in the manuscript. As can be seen, despite the high structural similarity between alanine/cysteine and serine, ZAF(Ala) and ZAF(Cys) exhibited catalytic reaction rates almost identical to that of pure ZAF under the same amounts of zinc ions, due to the absence of hydroxyl group and thus absence of hydrogen bonding involved active site. **In addition, we prepared samples of ZAF with physically adsorbed serine, which exhibited no evident difference in activity compared with pure ZAF (Supplementary Fig. 42), suggesting that hydrolytic activity of ZAF stemming from the Lewis acidity of the metal active site was primarily regulated by the coordinated BTA, but not the incorporated amino acid.** Therefore, their activities originated from the zinc ions via the Lewis acid-mediated hydrolysis pathway, which eliminated that possibility that change of surface area and the Lewis acidity led to activity change. Thus, the higher catalytic reaction rate of ZAF(x) compared with ZAF is attributed to the amount of coordinated amino acids inside.

Based on this, the contribution of each Lewis acidic active site of Zn in ZAF(x) and ZAF is considered identical, which can be calculated via dividing the catalytic reaction rate of ZAF by the amount of zinc. By subtracting the activity contribution of zinc (amount of zinc multiplied by the activity contribution of each zinc active site) from ZAF(x), the total activity contribution from the amino acid involved active site (the incremental activity) is obtained. Thus, activity contribution of each amino acid involved active site was calculated via dividing the incremental activity by the amount of incorporated amino acids.

As shown in Figure 3c, the catalytic activity contribution of each active site is clearly shown. Each mole of Lewis acidic Zn showed a hydrolytic rate of $1.2 \times 10^5 \mu\text{M min}^{-1} \text{mol}^{-1}$, while each serine involved active site showed a hydrolytic rate of $1.3 \times 10^6 \mu\text{M min}^{-1} \text{mol}^{-1}$, which was 11-fold higher. Yet each alanine and cysteine involved active site showed negligible activity. On the contrary, each homoserine involved active site contribute a hydrolytic rate of $1.7 \times 10^6 \mu\text{M min}^{-1} \text{mol}^{-1}$.

Moreover, a series of a series of xSer@ZAF analogues with varying serine contents was prepared via postsynthetic modification, and the subscript x stands for the molar ratio of serine and zinc ions during the synthetic process. Calculation of activity showed a linear relationship between the incremental activity and the amount of serine incorporated (Supplementary Fig. 48).

Combining these experiment results, it can be concluded that hydroxyl group from the amino acid and the potential formation of hydrogen bonding inside the catalyst led to activity enhancement, which can be quantified. We believed that this is a more in-depth analysis which quantified the activity contribution of the first coordination sphere and the second coordination sphere in the same scaffold.

Revision:

Manuscript, Page 14, Line 275-285:

“Thus, activity of ZAF originated from the only one Lewis acidic active site of zinc, while activity of ZAF(x) can be divided into two kinds of active site, the Lewis acidic active site of zinc and the amino acid involved active site. The quantitative activity contribution of zinc was obtained by dividing the activity of ZAF by the amount of zinc. While the quantitative activity contribution of each amino acid involved active site was calculated by subtracting the contribution of zinc from the total activity of ZAF(x). Calculation result revealed that each serine or homoserine involved catalytic active site in ZAF(Ser) and ZAF(HSer) exhibited a $1.3 \times 10^6 \mu\text{M min}^{-1} \text{mol}^{-1}$ and $1.66 \times 10^6 \mu\text{M min}^{-1} \text{mol}^{-1}$ in hydrolysis rate, which was 11-fold and 14-fold higher than the catalytic activity of zinc ions serving as Lewis acidic active sites (Fig. 3c). While the activity contribution of cysteine and alanine involved active site was almost negligible.”

Manuscript, Page 19-20, Line 366-393:

“Furthermore, to eliminate the possibility that ZAF(Ser) presented in the form of ZAF adsorbed serine (denoted as $x\text{Ser}/\text{ZAF}$). Thus $x\text{Ser}/\text{ZAF}$ with various amount of serine as the adsorbate and ZAF as the adsorbent was prepared (Supplementary Figs. 42-43). Catalytic activity showed that the even with serine adsorption capacity of 75.1 mg g^{-1} , the catalytic activity of $x\text{Ser}/\text{ZAF}$ was identical to pure ZAF (Supplementary Fig. 42b), indicating that the enhanced catalytic activity of ZAF(Ser) compared with both ZAF and ZnSer can not be ascribed to simple mixture of ZAF and ZnSer particles or ZAF with serine adsorbed.

To quantify the contribution of the hydrogen bonding to the catalytic activity of ZAF(Ser), a series of $x\text{Ser}@/\text{ZAF}$ analogues with varying serine contents was prepared via postsynthetic modification, and the subscript x stands for the molar ratio of serine and zinc ions during the synthetic process. PXRD (Supplementary Note 3), FT-IR and TGA characterization revealed that ZAF(Ser) obtained via the one-step self-assembly process and two-step postsynthetic modification method shared the same structure (Supplementary Figs. 44-46). Notably, $x\text{Ser}@/\text{ZAF}$ featured lower surface area than that of ZAF (Supplementary Fig. 47 and Supplementary Table 9), but increasing the serine content in the composite led to proportional activity enhancements from $\text{Ser}@/\text{ZAF}$ (Fig. 3j, Supplementary Table 9-10), which corroborated the vital role of serine in hydrolysis. Careful calculations for ZAF(Ser) with varying contents of serine showed a linear relationship between the incremental activity and the amount of serine incorporated (Supplementary Fig. 48). The linear relationship between the serine content coordinated with zinc and the catalytic activity suggested that serine was involved in hydrogen bonding, which generated an entirely new catalytic active site independent of the Lewis acidic active site in the framework. This pathway is similar to that of the natural hydrolytic enzyme protease³⁵, but this hydrogen-bond mediated hydrolytic pathway³⁶ has rarely been reported for artificial enzymes.”

Manuscript, Page 21, revised Fig. 3c:

Fig. 3c Catalytic activity contribution of Zn in ZAF and amino acid incorporated including Ser in ZAF(Ser), HSer in ZAF(HSer), Ala in ZAF(Ala) and Cys in ZAF(Cys) under identical amount of zinc.

Supplementary Information, Page S43:

Supplementary Fig. 42. (a) Adsorption capacity of serine for ZAF at different times. (b) Hydrolysis rate of ZAF with different adsorption capacity.

Supplementary Information, Page S49:

Supplementary Fig. 48. Quantification relationship of the activity contribution of incorporated Ser with molar ratio of Ser to Zn in x Ser/ZAF under identical amount of Zn.

Comment 25 25. Consider removing comparisons with natural enzyme from the main text. Comparison with natural enzyme, though interesting, is not exactly surprising or highly relevant. It has been demonstrated before for other MOF materials designed to environmental remediation. If authors wish to keep it in the main text, equivalence of catalyst concentration should be not done by using equal mass amounts of both. Instead, the number of catalytic active sites in order to claim higher MOF activity should be used, as an enzyme has only one catalytic site, while the same mass of MOF could bear multiple ones.

Response: Thank you for your comment. Up to now, the active site of MOFs artificial enzyme is still far below natural enzyme due to the higher recognition and bind capability of native enzyme stemming from the much more complex spatial organization of amino acids. The emphasis of this research was not to claim a higher MOFs activity compared with native enzyme. The main idea lies that learning from enzyme engineering and structure of native enzyme, engineering the second coordination sphere via construction of weak interaction such as hydrogen bonding is expected to endow the parent MOFs with remarkably enhanced catalytic activity. The contribution of the hydrogen bonding involved active site exhibited 10-fold higher catalytic activity than the most common Lewis acidic active site. We hope that this study represents a proof-of-concept milestone in the biomimetic design of artificial enzyme. Compared with previous study, ZAF(Ser) exhibited satisfactory stability towards pH variations, thermal stability and higher tolerance towards organic solvents (Fig. 5a, 5b, 5c). More excitingly, ZAF(Ser) showed high degradation capability towards OTA in real samples including fermentation broth with high ionic strength and red wine with low pH, which highlighting its potential in practical environmental remediation scenarios.

Comment 26 26. Improve the discussion to reflect the advances of the paper. Currently, it stands as a paragraph more suited to the introduction, and offers few critical insights with respect to the paper's findings and concept.

Response: Thank you for your professional suggestion. We have re-organized the part of discussion to better reflect the advance of this paper and include the critical insights on the findings and concepts.

Revision:

Manuscript, Page 31, Line 634-658:

“Artificial enzymes are designed to overcome the challenges of low stability and limited operating range seen with native enzymes. Despite of the fact that regulation of Lewis acidity via the primary coordination sphere led to enhanced hydrolytic activity of the artificial enzyme, hydrogen bonding in the second coordination sphere, which is common in nature enzyme, is involved in attack on the carbonyl group, which plays a key role in the hydrolysis process and has not received enough attention. Yet, precise

organization of functional groups to generate suitable reaction microenvironments mimicking the active sites of native enzymes is challenging. Additionally, coupling of the Lewis acidic active site and the hydrogen bonding active site in a single scaffold has not yet been realized.

In this study, the obtained ZAF(Ser) with dual catalytic active sites incorporated was realized via a fine-tuned organization of ligands and amino acids around metal ions. In addition to Lewis acidity regulation of the metal ion in the primary coordination environment, formation of hydrogen bonds in the second coordination sphere enabled a more energy favorable catalytic profile, which endowed the obtained MOFs with higher activities than their counterparts installed with a single Lewis acidic active site. The developed ZAF(Ser) displayed high stability, recyclability, and a wide substrate range, and exhibited great potential in the field of environmental remediation including biological fermentation waste water and in healthcare-related applications. Given the abundance and structural versatility of amino acids and peptides, interactions among the biological motifs, organic ligands and metal ions is anticipated to provide sophisticated and elaborate microenvironment resembling that of native enzyme. Thus, this study provides a rational method for design of artificial enzymes and enzyme engineering, which hold promise for environmental and healthcare applications.”

Comment 27 27. Improve graphical displays, as they are generally too small, and difficult to read. A two-column format should be used, at least.

Response: Thank you for your suggestion for helping improve the presentation of this manuscript. We have enlarged the figures in the manuscript and provided figures with two-column format with higher resolution according to your suggestion. We kindly ask the reviewer to refer to the revised figures in the revised manuscript.

Reviewer #2 (Remarks to the Author):

Comment 1 This report by Wen-Yong Lou and co-workers reports some interesting catalytic activity from a mixed ligand preparation of Zn using benzotriazole and various amino acids, especially serine. Generally a lot of work has been done here, and the subject of the work is likely to be of interest to the community. However, in my view the authors have based a lot of their analysis on the basis of a somewhat flimsy hypothesis as to what the active material actually is here. From the evidence produced, I suspect the actual composition of "ZAF(Ser)" is more likely just a mixture of particles of a (known) non-porous coordination polymer Zn(btz) and a more active zinc serine complex.

The authors have fundamentally misinterpreted their PXRD and single crystal data - the PXRD in figure 2(c) does not "indicat(e) coexistence of the coordination structures of ZAF and ZnSer in the framework of ZAF(Ser)", it indicates coexistence of separate particles of ZnSer and ZAF(Ser)! PXRD does not work like that - if the active species had the structure that the authors propose (with partial replacement of a btz ligand with a Ser ligand within the same ZAF structure), the PXRD pattern of the bulk would either resemble that of pure ZAF or would take on an entirely new pattern based on another crystal structure with different symmetry.

The authors' own SC-XRD data show that the ZAF structure is essentially unchanged. Taking this hypothesis forwards it also explains the gas adsorption results - presumably the ZnSer and ZnCys structures have some porosity, while the ZAF doesn't (this is clear from the crystal structure - it is a close packed, non porous substance, and the "mesoporosity" of <30 m²/g is essentially negligible in MOF terms). Much of the remaining data then carries this hypothesis forwards but does nothing to disprove the alternative explanation - IR, elemental analysis etc are all done on bulk samples and would look the same with a hypothetical "ZAF(Ser) as they would with a mixture of ZnSer and ZAF particles in various ratios.

Response: We sincerely acknowledge your words of appreciation on our work. And thank you for your constructive suggestion. We strongly agree with the reviewer that it is essential to demonstrate what led to the activity enhancement of ZAF(Ser): the mixture of ZnSer and ZAF, or other possible structure, or the formation of a new active site with hydrogen bonding similar to that of a native enzyme as we proposed, which is vital for highlighting the findings of this study. We also agree that IR, elemental analysis and part of the other characterization technologies are all done on bulk samples, which cannot totally eliminate the possibility of ZAF(Ser) in the form of physical mixture of ZAF and ZnSer, as it would lead to similar results with our hypothesis. And as commented by Reviewer #1 "since studying molecular mechanistic features in heterogeneous catalysis is very challenging." **Despite this, the plentiful of experiments from different perspectives we have done including the structure characterization is believed to provide powerful evidence for demonstrating our hypothesis**, including (1) physical mixture of ZAF and ZnSer did not lead to activity enhancement (Fig. 3j and Supplementary Fig. 40); (2) amount of incorporated serine is

highly related with catalytic activity of the obtained ZAF(Ser) (Fig. 3j and Supplementary Fig. 48, Supplementary Table 2); (3) hydroxyl group from serine is vital for activity enhancement of ZAF(Ser) (Fig. 3b, Fig. 3c, Supplementary Fig. 52 and Supplementary Fig. 54); (4) reversibility of the hydrogen bond in ZAF(Ser) is highly correlated with its catalytic activity (Fig. 3h, Fig. 3i, and Supplementary Fig. 37); (5) structure modelling showing the formation of hydrogen bonding and (6) characterization of XPS and XAFS revealing the difference of chemical environment of Zn in ZAF(Ser), ZAF and ZnSer from a more microscopic level (Fig. 2f-2i).

Besides, following your helpful suggestion, **we have carried out a series of new experiments to further prove the composition of ZAF(Ser) and the presence of the active site we proposed**, including activity measurement of physical mixture of varying serine with ZAF (ZAF+Ser) and serine with ZAF(Ser) (ZAF(Ser)+Ser) (Fig. 3j), preparation of cysteine incorporated ZAF (ZAF(Cys)) with varying amount of cysteine introduced followed by enzymatic assay (Supplementary Fig. 25), activity measurement of physical mixture with varying ratio of ZAF to ZnSer (Supplementary Fig. 40), activity of physical mixture of ZAF and ZnSer under high temperature (Supplementary Fig. 41), activity assessment of serine adsorped ZAF(denoted as x Ser/ZAF) (Supplementary Fig. 42), to further support our hypothesis.

Also, we agree with the reviewer that partial replacement of BTA in ZAF with serine ligands to obtain ZAF(Ser) will not lead to substantial change of XRD patterns of ZAF, which has been confirmed by SCXRD data (Supplementary Fig. 32a in manuscript). The confused difference of XRD patterns of ZAF(Ser) is that ZAF(Ser) in the form of polycrystal used in the study (Fig. 2c) is based on the optimization result of the preparation condition, which is different from that of ZAF(Ser) single crystal. The amount of serine in ZAF(Ser) single crystal is lower than that of ZAF(Ser) used in the study (Supplementary Table 8), the latter of which inevitably contains tiny amount of ZnSer. However, this tiny amount of ZnSer did not contribute to the activity enhancement of ZAF as the activity of ZnSer with identical amount of Zn to that of ZAF is much lower (Supplementary Fig. 18). We have revised the expression about XRD analysis according to the suggestion from the reviewer to make it more accurate.

Combined all these experiment results together, it pointed to the structure of active sites we proposed that led to activity enhancement, but not simple mixture of pure ZAF and ZnSer. How we came to such a conclusion is illustrated in brief as below.

First of all, we have prepared ZAF and ZnSer separately. As shown in SEM images, quite different from the relatively flat surface and lamellar structure of ZnSer (Supplementary Fig. 39a), ZAF and ZAF(Ser) featured rough surface and irregular particle with size of ~ 200 nm (Fig. 2a, Supplementary Fig. 39b). The SEM images of pseudo-ZAF(Ser) (physical mixture of ZAF and ZnSer) clearly observed two different morphologies of ZAF and ZnSer (Supplementary Fig. 39c). Besides, enzymatic assay showed that ZAF alone or ZnSer alone exhibited remarkably lower hydrolytic activity than ZAF(Ser) under identical amount of Zn under the same reaction condition (Supplementary Fig. 16, $2.1 \mu\text{M min}^{-1}$ for ZAF, $0.9 \mu\text{M min}^{-1}$ for ZnSer, and $6.6 \mu\text{M min}^{-1}$ for ZAF(Ser)). Also, adding serine into the catalytic reaction system of either ZAF

or ZAF(Ser) did not lead to activity change (Fig. 3j in manuscript).

Taken the reviewer's suggestion into consideration, if we assumed that ZAF(Ser) is exclusively in the form of ZAF and ZnSer, denoted as pseudo-ZAF(Ser), based on the result of ICP-OES and elemental analysis, the molar ratio of ZAF to ZnSer is 9:1. This pseudo-ZAF(Ser) is then prepared by mixing powders of ZAF and ZnSer accordingly. Activity measurement showed that this pseudo-ZAF(Ser) showed a hydrolysis rate of $1.7 \mu\text{M min}^{-1}$, which is ~ 4 times lower than that of ZAF(Ser) ($6.6 \mu\text{M min}^{-1}$) under identical zinc amount (Supplementary Fig. 40 in manuscript). In addition, we further measured the catalytic activities of physical mixtures containing varying ratio of ZAF to ZnSer, by fixing the amount of zinc identical to that of ZAF(Ser). The catalytic activity of physical mixtures is a sum up of the activity of single ZAF and ZnSer (varying from $1.8 \mu\text{M min}^{-1}$ to $1.5 \mu\text{M min}^{-1}$), which was much lower than that of ZAF(Ser) (Supplementary Fig. 40 in manuscript). Furthermore, to eliminate the possibility that ZAF(Ser) presented as the form of ZAF adsorped serine, we additionally prepared the sample of using ZAF as the absorbent and the serine as the absorbate (denoted as xSer/ZAF). FTIR spectra and XRD patterns did not lead to appearance of new bands and crystallinity change respectively (Supplementary Fig. 42 in manuscript). Catalytic activity showed that the even with serine adsorption capacity of 75.1 mg g^{-1} , the catalytic activity of xSer/ZAF showed no variation compared with pure ZAF (Supplementary Fig. 42). Therefore, the enhanced catalytic activity of ZAF(Ser) compared with both ZAF and ZnSer can not be ascribed to simple mixture of ZAF and ZnSer particles or ZAF with serine adsorped, but pointed to the presence of the coordinated serine in the composite of ZAF(Ser).

Thus, to investigate the effect of serine on the catalytic activity of the obtained ZAF(Ser), we prepared two series of samples, 1) by varying the amount of serine during the in-situ incorporation process and fixing the amount of Zn and BTA (Supplementary Table 2); 2) by coordinating serine into the pre-formed ZAF via post-modification method (denoted as xSer@ZAF). The coordination of serine with zinc was confirmed by FTIR (Supplementary Fig. 44 in manuscript). PXRD patterns showed that xSer@ZAF exhibited the crystal structure of ZAF (Supplementary Fig. 45 in manuscript) only, suggesting the absence of nanoparticle of ZnSer. The simultaneous increase of catalytic activity and amount of serine incorporated at identical Zn content (Fig. 3j in manuscript) with almost unchanged surface area (Supplementary Fig. 47, Supplementary Table 9 in manuscript) suggested the presence of a new catalytically active site or the enhanced Lewis acidity/increased accessible Lewis acidic active sites after serine incorporation.

To eliminate the possibility of effect of Lewis acidity strength or active site number after the hybrid coordination of amino acid that led to activity increase, we prepared a series of ZAF(Cys) by using the cysteine with same structure with serine except the thiol group replacing hydroxyl group (Supplementary Fig. 25 in manuscript,). Activity measurement showed similar activity to that of ZAF (Fig. 3b in manuscript), despite the fact that cysteine was incorporated and surface area of ZAF(Cys) was 16 times higher than that of ZAF (Fig. 2d in manuscript,). Increased surface area did not led to activity enhancement was further confirmed by the case of alanine (Ala), where

ZAF(Ala) bearing 5 time higher surface area exhibited almost identical catalytical activity to that of ZAF (Supplementary Fig. 25 and Supplementary Table 4 in manuscript). In addition, x Cys@ZAF were prepared by high-temperature secondary doping of Cys into ZAF, which showed no significant change in catalytic activity compared with ZAF (Supplementary Fig. 54 in manuscript).

Similar activity was observed in ZAF(Glu) and ZAF(Asp) (Supplementary Table 4 in manuscript). Thus, activity enhancement of ZAF(Ser) can not be attributed to the enhanced Lewis acidity due to hybrid coordination or increased number of Lewis acid active sites due to larger surface area, but resulted from the generation of new active sites due to presence of serine.

To further confirm the importance of hydrogen bonding for activity enhancement, we first examined the necessity of the presence of hydroxyl group from amino acids in ZAF(Ser). Thus, cysteine (Cys) and alanine (Ala) with similar carbon chain **without hydroxy group**, ZAF(O-tBu-Ser) and ZAF(O-Ac-Ser) **with hydroxyl group shielded by other group**, threonine (Thr) and homoserine (HSer) **bearing one hydroxyl group** were used instead of serine during the synthesis to investigate the effect of the hydrogen bonding on the catalytic activity of ZAF(Ser). As expected, once the possibility of forming the hydrogen bonding disappeared, for example, in the case of Cys, Ala, O-tBu-Ser, and O-Ac-Ser, the obtained ZAF(x) composite showed similar catalytic activity with ZAF (Supplementary Fig. 52 and Supplementary Table 4), even if the incorporated amino acid enhanced the surface area and thus more active sites will be accessible to the substrates (Surface area of ZAF(Cys) and ZAF(Ala) were 16 and 5 times higher than that of ZAF). On the contrary, in the case of Thr and HSer, where presence of hydroxyl group possibly enabled the formation of hydrogen bonding, catalytic activity enhancement was observed even with surface area unchanged (Supplementary Table 4). Thus, the necessity of the presence of hydroxyl group and the resultant formation of hydrogen bonding for activity enhancement of ZAF(Ser) is confirmed.

Furthermore, the reversibility of hydrogen bonding was utilized to demonstrate our hypothesis. We carried out control experiments by disrupting hydrogen bonding under high temperature and using typical denaturing agent guanidine hydrochloride, both of which are commonly used in protein research. To the expectation, when the reaction system is carried out under high temperature (70 °C or above), the system containing ZAF(Ser) as the catalyst showed a remarkably activity decrease, with reaction rate of $2.5 \mu\text{M min}^{-1}$, almost identical to that of the case containing ZAF (Fig. 3h), which suggested the disruption of hydrogen bonding and the consequent disappearance of activity enhancement. Similar activity decrease of ZAF(Ser) was observed when adding guanidine hydrochloride into the reaction system to disrupt the hydrogen bonding (Fig. 3i) While reaction system containing ZAF as the catalyst showed no variation of catalytic activity as no hydrogen bonding was involved in this case. Due to its reversibility, reformation of the disrupted hydrogen bonding occurs once the temperature is reduced. As ZAF(Ser) was first incubated at high temperature (80 °C or above) and then cooled down, it would restored the catalytic activity as it did under room temperature (Supplementary Fig. 37).

As for the molecular simulation and XAFS characterization which demonstrated structure of the active site, we kindly ask the reviewer to refer to our response to Comment 16-18 from Reviewer #1.

Revision:

Manuscript, Page 8, Line 139-143:

“Powder X-ray diffraction (PXRD) pattern of ZAF(Ser) resembled that of ZAF after serine incorporated, which is consistent with previous studies^{17,18} (Fig. 2c), which was further validated by Fourier transform infrared (FT-IR) spectroscopy (Supplementary Fig. 6 and Supplementary Note 1). A small characteristic peak of ZnSer suggested the presence of the tiny amount of ZnSer nanoparticle inside.”

Manuscript, Page 11-14, Line 213-266:

“The vast difference in activities of ZAF(Ser) and ZAF was investigated in depth. It was speculated that, during the catalytic processes of artificial metalloenzymes, increased surface area or enhanced Lewis acidity at the metal active site would allow the artificial enzyme to accelerate hydrolysis of the amide substrate^{26,27}. Brunauer-Emmett-Teller (BET) specific surface areas of 174.8 m² g⁻¹ and 21.9 m² g⁻¹ were determined for ZAF(Ser) and ZAF, respectively, which is in agreement with previous studies that ZIFs with triazoles as the ligand exhibited essentially low surface area²⁸⁻³⁰. And mesopores with sizes in the range 2~12 nm were indicated by porosity distribution calculations performed with the DFT method (Fig. 2d, Supplementary Fig. 19 and Supplementary Table 4). Despite the relatively lower surface area of pure ZAF compared with ZAF(Ser), the mass transfer rate monitored with a fluorescein probe and confocal laser scanning microscopy showed negligible differences in fluorescein distribution and fluorescence intensity (inset of Fig. 2d and Supplementary Fig. 20), implying that mesopores in ZAF(Ser) and ZAF provided transport pathways for diffusion of substrates and thus facilitated entrance of the substrate into the accessible active site of the catalyst during the hydrolytic process. Furthermore, ZAF(Cys) with cysteine substituted for serine was fabricated by following the same procedure (Fig 3b, Supplementary Fig. 19, Fig. 2d), which featured a 16 times larger surface area than that of ZAF but exhibited no evident difference in activity compared with ZAF under identical amount of zinc. Thus, increased surface area did not lead to activity enhancement, which was further confirmed by the case of alanine (Ala), where ZAF(Ala) bearing 5 times higher surface area exhibited almost identical catalytical activity to that of ZAF (Supplementary Fig. 21, Supplementary Table 4). Thus, the enhanced catalytic activity of ZAF(Ser) compared with ZAF can not be simply ascribed to the increased surface area. Furthermore, MOFs with missing linker or with electron-withdrawing group on the linker has been reported to increase the Lewis acidity of metal sites³¹. Thus, ZAF(x) with amino acids bearing various side chain groups other than OH was prepared (Supplementary Figs. 22-24). ZAF(Asp) and ZAF(Glu) with carboxylate groups showed activities similar to that of ZAF at identical Zn content, although the surface area was ~3 times higher than that of ZAF (Supplementary Table

4 and Supplementary Fig. 21). Moreover, by normalizing the reaction rate with specific surface area, the catalytic activity of ZAF(Asp), ZAF(Glu) is almost identical to that of ZAF(Ser) (Supplementary Table 5), implying that change of amino acid with different electron-withdrawing capability and the resultant enhanced Lewis acidity did not lead to activity change of the obtained composite. This was further confirmed by the almost unchanged activity of ZAF(Cys) with varying ratio of cysteine to BTA during synthetic process to regulated the Lewis acidity of Zn in the final composite (Supplementary Fig. 25). Thus, the possibility of enhanced Lewis acidity after the hybrid coordination of amino acid that led to activity increase is ruled out. These results together indicated that the remarkably higher activity of ZAF(Ser) cannot simply be ascribed to the defects and increased surface area due to incorporation of the small amino acids or the regulated Lewis acidity of the zinc ions in the primary coordination sphere but was generated by the structure of ZAF(Ser), which points to the presence of hydrogen bonds involving OH groups in serine and dangling N atoms from BTA, as inspired by the native enzyme. Before going further, the possibility that formation of hydrogen bonding resulting from the residual solvent molecules but not the incorporated amino acids was eliminated by using deuterium water (D₂O) instead of water used during the synthesis of ZAF(Ser) and deuterated methanol (CD₃OD) instead of methanol used during the washing process (Supplementary Figs. 26-29). Results showed that even with residual solvent inside, the unwashed ZAF did not show any activity enhancement (Supplementary Fig. 29). And after washing process, the amount of solvent molecule left inside the catalyst is negligible (Supplementary Note 2).”

Manuscript, Page 18-20, Line 322-343:

“To substantiate this hypothesis, the hydrogen bonding interaction inside ZAF(Ser) was disrupted by exposing the catalyst to high temperature (Fig. 3h) and guanidine hydrochloride treatment (Fig. 3i), where guanidine hydrochloride is commonly used for protein denaturation. The catalytic activity of ZAF exhibited negligible variation when the temperature was within the range of 20 to 80 °C, suggesting the high thermal stability of ZAF. However, for ZAF(Ser), when the temperature exceeded 70 °C, the rate for hydrolysis of HPPA showed a drastic decrease (70%) to 2.5 μM min⁻¹, indicating that the enhancement contributed by serine disappeared for the intact structure of ZAF(Ser) (Fig. 3h). The drastic decrease in the activity of ZAF(Ser) and the identical catalytic activities for pure ZAF and ZAF(Ser) possibly resulted from the absence of hydrogen bonding between serine and BTA at higher temperatures. Hydrogen bonding between amino acid residues of the protein backbone stabilizes the three-dimensional structure of the native enzyme and determines the catalytic activity. Deactivation of the enzyme at elevated temperature is partly ascribed to the loss of hydrogen bonding, which led to collapse of the structure that had been stabilized by hydrogen bonding. As with deactivation of proteins, ZAF(Ser) failed to perform hydrogen bonding-mediated amide cleavage at high temperature due to breakage of the hydrogen bond. Benefitting from the reversibility of hydrogen bonding, when ZAF(Ser) was heated at 80 °C and then cooled down to room temperature and exposed to enzymatic assay, it restored all its catalytic activity (Supplementary Fig. 37), which

maintained intact structure and high thermal stability (Supplementary Fig. 38). This result verified the presence of hydrogen bonding from another aspect.”

Manuscript, Page 18-20, Line 354-412:

“The possibility that combination of ZAF and ZnSer (denoted as pseudo-ZAF(Ser)) that led to activity enhancement was also ruled out. The pseudo-ZAF(Ser) with molar ratio of ZAF to ZnSer of 9: 1 was first prepared by physical mixing the powders of ZAF and ZnSer based on the result of ICP-OES and elemental analysis of ZAF(Ser). Both the morphologies of ZAF and ZnSer (Supplementary Fig. 39) were observed in SEM images of pseudo-ZAF(Ser), which were completely different from ZAF(Ser) (Fig. 2a). Activity measurement showed that hydrolysis rate of pseudo-ZAF(Ser) was ~ 4 times lower than that of ZAF(Ser) ($6.6 \mu\text{M min}^{-1}$) under identical zinc amount (Supplementary Fig. 40). Furthermore, When this pseudo-ZAF(Ser) was used under high reaction temperature, no obvious activity change was observed (Supplementary Fig. 41), which is quite different from that of ZAF(Ser) (Fig. 3h), further implying the absence of hydrogen bond-mediated active sites in the pseudo-ZAF(Ser). Furthermore, to eliminate the possibility that ZAF(Ser) presented in the form of ZAF adsorbed serine (denoted as $x\text{Ser}/\text{ZAF}$). Thus $x\text{Ser}/\text{ZAF}$ with various amount of serine as the adsorbate and ZAF as the adsorbent was prepared (Supplementary Figs. 42-43). Catalytic activity showed that the even with serine adsorption capacity of 75.1 mg g^{-1} , the catalytic activity of $x\text{Ser}/\text{ZAF}$ was identical to pure ZAF (Supplementary Fig. 42b), indicating that the enhanced catalytic activity of ZAF(Ser) compared with both ZAF and ZnSer can not be ascribed to simple mixture of ZAF and ZnSer particles or ZAF with serine adsorbed.

To quantify the contribution of the hydrogen bonding to the catalytic activity of ZAF(Ser), a series of $x\text{Ser}@/\text{ZAF}$ analogues with varying serine contents was prepared via postsynthetic modification, and the subscript x stands for the molar ratio of serine and zinc ions during the synthetic process. PXRD (Supplementary Note 3), FT-IR and TGA characterization revealed that ZAF(Ser) obtained via the one-step self-assembly process and two-step postsynthetic modification method shared the same structure (Supplementary Figs. 44-46). Notably, $x\text{Ser}@/\text{ZAF}$ featured lower surface area than that of ZAF (Supplementary Fig. 47 and Supplementary Table 9), but increasing the serine content in the composite led to proportional activity enhancements from $\text{Ser}@/\text{ZAF}$ (Fig. 3j, Supplementary Table 9-10), which corroborated the vital role of serine in hydrolysis. Careful calculations for ZAF(Ser) with varying contents of serine showed a linear relationship between the incremental activity and the amount of serine incorporated (Supplementary Fig. 48). The linear relationship between the serine content coordinated with zinc and the catalytic activity suggested that serine was involved in hydrogen bonding, which generated an entirely new catalytic active site independent of the Lewis acidic active site in the framework. This pathway is similar to that of the natural hydrolytic enzyme protease³⁵, but this hydrogen-bond mediated hydrolytic pathway³⁶ has rarely been reported for artificial enzymes. In this case, the uncoordinated hydroxyl group in serine was anticipated to form hydrogen bonds with the vacant nitrogen dangling from the BTA ligand in the ZAF secondary coordination sphere. The activated

oxygen from serine acted as a nucleophile and attacked the carbonyl carbon of an amide bond, which accelerated amide bond cleavage.

In addition, the role of hydrogen bonding was further confirmed by using serine with OH shielded by acetyl, and t-butyl was adopted for the synthesis of ZAF(O-Ac-ser) and ZAF(O-tBu-ser) (Supplementary Figs. 49-50). ZAF(O-tBu-Ser) and ZAF(O-Ac-Ser) also showed a 3.1-fold and 3.4-fold higher surface area compared with ZAF (Supplementary Fig. 51 and Supplementary Table 4), but featured almost identical activity with ZAF under identical zinc amount (Supplementary Fig. 52). On the contrary, ZAF(Thr) with Thr bearing -OH groups (Supplementary Fig. 53) showed almost identical surface area with ZAF, but exhibited a 232% catalytic activity relative to ZAF. Furthermore, the catalytic results of *x*Cys@ZAF obtained following the postsynthetic modification that similar to *x*Ser@ZAF showed that even under high content of cysteine, *x*Cys@ZAF displayed almost identical catalytic activity to that of ZAF under identical amount of zinc (Supplementary Fig. 54 and Supplementary Table 11). This finding further corroborated the function of the OH group and the vital role of hydrogen bonding in the catalysis.”

Manuscript, Page 21:

Fig. 3h. The presence and gradual disappearance of hydrogen bonds induced by temperature.

Manuscript, Page 21,

Figure 3j. Catalytic activity of serine introduced in ZAF (Ser@ZAF), a physical mixture of serine and ZAF(Ser) (ZAF(Ser)+Ser) and a physical mixture of serine and ZAF (ZAF+Ser) with varying concentrations of serine.

Supplementary Information, Page S19:

Supplementary Fig. 18. Reaction rate of ZAF(Ser), ZAF, ZnSer, Zn, BTA, Ser and physically mixture of the building blocks under identical assay conditions.

Supplementary Information, Page S22:

Supplementary Fig. 21. Nitrogen sorption curves and pore size distribution of (a) ZAF(Ala), (b) ZAF(His), (c) ZAF(Asp) and (d) ZAF(Glu).

Supplementary Information, Page S26:

Supplementary Fig. 25. Catalytic activity of ZAF(Cys) with varying the molar ratios of BTA: Cys.

Supplementary Information, Page S38:

Supplementary Fig. 37. Catalytic activity of ZAF(Ser) at room temperature (100% reference), ZAF(Ser) with reaction temperature at 80 °C (reaction at 80 °C), and incubation of ZAF(Ser) at 80 °C and then cooled down to room temperature followed by enzymatic assay (incubated at 80 °C).

Supplementary Information, Page S40:

Supplementary Fig. 39. SEM images of (a) ZnSer, (b) ZAF, (c) physical mixture of ZnSer and ZAF nanoparticles (ZAF+ZnSer).

Supplementary Information, Page S41:

Supplementary Fig. 40. Catalytic activity of physical mixture of ZAF and ZnSer complex with different molar ratio of ZAF to ZnSer.

Supplementary Information, Page S42:

Supplementary Fig. 41. Catalytic activity of physical mixture of ZAF and ZnSer with high reaction temperature.

Supplementary Information, Page S43:

Supplementary Fig. 42. (a) Adsorption capacity of Ser for ZAF with different adsorption time. (b) Reaction rate of ZAF with different amount of Ser adsorbed (denoted as Ser/ZAF).

Supplementary Information, Page S44:

Supplementary Fig. 43. (a) PXRD patterns and (b) FT-IR spectra of Ser/ZAF with different adsorption time.

Supplementary Information, Page S48:

Supplementary Figure 47. Nitrogen sorption isotherms and pore size distribution for x Ser@ZAF obtained via post-synthetic modification.

Supplementary Information, Page S49:

Supplementary Fig. 48. Quantification relationship of the activity contribution of incorporated Ser with molar ratio of Ser to Zn in x Ser/ZAF under identical amount of Zn.

Supplementary Information, Page S52:

Supplementary Fig. 51. Nitrogen sorption curves and pore size distribution of (a) ZAF(HSer), (b) ZAF(Thr), (c) ZAF(O-Ac-Ser) and (d) ZAF(O-tBu-Ser).

Supplementary Information, Page S53:

Supplementary Fig. 52. Catalytic activity of ZAF(Ser), ZAF(Thr), ZAF(O-tBu-Ser), and ZAF(O-Ac-Ser).

Supplementary Information, Page S55:

Supplementary Fig. 54. Catalytic activity of x Cys@ZAF obtained via post-synthetic modification method.

Supplementary Information, Page S81:

Supplementary Table 2. Zn content determined by ICP-OES of ZAF(Ser) obtained by varying ratio of precursors and the corresponding catalytic activities under identical amount of Zn.

Molar ratio of Zn ²⁺ : BTA: Ser	Zn content (wt %)	Reaction rate ($\mu\text{M min}^{-1}$)	Relative activity (%)
4: 2: 2	24.4	4.0 ± 0.17	60.6
4: 2: 3	19.5	5.2 ± 0.63	78.8
4: 2: 4	22.1	4.8 ± 0.06	72.7
4: 3: 2	23.1	2.7 ± 0.12	40.9
4: 3: 3	37.3	4.6 ± 0.04	69.7
4: 3:4	28.3	5.6 ± 0.13	84.8
4: 4: 2	17.9	4.5 ± 0.24	68.2
4: 4: 3	28.3	5.7 ± 0.07	86.4
4: 4: 4	18.6	6.6 ± 0.21	100

Supplementary Information, Page S83:

Supplementary Table 4. BET surface area, pore volume of sample, zinc content and corresponding reaction rate of different ZAF composites.^a

Entry	Sample	Zn content (wt %)	BET surface area ($\text{m}^2 \text{g}^{-1}$)	Total pore volume ($\text{cm}^3 \text{g}^{-1}$) calculated by DFT method	Reaction rate under identical amount of Zn ($\mu\text{M min}^{-1}$)
0	ZAF	32.1	21.9	0.053	2.1 ± 0.04
1	ZAF(Ser)	18.6	174.8	0.256	6.6 ± 0.21
2	ZAF(Cys)	17.9	272.3	0.335	2.4 ± 0.44
3	ZAF(Ala)	19.8	99.94	0.161	1.8 ± 0.09
4	ZAF(His)	29.6	97.73	0.363	3.9 ± 0.23
5	ZAF(Glu)	20.1	73.00	0.0689	2.5 ± 0.34
6	ZAF(Asp)	29.1	62.51	0.2796	2.4 ± 0.03
7	ZAF(Hser)	31.5	97.40	0.3433	6.8 ± 0.66
8	ZAF(Thr)	29.3	23.69	0.0364	4.6 ± 0.08
9	ZAF(O-Ac-Ser)	33.3	73.51	0.3189	2.1 ± 0.03
10	ZAF(O-tBu-Ser)	29.9	67.80	0.3189	2.5 ± 0.23

a: Zn content (1.72×10^{-5} mol) in 6mg of ZAF is used as a reference.

Supplementary Information, Page S84:

Supplementary Table 5. BET surface area and Surface-area-normalized reaction rate of ZAF composites under identical amount of zinc.

Entry	Sample	BET surface area (m ² g ⁻¹)	Surface-area-normalized reaction rate under identical amount of Zn (g nM min ⁻¹ m ⁻²)
1	ZAF	21.9	91.3
2	ZAF(Ser)	174.8	37.7
3	ZAF(Cys)	272.3	8.8
4	ZAF(Ala)	99.94	18.0
5	ZAF(His)	97.73	39.9
6	ZAF(Glu)	73.00	34.2
7	ZAF(Asp)	62.51	38.4

Supplementary Information, Page S87:

Supplementary Table 8 Elemental analysis and ICP-OES of ZAF(Ser) powder and single crystal.

ZAF(Ser)	Zn (wt %)	N (wt %)	C (wt %)	Amount of Ser (wt %)	Molar ratio of Ser to Zn
Powder	18.6	28.1	47.3	6.1	0.2
Single crystal	22.2	26.6	46.5	2.7	0.091

Supplementary Information, Page S88:

Supplementary Table 9. The specific surface area and pore volume of *x* Ser@ZAF with amounts of serine doping in ZAF.^a

Sample	Amount of added serine (mmol g ⁻¹ ZAF)	BET surface area (m ² g ⁻¹)	Total pore volume (cm ³ g ⁻¹) calculated by DFT method	Reaction rate under identical amount of Zn (μM min ⁻¹)
ZAF	0	21.9	0.053	2.0 ± 0.06
1 Ser@ZAF	1	11.4	0.047	2.1 ± 0.02
2 Ser@ZAF	2	10.1	0.025	2.7 ± 0.08
3 Ser@ZAF	3	7.4	0.021	3.7 ± 0.10
5 Ser@ZAF	5	5.4	0.024	5.9 ± 0.12
10 Ser@ZAF	10	6.9	0.018	7.6 ± 0.40
15 Ser@ZAF	15	3.4	0.018	8.2 ± 0.46

a: Zn content (1.72×10^{-5} mol) in 6mg of ZAF is used as a reference.

Supplementary Information, Page S90:

Supplementary Table 11. Elemental analysis of *x* Cys@ZAF with amounts of serine doping in ZAF.

Sample	Amount of added Cys (mmol g ⁻¹ ZAF)	Zn (wt %)	N (wt %)	C (wt %)	S (wt %)	Molar ratio of Cys to Zn
1 Cys@ZAF	1	32.8	19.9	34.2	3.0	0.19
2 Cys@ZAF	2	32.4	17.1	31.8	5.5	0.34
3 Cys@ZAF	3	32.0	15.8	29.0	7.2	0.45
4 Cys@ZAF	4	31.8	13.5	26.6	9.9	0.64
6 Cys@ZAF	6	31.6	12.6	24.0	10.2	0.70

Comment 2 The catalytic activity seems to be higher in the mixture than with the individual components - this is an interesting element, but it is again not fully explained by the hypothesis put forward by the authors. The modelling of the active site looks nice, but neglects the extended structure of the material. We have already seen that ZAF is non-porous (this phase has been known since 1981), so presumably this is all happening on the particle surfaces? If so, the real active site could take any number of configurations.

Regardless of the potentially interesting catalytic activity of this mixture, the chemistry is not sufficiently characterised here and the authors have not sufficiently proven the identity of the active species, and so in my view this work is not yet ready for publication.

Response: Thank you for your suggestion. The enhanced catalytic activity of the obtained ZAF(Ser) and the mechanism has been demonstrated. **We kindly ask the reviewer to refer to our response to Comment 1** and the detailed revision listed. For the modelling of the active site, as this study aims at engineering the second coordination environment of zinc to generate the second catalytically active site with hydrogen bonding involved, which is quite different from the first coordination sphere with Lewis acidic active site, therefore, the focus of the modelling was to demonstrate the possibility of formation of the hydrogen bonding in the scaffold of ZAF. The formed structure of the materials was the extension of the duplicated active site. **The non-porosity of the catalyst is not directly related with the position of where catalysis occurs.** In fact, enzyme which is composed of hundreds of amino acids is non-porous. Yet, the huge difference between enzymatic catalysis and non-enzymatic catalysis lies that, enzymatic catalysis usually occurs in a deeply buried catalytic pocket composed of specifically organized amino acids sequence, but not on the surface with arbitrary active site structure. Beyond that, the adjacent amino acids near the catalytically active site in space also played an indispensable role for substrate binding via the hydrogen bonding, ionic interaction and vander Waals interaction. Therefore, the key catalytic element of an enzyme is composed of a series of amino acids with specific spatial organizations inside, which is accountable for the high activity and specificity.

In this study, we made an attempt to learn from the nature of enzyme and introduced the hydrogen bonding inside the ordered coordination structure of ZAF, though in tiny amount but accounted for a multifold catalytic activity enhancement compared with the parent material ZAF. The hydrogen bonding combined with the specific organization of zinc present in ZAF formed a new active site inside ZAF(Ser), which followed an enzyme-like catalytic process, and led to remarkable activity enhancement. A separate hydrogen bonding will not induce the catalytic process. And the separate coordination unit in ZAF showed much lower catalytic activity. Only inside the confinement of ZAF, the incorporation of hydrogen bonding will have the power to accelerate the catalytic process. These evidences together pointed to the catalytic process as we showed in the manuscript. Also, we carried out the simulation to calculate the energy barriers with these active sites, which confirmed our hypothesis.

Besides, previous study also confirmed that sole metal-amino acid complexes

without confinement or three-dimensional structure exhibited low catalytic activity, which is also verified in this study (ZnSer complex showed remarkably poor catalytic activity). These studies on the other hand supported that active site without elaborate space organization showed less catalytic activity. Therefore, the catalytic process of ZAF(Ser) is not attributed to surface catalysis. And we thank the reviewer for proposing such a professional question, which would guide us to investigate the function of adjacent coordination structure inside ZAF(Ser) for catalysis, a big topic deserves in-depth investigation.

Reviewer #3 (Remarks to the Author):

Comment 1 *The authors present the rational design of a new MOF with post-synthesis modifications at the second coordination sphere that enhance the catalytic degradation of C-N bond via hydrolysis. The authors are inspired by enzymatic analogues where specific functional groups (-OH) of amino acids increase the hydrolytic activity. After the presentation of the synthesis and a detailed characterization of the material, the authors are presenting mechanistic insights from computations. For the computed reaction mechanism, the authors have nicely frozen key atoms of the ZAF material to introduce the “stiffness” of the MOF in the geometry optimizations. Finally, they provide results on the degradation of ochratoxin A via hydrolysis promoted by ZAF(Ser). Overall, I believe that the experimental part of the article is excellent, but I do have concerns with respect to the computational work. For that reason, the authors should consider the following revisions and provide additional clarifications on the computational aspects of their work: In Figure 4, we see in (a) a $\text{Zn}(\text{triazole})_3(\text{H}_2\text{O})$, (b) a $\text{Zn}(\text{triazole})_2(\text{serine})(\text{H}_2\text{O})$, and in (c) a $\text{Zn}(\text{triazole})_3(\text{serine})$. I do not understand how Zn lost a triazole linker in b. Is this a demonstration of a defect? It is also unclear how the reaction profiles of Figures 4d and e are connected to the reaction cycles of Figures 4f and g, since there are no common labels between profiles and cycles. Please add the labels of the reaction intermediates in the cycle. It is also not clear why the authors decide to add a H_3O^+ on the second mechanism and a H_2O on the first. After all, the catalytic reaction with ZAF(Ser) is independent of the pH (Supplementary Figure 46.).*

Response: We deeply thank you for your positive comments and the appreciation on our research. And thank you for your professional suggestions and comments on the part of computational work, which will help improve the quality of this manuscript. As the reviewer mentioned the structure of active site, it is a defect when introducing serine into the parent work of ZAF. The structure of (a) $\text{Zn}(\text{triazole})_3(\text{H}_2\text{O})$, (b) $\text{Zn}(\text{triazole})_2(\text{serine})(\text{H}_2\text{O})$ focused on the Lewis acidic zinc in ZAF and ZAF(Ser), which followed the Lewis acid active site mediated hydrolytic process. The difference of these two Lewis acidic active sites lies in the possible different acidity strength due to the coordination of serine, which was taken into consideration during the calculation process. While structure of (c) $\text{Zn}(\text{triazole})_3(\text{serine})$ in ZAF(Ser) emphasized the hydrogen bond-mediated active site, which followed an enzyme-like catalytic process.

And following your kind suggestion, we have added the labels of the intermediates in the cycles which ensured a better comparison between profiles and cycles (revised Figure 4).

Despite the fact that inert amide bond can be hydrolyzed via either the Lewis acid mediated catalytic process or catalytic triad/dyad, the catalytic mechanisms of the two active sites in native enzyme varied a lot.

In the typical Lewis acid catalysis process, similar to the metallo-carboxypeptidases¹, Zn^{2+} combines with H_2O to form the $\text{Zn}-\text{H}_2\text{O}$ complex, activated

by the triazole group which allows the loss of the proton to form the Zn-OH⁻ active site. In the proposed metallohydrolase-like hydrolytic process (Fig. 4f). The substrate amide molecule coordinates to one Zn(II) ion, which is activated by water molecules interacting with the carbonyl oxygen (i, ii). Next, the hydroxide formed by deprotonation of a Zn²⁺-bound water (Zn-OH⁻) acts as a nucleophile and attacks the carbonyl carbon to form an intermediate (iii), followed by elimination of the leaving amine and subsequent displacement of the hydrolyzed product by water (iv and v), respectively.

Differently, the hydrogen bonding mediated catalytic process ZAF(Ser) is supposed to similar to that of catalytic triad/dyad in native enzyme², which follows the combination of acid-base catalysis and co-valent catalysis. In acid-base catalysis, a substance besides water molecules is needed to provide protons or accept protons, where H₃O⁺ provides protons for this catalysis, while the triazole group acts as a general base to accept protons, promoting the formation of the activated oxygen atom with strong nucleophilic ability in serine. As with the catalytic triad-enabled hydrolytic process of the enzyme, nucleophilic attack of the activated oxygen atom on the side chain of serine near the scissile amide carbonyl of the substrate occurs. Breakage of the C-N bond leads to release of the first product (amine). Water molecules from solution act as the nucleophiles, and the BTA triazole acts as the general base to regenerate the catalyst and liberates the acid product^{3,4}.

In addition, during the DFT calculation of the catalytic pathway, based on energy calculation, the addition of H₂O alone cannot form a thermodynamically stable intermediate, while the addition of H₃O⁺ successfully completes the entire reaction pathway. And this calculation result is also consistent with the theory of this reaction mechanism.

Reference

1. Xu, D. & Guo, H. Quantum mechanical/molecular mechanical and density functional theory studies of a prototypical zinc peptidase (Carboxypeptidase A) suggest a general acid-general base mechanism. *J. Am. Chem. Soc.* **131**, 9780-9788 (2009).
2. Richter, F. et al. Computational design of catalytic dyads and oxyanion holes for ester hydrolysis. *J. Am. Chem. Soc.* **134**, 16197-16206 (2012).
3. Wood, P. A., Allen, F. H. & Pidcock, E. Hydrogen-bond directionality at the donor H atom-analysis of interaction energies and database statistics. *CrystEngComm* **11**, 1563-1571 (2009).
4. Nothling, M. D. et al. Synthetic catalysts inspired by hydrolytic enzymes. *ACS Catal.* **9**, 168-187 (2018).

Comment 2 *Most importantly, both catalytic cycles are endothermic, and both involve reaction steps with very large energy barriers. In the first case, the reaction barriers of 39.2 and 42.0 kcal/mol are prohibitively large and in the second case, the reaction is be trapped on INT-2' since this is the more thermodynamically stable intermediate*

(more stable than reactants/products). In addition, experimentally, the authors report (low) product formation for ZAF, but the proposed mechanism with the 42.0 kcal/mol barrier is again prohibitively large.

Response: We sincerely thank you for your valuable comments, which help us to improve the manuscript. From a thermodynamic point of view, the intermediate with too high or too low energy will result in the stop of the whole reaction at a specific intermediate step. In the DFT calculations for this reaction, although INT-2' (-3.9 kcal mol⁻¹) is thermodynamically a more stable state compared to the reactants, this energy is able to trigger the formation of the transition state TS-3' in the next step, in agreement with the prediction. Therefore, INT-2' will not trap the whole the reaction to stop at this reaction stage. On the other hand, the present model is based on a local cluster, without the constraint of a crystal structure. The potential energy surface is not completely consistent with the real environment, which aimed at qualitatively reflecting the energy differences caused by the coordinated serine.

Similar cases of catalytic pathway calculations have been reported in previous studies. In the hydrolysis process of a native hydrolase, a more stable intermediate (-0.3 kcal mol⁻¹) than the reactants (2.7 kcal mol⁻¹) forms in the His-Asp-Ser catalytic triad of natural enzymes¹. And the reaction successfully goes on with the next intermediate or transition state with a reaction barrier of up to 17.6 kcal mol⁻¹. In another study which also aimed at the acylation and deacylation of peptidases², a stable intermediate of -20.2 kcal mol⁻¹ formed in the reaction, which subsequently converted to a transition state of 32.9 kcal mol⁻¹ with an energy barrier of 53.1 kcal mol⁻¹. In addition, in pathway of hydrolysis of C-N bond³, a high-energy of end-product (45.5 kJ mol⁻¹) is generated. Therefore, appearance of the intermediates with lower energies than the reactants/products during the reaction profiles did not suggest that the reaction will stop here.

As for the high energy barrier in the first case, previous study have demonstrated that an energy barrier as high as 48.5 kcal mol⁻¹ during the hydrolysis of amide bond calculated by DFT is able to proceed in native hydrolase, which follows a Lewis acidic active site mediated process⁴. Similarly, in the self-assembled peptide nanozyme with like-esterase activity⁵, the energy barrier of ester hydrolysis calculated by DFT is as high as 40 kcal mol⁻¹, which follows a Zn-Lewis catalysis process. Thus, in this study, a lower energy barrier (42 kcal mol⁻¹) of this reaction is supposed to be feasible.

We have included more discussion about this to make it clear and included new references in the revised manuscript.

Reference

1. Lešćić Ašler, I., Štefanić, Z., Maršavelski, A., Vianello, R. & Kojić-Prodić, B. Catalytic Dyad in the SGNH Hydrolase Superfamily: In-depth Insight into Structural Parameters Tuning the Catalytic Process of Extracellular Lipase from *Streptomyces rimosus*. *ACS Chem. Biol.* **12**, 1928-1936 (2017).
2. Cheng, Q. & DeYonker, N.J. Acylation and deacylation mechanism and kinetics of penicillin G reaction with streptomyces R61 DD-peptidase. *J. Comput. Chem.* **41**,

1685-1697 (2020).

3. Ion, B.F., Meister P.J. & Gauld, J.W. Multiscale Computational Study on the Catalytic Mechanism of the Nonmetallo Amidase Maleamate Amidohydrolase (NicF). *J. Phys. Chem. A* **123**, 7710-7719 (2019).
4. Serafim, L.F., Jayasinghe-Arachchige, V.M., Wang, L. & Prabhakar, R. Promiscuous Catalytic Activity of a Binuclear Metallohydrolase: Peptide and Phosphoester Hydrolyses. *J. Chem. Inf. Model.* **62**, 2466-2480 (2022).
5. Chen, Y. et. Al. Self-Assembled Peptide Nano-Superstructure towards Enzyme Mimicking Hydrolysis. *Angew. Chem.* **133**, 17301-17307 (2021).

Comment 3 *It would be nice if there is a direct comparison of the hydrogen bonding-mediated catalytic pathway with the biological equivalent reaction mechanisms of Supplementary Figure 42.*

Response: We sincerely appreciate your valuable comments. According to your suggestion, we have carried out new MD calculation by using catalytic triads (Revised Supplementary Fig. 56) which followed the hydrogen bonding-mediated catalytic pathway in native enzyme to enable a direct comparison of mechanisms with revised Supplementary Fig. 56. Also, the comparison of hydrogen bonding mediated catalytic process in revised Supplementary Fig. 59 was compared with revised Supplementary Fig. 56. More discussion was added in the revised manuscript as suggested.

The binding energy of three types of catalytic triad (**Asp-His-Ser**, **Asp-His-Cys**, **Asp-His-Ala**) with the substrates was calculated by molecular docking simulation to investigate the energy difference of active site with serine and serine mutated to cysteine and alanine in natural enzymes. As shown in Supplementary Fig. 60, compared with catalytic triad of Asp-His-Ser which bearing serine that contains hydroxy group, the binding energy of catalytic triad with substrate HPPA were $-8.8 \text{ kcal mol}^{-1}$ due to the formation of more hydrogen bonding involved interaction. When serine is mutated to cysteine and alanine, the binding energies was $-7.7 \text{ kcal mol}^{-1}$ and $-7.8 \text{ kcal mol}^{-1}$, respectively. The calculation indicates that the substrate binds more readily to the catalytic triad of Asp-His-Ser containing serine. The calculations for the catalytic triad in natural enzymes are consistent with our experiments obtained in ZAF(Ser), ZAF(Cys) and ZAF(Ala) (Figure 3), which further supported our assumption that engineering the second coordination sphere in MOFs based artificial enzyme is beneficial for activity enhancement.

In addition, in native enzyme, hydrogen bond deprotonated the hydroxyl group of serine to generate an oxygen anion in the Asp-His-Ser triad of enzyme. Water molecules act as the nucleophiles, and imidazole from histine acts as the general base in the catalytic reaction of enzyme. While in ZAF(Ser), the hydrogen bonding-mediated catalytic pathway is similar to catalytic mechanisms of ASP-His-Ser of enzyme. However, the difference is that H_3O^+ is used as the nucleophile and the triazole group is used as the general base to regenerate the catalyst and release the acid product.

Revision:

Manuscript, Page 26, Line 518-524:

“For comparison, the binding energy of typical catalytic triad (Asp-His-Ser) in native enzyme and its mutations of Asp-His-Cys and Asp-His-Ala with the substrates was calculated by molecular docking simulation (Supplementary Fig. 60). The calculation showed that that Asp-His-Ser containing serine and formation of hydrogen bonding is more energy preferred, which is consistent with the experiments obtained in ZAF(Ser), ZAF(Cys) and ZAF(Ala) (Fig. 3d, 3f and 3g).”

Supplementary Information, Page S61:

Supplementary Fig. 60. Representative conformations from MD simulations of catalytic triad in enzyme. Docking modes of the substrate hippuryl-L-phenylalanine (HPPA) to the Asp-His-Ser triad in native hdyrolase (a), Asp-His-Cys triad, with serine mutated to cysteine without hydroxyl group (b), and Asp-His-Ala triad, with serine mutated to alanine without hydroxyl group (c).

Comment 4 Finally, the statement “reduction of the energy barrier thus accelerates the reaction which is consistent with the EXAFS result” needs additional justification. This will help the authors to connect computational findings with experimental results.

Response: Thank you for your valuable advice which helped improve the manuscript. The reduction of energy barrier is determined by the structure of the catalytic active site and thus the different catalytic mechanism. The EXAFS results (Fig. 2h and 2i) indicate the presence of Zn-O coordination in ZAF(Ser), which guided us to investigated in-depth. Meanwhile, a series of experiments and calculations (Fig. 3 and Fig. 4, Supplementary Figs 18-58, Supplementary Tables 4-11) proved that zinc coordinated serine forms hydrogen bond-mediated active sites, which mimic the secondary coordination sphere environment of natural enzymes. As shown in Fig. 4d and Fig. 4f, the hydrogen bond-mediated active site is able to lower the energy barrier to enhance the catalytic activity.

We have supplemented and revised this statement to make it more accurate.

Revision:

Manuscript, Page 23, Line 451-454:

“The coordinated serine forms a hydrogen bond-mediated active site to reduce the energy barrier thus accelerating the reaction^{38,39}, which is consistent with the EXAFS result (Fig. 2h) and activity result of a series of ZAF based composites.”

Comment 5 *I believe that the title is too general and does not fully reflect the scope of the work presented in this manuscript.*

Response: Thank you for your valuable advice. We have revised the title to “Hydrolase mimic via second coordination sphere engineering in metal-organic frameworks for environmental remediation”

Comment 6 *Please change the section title “Creation and structural...” to “Synthesis ...” or “Formation...”.*

Response: Thank you for your suggestion. We have changed the section title as suggestion.

Revision:

Manuscript, Page 7, Line 115:

“Synthesis and structural characterization of ZAF(Ser)”

Detailed modifications can be found in the attached revised manuscript with yellow highlights. We greatly thank the Editor and Reviewers for further consideration our work. We believe that the presented modifications have made the manuscript clearer and more accurate and that it is now compliant with the high publication standards of Nature Communications.

Yours sincerely,

Xiaoling Wu, Jun Ge and Wenyong Lou

Reviewers' Comments:

Reviewer #1:

Remarks to the Author:

In the revised version of the manuscript the authors have addressed most of the critical points raised in my previous report. They have performed additional experiments and rewrote the manuscript accordingly. Although poor crystallinity of some materials remains an issue, the additional data, in particular those related to surface areas, support the central hypothesis of the paper. The revised version is more coherent with the main claim of the manuscript that the second coordination sphere in the MOFs can be engineered to enhance catalysis with MOFs to mimic natural hydrolases.

Reviewer #2:

Remarks to the Author:

Following the first round of reviewing, the authors have made significant improvements to their manuscript, and include a quite substantial body of work in further control experiments. I have no further suggestions for additional work - there is no doubt there is something interesting happening in the catalytic performance of this material. I am still not fully convinced that it is the mechanism the authors suspect it is, but they have done admirably in uncovering as much information as is reasonably possible here. In the interests of disseminating this result to the community for further study, I recommend publication without the need for further revision.

Reviewer #3:

Remarks to the Author:

The authors have performed a remarkable amount of revisions for addressing the criticism from all reviewers. They have performed new experiments and computations to support the initial hypothesis, and the scholar presentation of their work has been upgraded.

I am still not convinced about the feasibility of the reaction mechanism shown in Figure 4f. The intermediate INT-2' is significantly stable, and the reaction needs to pass from a barrier of 28.1 kcal/mol in order to reach a thermodynamically less stable product. In addition, key reaction steps and intermediates are missing from the reaction profiles. I refer to step (i) from the left-hand side of Figure 4f (O-H cleavage of a H₂O molecule) and step (v) from right-hand side Figure 4f. Those energies need to be added in the reaction profiles of Figures 4d and e.

Response to the reviewer comments for manuscript NCOMMS-23-05673B

Thank you very much for all the comments and suggestions from three reviewers, which greatly improved our manuscript. We provided a point-to-point response to all comments and listed all new experiments and revisions in blue font.

REVIEWER COMMENTS

Reviewer #1 (Remarks to the Author):

In the revised version of the manuscript the authors have addressed most of the critical points raised in my previous report. They have performed additional experiments and rewrote the manuscript accordingly. Although poor crystallinity of some materials remains an issue, the additional data, in particular those related to surface areas, support the central hypothesis of the paper. The revised version is more coherent with the main claim of the manuscript that the second coordination sphere in the MOFs can be engineered to enhance catalysis with MOFs to mimic natural hydrolases.

Response: Thank you for recognizing and supporting our first round of revisions. As the reviewer mentioned, the data added in the first round can support the central hypothesis of the paper, but there are some defects in the crystallinity of ZAF(Ala) and ZAF(His). Therefore, we tried to improve the crystallinity of these two materials by extending the synthesis time without changing the ratio of synthesized raw materials. The XRD results (Fig. 1a) showed a significant increase in the crystallinity of ZAF(Ala) when the synthesis time was extended to 4 days. When the synthesis time was extended to 6 days, ZAF(Ala) exhibited sharp characteristic peaks consistent with ZAF. In addition, FT-IR characterization (Fig. 1b) confirmed that both 4 day and 6 day synthesized ZAF(Ala) contained Zn-N (550 cm^{-1}) and Zn-O (435 cm^{-1}) characteristic peaks, and the two ligands were successfully coordinated to Zn. Activity measurement (Fig. 1c) showed that both samples exhibited almost identical activity to the amorphous ZAF(Ala) synthesized for one day. Although the synthesis time of ZAF(His) was extended to 6 days, its crystallinity was still unsatisfactory (Fig. 2), possibly due to the specific properties of the histidine ligand, which prevented the acquisition of highly crystallinity MOFs. However, as the reviewer notes, the additional data, particularly in relation to surface area, could supported the central hypothesis of the paper.

The XRD of ZAF(Ala) synthesized for 6 days of reaction is revised in **Supplementary Fig. 24**.

Fig. 1. (a) XRD patterns and FT-IR spectrum of ZAF(Ala) with synthesis time of 4 day and 6 day. (c) Catalytic activity of ZAF(Ala).

Fig. 2. XRD patterns of ZAF(His) with synthesis time of 4 day and 6 day.

Revision:

Supplementary Information, Page S25:

Supplementary Fig. 24. PXRD patterns of ZAF(x) with varying amino acids as substitute of serine (ZAF(Ala) with synthesis time of 6 days).

Reviewer #2 (Remarks to the Author):

Following the first round of reviewing, the authors have made significant improvements to their manuscript, and include a quite substantial body of work in further control experiments. I have no further suggestions for additional work - there is no doubt there is something interesting happening in the catalytic performance of this material. I am still not fully convinced that it is the mechanism the authors suspect it is, but they have done admirably in uncovering as much information as is reasonably possible here. In the interests of disseminating this result to the community for further study, I recommend publication without the need for further revision.

Response: We are grateful for the reviewer's efforts to improve the quality of our paper. Meanwhile, thank you very much for recognizing and supporting us in the first round of revisions.

Reviewer #3 (Remarks to the Author):

The authors have performed a remarkable amount of revisions for addressing the criticism from all reviewers. They have performed new experiments and computations to support the initial hypothesis, and the scholar presentation of their work has been upgraded.

I am still not convinced about the feasibility of the reaction mechanism shown in Figure 4f. The intermediate INT-2' is significantly stable, and the reaction needs to pass from a barrier of 28.1 kcal/mol in order to reach a thermodynamically less stable product. In addition, key reaction steps and intermediates are missing from the reaction profiles. I refer to step (i) from the left-hand side of Figure 4f (O-H cleavage of a H₂O molecule) and step (v) from right-hand side Figure 4f. Those energies need to be added in the reaction profiles of Figures 4d and e.

Response: We sincerely thank you for your valuable comments, which help us to improve the manuscript. From a thermodynamic point of view, the intermediate with too high or too low energy will result in the stop of the whole reaction at a specific intermediate step. In the DFT calculations for this reaction, although INT-2' (-3.9 kcal mol⁻¹) is thermodynamically a more stable state compared to the reactants, this energy is able to trigger the formation of the transition state TS-3' in the next step, in agreement with the prediction. Therefore, INT-2' will not trap the whole the reaction to stop at this reaction stage. On the other hand, the present model is based on a local cluster, without the constraint of a crystal structure. The potential energy surface is not completely consistent with the real environment, which aimed at qualitatively reflecting the energy differences caused by the coordinated serine.

Similar cases of catalytic pathway calculations have been reported in previous studies. In the catalytic process of Ni-MIL-127 (**Fig. 3**), a more stable intermediate (-4.54 kcal·mol⁻¹) than the reactants (9.08 kcal mol⁻¹) forms in the catalytic site of Ni-MIL-127 MOF¹. And the reaction successfully goes on with the next intermediate or transition state with a reaction barrier of up to 31.06 kcal·mol⁻¹ (130 kJ·mol⁻¹). Meanwhile, the free energies of the intermediates and products are -22.93 and -4.54 kcal mol⁻¹ respectively, but the intermediates can successfully cross the high energy barrier to produce the product. In another study which also aimed at alkane dehydrogenation reaction² (**Fig. 4**), a stable intermediate of -52.75 kcal·mol⁻¹ formed in the reaction, which subsequently converted to a transition state of -19.77 kcal·mol⁻¹ with an energy barrier of 32.98 kcal·mol⁻¹. Eventually, a more stable product (-25.98 kcal·mol⁻¹) is successfully generated than the reactant. Interestingly, in both reaction

pathways, the final product is less stable than the intermediate, but the reaction was completed successfully. Therefore, appearance of the intermediates with lower energies than the reactants/products during the reaction profiles did not suggest that the reaction will stop here.

For high-energy barrier reactions in this case to generate products that are thermodynamically less stable than the intermediates, previous studies have also demonstrated the possibility. An energy barrier as high as 23.4 kcal mol⁻¹ during the conversion of olefin calculated by DFT (**Fig. 5**) is able to proceed a product (14.3 kcal·mol⁻¹) less stable than the intermediate (2.6 kcal·mol⁻¹)³. Similarly, in the dehydrogenation of methanol to formaldehyde⁴, the energy barrier calculated by DFT is as high as 30.0 kcal·mol⁻¹ (**Fig. 6**), which generates a product that is less thermodynamically stable than the intermediate. Moreover, some other types of reaction pathways⁵⁻⁷ also have demonstrated DFT calculations are similar to the results in this paper. Thus, in this study, an energy barrier of 28.1 kcal·mol⁻¹ of this reaction is supposed to be feasible, which reach a thermodynamically less stable product compared with intermediate INT-2'.

We have included more discussion about this to make it clear and included new references in the revised manuscript.

The reaction mechanism in the left-hand side of Fig. 4f is the adsorption of H₂O onto the catalyst to form Cat-H₂O (step 1). Then, Cat-H₂O (*-H₂O) bound to the substrate to form a complex in the step 2 (INT-1). Zn²⁺ ion plays dual roles of polarizing both the substrate and the catalytic water molecule. In the third step, Zn-bound water molecule was activated by the Zn²⁺ ion, to generate Zn-OH⁻, which acts as the active nucleophile and attacks the carbonyl carbon of amide bond to form a transition state 1 (TS-1) and then intermediate (INT-2). In the fourth step, the water molecules attack the C-N bond of the substrate to form transition state 2 (TS-2). The amine is removed from TS-2 to form intermediate 3 (INT-3). Finally, the carboxylic acid-Zn dissociates to release the carboxylic acid (TS-3). **In the left-hand side of Figure 4f, the energy for O-H cleavage of a H₂O molecule is actually the difference between the energy of the INT-1 (*-H₂O-HPPA) intermediate to the transition state of TS-1 (*-OH-HPPA) (Supplementary Fig. 57).**

As for step (v) from right-hand side Figure 4f, this energy is actually the reaction process of intermediate INT-3' generated into TS-4' (Supplementary Fig. 59 and Fig. 4f in manuscript), which has been shown in detail in Fig. 4e of the manuscript.

Reference

1. Yeh, B. et al. Structure and Site Evolution of Framework Ni Species in MIL-127 MOFs for Propylene Oligomerization Catalysis. *J. Am. Chem. Soc.* **145**, 3408-3418 (2023).

- Solowey, D.P. et al. A new and selective cycle for dehydrogenation of linear and cyclic alkanes under mild conditions using a base metal. *Nat. Chem.* **9**, 1126-1132 (2017).
- Solans-Monfort, X., Coperet C. & Eisenstein O. Shutting Down Secondary Reaction Pathways: The Essential Role of the Pyrrolyl Ligand in Improving Silica Supported d⁰-ML⁴ Alkene Metathesis Catalysts from DFT Calculations. *J. Am. Chem. Soc.* **132**, 7750-7757 (2010).
- Li, H. & Hall, M.B. Role of the Chemically Non-Innocent Ligand in the Catalytic Formation of Hydrogen and Carbon Dioxide from Methanol and Water with the Metal as the Spectator. *J. Am. Chem. Soc.* **137**, 12330-12342 (2015).
- Willig, F. et al. Polyfunctional Imidazolium Aryloxy Betaine/Lewis Acid Catalysts as Tools for the Asymmetric Synthesis of Disfavored Diastereomers. *J. Am. Chem. Soc.* **141**, 12029-12043 (2019).
- Pannilawithana, N., Pudasaini, B., Baik, M.H. & Yi, C.S. Experimental and Computational Studies on the Ruthenium-Catalyzed Dehydrative C–H Coupling of Phenols with Aldehydes for the Synthesis of 2-Alkylphenol, Benzofuran, and Xanthene Derivatives. *J. Am. Chem. Soc.* **143**, 13428-13440 (2021).
- Yuan, P., Sun, Y., Xu, X., Luo, Y. & Hong, M. Towards high-performance sustainable polymers via isomerization-driven irreversible ring-opening polymerization of five-membered thionolactones. *Nat. Chem.* **14**, 294-303 (2022).

Fig. 3. Enthalpy (red) and free energy (blue) diagram for the synthesis of linear hexenes via propylene oligomerization through the Cossee-Arlman mechanism for

Ni-MIL-127. (Ref. 1)

Fig. 4. Computed Gibbs free energy profile for the dehydrogenation of cyclohexane along with optimized structures. (Ref. 2)

Fig. 5. Alkene metathesis pathway (formation and decomposition of the TBP metallacyclobutanes): energies of intermediates and transition states in kcal·mol⁻¹ with respect to ASBP (Y = OSiH₃, R₁ = Ph, R₂ = CH₃). (Ref. 3)

Fig. 6. Energetic profiles for the dehydrogenation of methanol to formaldehyde. (Ref. 4)

Revision:

Manuscript, Page 25, Line 504-506:

“DFT calculations of several studies⁴¹⁻⁴⁴ also confirm the rationality and feasibility of energy calculations in hydrogen bond-mediated catalytic reaction pathways.”

Supplementary Information, Page S58:

Supplementary Fig. 57. Proposed reaction pathway of ZAF with Lewis acid mediated active site (Zn-OH active site without serine).

Supplementary Information, Page S60:

Supplementary Fig. 59. Detailed calculation path diagram of ZAF(Ser) artificial enzyme with hydrogen bonding mediated active site (Ser-O-H-N active site).

Fig. 4. Proposed catalytic reaction pathways of ZAF(Ser). Schematic structure model of the Lewis acidic Zn(II) active site in ZAF (**a**), ZAF(Ser) (**b**) and the hydrogen bonds mediated active site in ZAF(Ser) (**c**). (**d**) Comparison of Gibbs free energy profiles for the catalytic processes of ZAF (blue dotted line) and ZAF(Ser) (red dotted line), which are analogous to those of metallohydrolases with Lewis acidic Zn(II)-mediated catalytic pathways. (**e**) Gibbs free energy profile for the catalytic process of ZAF(Ser) proceeding via the hydrogen bonding-mediated catalytic pathway. (**f**) Proposed catalytic mechanism of ZAF(Ser) involving the Lewis active site-mediated pathway (left) and hydrogen bonding-mediated pathway (right).

Reference

Manuscript, Page 43, Line 888-899:

41. Yeh, B. et al. Structure and Site Evolution of Framework Ni Species in MIL-127 MOFs for Propylene Oligomerization Catalysis. *J. Am. Chem. Soc.* **145**, 3408-3418 (2023).

42. Solowey, D.P. et al. A new and selective cycle for dehydrogenation of linear and cyclic alkanes under mild conditions using a base metal. *Nat. Chem.* **9**, 1126-1132 (2017).
43. Solans-Monfort, X., Coperet, C. & Eisenstein, O. Shutting Down Secondary Reaction Pathways: The Essential Role of the Pyrrolyl Ligand in Improving Silica Supported d⁰-ML₄ Alkene Metathesis Catalysts from DFT Calculations. *J. Am. Chem. Soc.* **132**, 7750-7757 (2010).
44. Li, H. & Hall, M.B. Role of the Chemically Non-Innocent Ligand in the Catalytic Formation of Hydrogen and Carbon Dioxide from Methanol and Water with the Metal as the Spectator. *J. Am. Chem. Soc.* **137**, 12330-12342 (2015).

Reviewers' Comments:

Reviewer #1:

Remarks to the Author:

The authors have revised the manuscript according to referee's suggestions, the improved version is suitable for publication.

Reviewer #3:

Remarks to the Author:

I would like to thank the authors for their detailed response. I do not have any further comments at this point, perhaps the reaction mechanism can be revised in the future. The manuscript stands at a great level so I believe that it should be published without any further revisions.

REVIEWERS' COMMENTS

Reviewer #1 (Remarks to the Author):

The authors have revised the manuscript according to referee's suggestions, the improved version is suitable for publication.

Response: Thanks very much for your kind work and consideration on publication of our paper.

Reviewer #3 (Remarks to the Author):

I would like to thank the authors for their detailed response. I do not have any further comments at this point, perhaps the reaction mechanism can be revised in the future. The manuscript stands at a great level so I believe that it should be published without any further revisions.

Response: We are grateful for the reviewer's efforts to improve the quality of our paper. Meanwhile, thank you very much for recognizing and supporting us in the round of revisions.